# A genetic and linguistic analysis of the admixture histories of the islands of Cabo Verde

**Romain Laurent[1], Zachary A Szpiech[2,3], Sergio S da Costa[1], Valentin Thouzeau[4,5], Cesar A Fortes-Lima[6], Françoise Dessarps-Freichey[1], Laure Lémée[7], José Utgé[1], Noah A Rosenberg[8], Marlyse Baptista[9,10], Paul Verdu[1]***

[1]UMR7206 Eco-anthropologie, CNRS-MNHN-Université Paris Cité, Paris, France; [2]Department of Biology, Pennsylvania State University, University Park, United States; [3]Institute for Computational and Data Sciences, Pennsylvania State University, University Park, United States; [4]UMR7534 Centre de Recherche en Mathématiques de la Décision, CNRS-Université Paris-Dauphine-PSL University, Paris, France; [5]Département d'Etudes Cognitives, Laboratoire de Sciences Cognitives et Psycholinguistique, ENS-PSL University-EHESS-CNRS, Paris, France; [6]Department of Organismal Biology, Sub-department of Human Evolution, Evolutionary Biology Centre, Uppsala University, Uppsala, Sweden; [7]Plateforme Technologique Biomics–Centre de Ressources et Recherches Technologiques (C2RT), Institut Pasteur, Paris, France; [8]Department of Biology, Stanford University, Stanford, United States; [9]Department of Linguistics, University of Michigan, Ann Arbor, United States; [10]Department of Afroamerican and African Studies, University of Michigan, Ann Arbor, United States

***For correspondence:**
paul.verdu@mnhn.fr

**Competing interest:** The authors declare that no competing interests exist.

**Abstract** From the 15th to the 19th century, the Trans-Atlantic Slave-Trade (TAST) influenced the genetic and cultural diversity of numerous populations. We explore genomic and linguistic data from the nine islands of Cabo Verde, the earliest European colony of the era in Africa, a major Slave-Trade platform between the 16th and 19th centuries, and a previously uninhabited location ideal for investigating early admixture events between Europeans and Africans. Using local-ancestry inference approaches, we find that genetic admixture in Cabo Verde occurred primarily between Iberian and certain Senegambian populations, although forced and voluntary migrations to the archipelago involved numerous other populations. Inter-individual genetic and linguistic variation recapitulates the geographic distribution of individuals' birth-places across Cabo Verdean islands, following an isolation-by-distance model with reduced genetic and linguistic effective dispersals within the archipelago, and suggesting that Kriolu language variants have developed together with genetic divergences at very reduced geographical scales. Furthermore, based on approximate bayesian computation inferences of highly complex admixture histories, we find that admixture occurred early on each island, long before the 18th-century massive TAST deportations triggered by the expansion of the plantation economy in Africa and the Americas, and after this era mostly during the abolition of the TAST and of slavery in European colonial empires. Our results illustrate how shifting socio-cultural relationships between enslaved and non-enslaved communities during and after the TAST, shaped enslaved-African descendants' genomic diversity and structure on both sides of the Atlantic.

## Editor's evaluation

This study leverages genetic and linguistic data from the islands of Cabo Verde, and provides a valuable example of how genetic ancestry patterns vary across admixed populations due in part to their unique local history and social practices of that time. The empirical and computational analyses supporting the claims of the authors are solid, and the tools developed will be useful for the study of genetically admixed individuals. The work will be of interest to human evolutionary biologists and anthropologists.

## Introduction

Between the 15th and 19th centuries, European colonization and the Trans-Atlantic Slave-Trade (TAST) put into contact groups of individuals previously isolated genetically and culturally. These forced and voluntary migrations profoundly influenced the descent of numerous European, African, and American populations, creating new cultures, languages, and genetic patterns (*Eltis and Richardson, 2015*; *Fortes-Lima and Verdu, 2021*).

Population geneticists have extensively described genetic admixture patterns in enslaved-African descendants in the Americas, and mapped their genomes for regions of ancestry recently shared with continental Africa and Europe (*Micheletti et al., 2020*; *Ongaro et al., 2019*). This allowed for reconstructing their detailed possible origins, as this knowledge is often intractable with genealogical records alone (*Eltis and Richardson, 2015*). Furthermore, genetic admixture-mapping methods have been used to identify genetic variation underlying phenotypic variation (*Winkler et al., 2010*; *Wojcik et al., 2019*), and to identify post-admixture natural selection signatures (*Patin et al., 2017*), thus revealing how admixture shaped human populations' recent evolution. Maximum-likelihood approaches based on linkage-disequilibrium (LD) patterns of admixed individuals *Gravel, 2012*; *Hellenthal et al., 2014* have repeatedly highlighted the diversity of admixture processes experienced by populations historically related to the TAST. In particular, they identified different European, African, and American populations, respectively, at the source of genetic admixture patterns, sometimes consistent with the preferred commercial routes of each European empire (*Gravel, 2012*; *Mathias et al., 2016*). Furthermore, they identified variable timing of admixture events during and after the TAST, sometimes consistent with major socio-historical events such as the expansion of the plantation economic system or the abolition of slavery (*Moreno-Estrada et al., 2013*; *Baharian et al., 2016*). From a cultural perspective, linguists have shown that novel contact-languages, such as creole languages (*Holm, 2000*; *Escure and Schwegler, 2004*), emerged from recurring interactions between socio-economically dominant Europeans and Africans and Americans. Furthermore, they identified the languages of origin of numerous linguistic traits in several creole languages (*Quint, 2000*; *Essegbey et al., 2013*; *Baptista, 2015*), and emphasized the complex histories of contacts that shaped language diversity on both sides of the Atlantic.

Numerous questions remain unsolved and novel interrogations have emerged concerning the history of admixture during and after the TAST. (*i*) While the genetic history of enslaved-African descendants in the Americas has been extensively studied, the influence of the TAST on genetic admixture in Africa remains under-investigated. Studying these questions in Africa would provide invaluable information about the influence of the onset and early stages of the TAST and the subsequent expansion of European empires on genetic admixture patterns on both sides of the Atlantic. (*ii*) While admixture-LD inference methods have repeatedly brought novel insights into the admixture processes experienced by enslaved-African descendant populations, they could only explore historical models with one or two pulses of admixture, a methodological limitation (*Gravel, 2012*; *Hellenthal et al., 2014*). Complex admixture histories may be expected as a result of the recurring flows of enslaved-Africans forcibly displaced between and within continents, changes of social relationships among enslaved and non-enslaved communities, and variable assimilation of new migrants in pre-existing communities, during and after the TAST (*Eltis, 2002*; *Berlin, 2009*). (*iii*) Finally, while the comparison of genetic and linguistic diversities has been the focus of numerous evolutionary anthropology studies at large geographical scales (*Creanza et al., 2015*; *Cavalli-Sforza et al., 1988*), it has rarely been endeavored for creole-speaking populations at a local scale in the historical context of the TAST (*Ansari-Pour et al., 2016*; *Verdu et al., 2017*; *Hagemeijer and Rocha, 2019*).

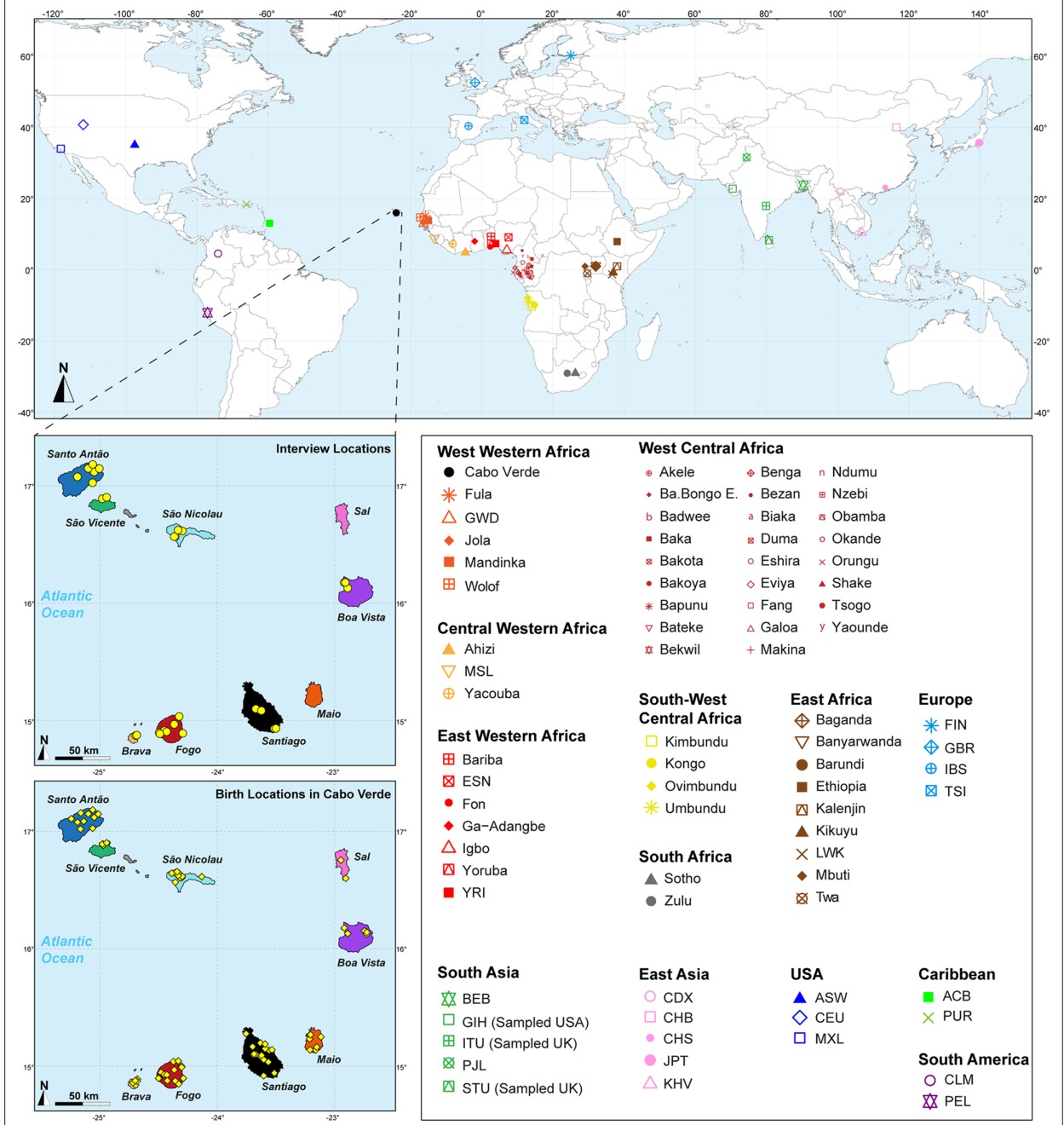

**Figure 1.** Sampling location of 233 unrelated Cabo Verdean individuals, merged with data on 4924 individuals from 77 worldwide populations. Birth-location of 225 individuals within Cabo Verde are indicated in the bottom map-panel, and birth locations outside Cabo Verde for 6 individuals are indicated in *Figure 1—source data 1*. Linguistic and familial anthropology interview, and genetic sampling for Cabo Verde participants were conducted during six separate interdisciplinary fieldworks between 2010 and 2018. Further details about populations are provided in *Figure 1—source data 1*.

The online version of this article includes the following source data for figure 1:

**Source data 1.** Population table corresponding to the map in *Figure 1* and sample inclusion in all analysis.

Here, we propose to reconstruct the detailed genetic and linguistic admixture histories of Cabo Verde, as this archipelago represents an ideal case to address these three understudied aspects of TAST history. First, Cabo Verde is the first European settlement-colony in Sub-Saharan Africa, located 500 kms West of Senegal in Western Africa (*Figure 1*), and settled in the 1460s by Portuguese immigrants and enslaved-Africans forcibly removed from the continental mainland. After 1492, and in particular after the 17th century expansion of the plantation economy in the Americas, Cabo Verde

served as a major slave-trade platform between continents (*Carreira, 2000*). Second, Cabo Verde forms an archipelago of nine inhabited islands that were settled over the course of three centuries due to the changing political, commercial, and migratory contexts (*Albuquerque and Santos, 1991*; *Albuquerque and Santos, 1995*; *Albuquerque and Santos, 2007*). Therefore, studying the admixture history of Cabo Verde will provide unique insights into the onset of the TAST before 1492, and into the history of slavery thereafter. This setting further promises to illustrate, at a micro-geographical scale, island-per-island, the fundamental socio-historical and serial founding migrations mechanisms having influenced genomic patterns in admixed populations throughout the TAST. Finally, Cabo Verdean Kriolu is the first creole language of the TAST, born from contacts between the Portuguese language and a variety of African languages (*Quint, 2000*; *Baptista, 2015*; *Baptista, 2003*; *Lang, 2009*). The archipelago thus represents a unique opportunity to investigate, jointly, genetic and linguistic admixture histories and their interactions since the mid-15th century.

Previous genetic studies exploring, first, sex-specific genetic diversity, and, then, genome-wide markers from several islands of the archipelago (*Brehm et al., 2002*; *Gonçalves et al., 2003*; *Beleza et al., 2012*; *Beleza et al., 2013*), attested to the dual, sex-biased, origins of the Cabo Verdean gene-pool, resulting mainly from admixture between African females and European males. Furthermore, these studies described variable admixture patterns between mainland Africa and Europe across islands without distinguishing source populations from different sub-regions within continents. Another, more recent, study investigated which continental mainland European and African populations may have contributed to the Cabo Verde gene-pool without focusing on possible variation across islands (*Micheletti et al., 2020*). Interestingly, adaptive-introgression signals for malaria resistance in Santiago island were recently identified as a result of migrations and genetic admixture during the TAST (*Hamid et al., 2021*). Finally, while joint analyses of genetic and linguistic diversities from the island of Santiago showed that genetic and linguistic admixture histories possibly occurred in parallel (*Verdu et al., 2017*), these previous studies did not attempt to formally reconstruct the admixture processes and detailed demographic histories that influenced the observed patterns of genetic or linguistic diversity on the islands of Cabo Verde.

Based on these previous studies, we propose to first determine which continental African and European populations in fact contributed to the genetic landscape of each Cabo Verdean island today. Indeed, which enslaved-African populations only briefly transited through the archipelago, and which remained for longer periods is largely debated by historians (*Carreira, 2000*; *Albuquerque and Santos, 1991*; *Albuquerque and Santos, 1995*); and, while Portuguese influence is clear, further details about which European migrations genetically influenced Cabo Verde remain to be assessed (*Soares, 2011*). These aspects are often crucial for understanding the genetic history of enslaved-African descendant populations on either side of the Atlantic (*Eltis and Richardson, 2015*; *Baharian et al., 2016*; *Berlin, 2010*; *Gouveia et al., 2020*). Second, we propose to further evaluate the possible parallels between genetic and linguistic admixture histories at a micro-geographical scale within each island. We aim at better understanding how contacts shaped cultural variation during the TAST by deciphering the parent-offspring dispersal behaviors within and across islands which shaped the biological and cultural diversity in the archipelago (*Verdu et al., 2017*). This can be achieved indirectly by exploring the influence of isolation-by-distance mechanisms on the distribution of genetic and linguistic diversity at very reduced geographical scale (~50 km) within a population (*Rousset, 1997*; *Barbujani, 1987*; *Malécot, 1975*; *Barbujani and Sokal, 1991*; *Cavalli-Sforza and Moroni, 2004*; *Verdu et al., 2010*). Finally, we reconstruct the detailed history of admixture dynamics in each island since the 15th century, using statistical inference of possible complex admixture histories with Approximate Bayesian Computation (Fortes-Lima et al., 2021). Altogether, this highlights the socio-historical mechanisms that shaped the genetic and linguistic diversity of the Cabo Verde population, the first to be born from the TAST.

## Results

We investigate genetic and linguistic variation in 233 family unrelated Kriolu speakers from the nine Cabo Verdean islands (Brava, Fogo, Santiago, Maio, Sal, Boa Vista, São Nicolau, São Vicente, Santo Antão, *Figure 1*, *Figure 1—source data 1*). With novel genome-wide genotyping autosomal data (*Appendix 1—figure 1*), we first describe genetic differentiation patterns in Cabo Verde and other enslaved-African descendants in the Americas from previous datasets, in particular with respect to

continental Africa and Europe. Next, we deploy local-ancestry inferences and determine the best proxy source-populations for admixture patterns in each Cabo Verde island. We then describe runs of homozygosity and genetic isolation-by-distance patterns at reduced geographical scale within Cabo Verde. We also investigate Kriolu linguistic diversity with respect to geography and socio-cultural co-variates and, then, investigate jointly genetic and linguistic admixture patterns throughout the archipelago. Finally, we infer the detailed genetic admixture history of each island using the machine-learning *MetHis*-Approximate Bayesian Computation (ABC) approach (Fortes-Lima et al., 2021).

## Cabo Verde and other TAST-related admixed populations in the worldwide genetic context

We explored genetic diversity patterns captured along the first three axes of the multi-dimensional scaling (MDS) projection of individual pairwise allele sharing dissimilarities (ASD *Bowcock et al., 1994*), computed from different individual subsets (*Figure 1—source data 1*). This ASD-MDS approach is mathematically analogous to PCA based on individual genotypes and therefore captures similar information about individual pairwise genetic differentiation (*Hastie et al., 2009*; Chap.-18.5.2). However, ASD-MDS allows to explore pairwise genetic differentiation for successive individual subsets much more efficiently computationally than classical PCA. Indeed, the individual pairwise ASD matrix only needs to be computed once and then simply subsampled before being projected, and successive subset of individual-pairwise ASD matrices are thus always several orders of magnitude smaller in dimensions than the genotype table to be projected with PCA which comprises, here, 455,705 SNPs in all cases. Detailed ASD-MDS decompositions are provided in Appendix 2 and *Appendix 2—figures 1–4*. Note that we considered seven geographical regions in Africa shown in *Figure 1*.

*Figure 2* shows that the second ASD-MDS axis distinguishes West Western African Senegambian populations from East Western Africans, while Central Western Africans are at intermediate distances between these two clusters. Moreover, the third MDS axis separates Northern and Southern European populations; the British-GBR and USA-CEU individuals clustering at intermediate distances between the Finnish-FIN, and a cluster represented by Iberian-IBS and Tuscan-TSI Western Mediterranean individuals. Consistently with previous results (*Micheletti et al., 2020*; *Verdu et al., 2017*), on the first three MDS axes, Cabo Verdean individuals cluster almost exclusively along a trajectory from the Southern European cluster to Senegambia (*Figure 2A–C*), with little traces of affinity with other African or European populations. Instead, the USA African-American ASW (*Figure 2—figure supplement 1 panels A-C*) and Barbadian-ACB (*Figure 2—figure supplement 1 panels D-F*) cluster along a trajectory going from the GBR and CEU cluster to Central and East Western Africa; and Puerto Ricans-PUR cluster along a trajectory going from the Southern European cluster to the Central Western African cluster (*Figure 2—figure supplement 1 panels G-I*).

## Genetic structure in Cabo Verde and other TAST-related admixed populations

Based on these results, we further investigated patterns of individual genetic structure among Cabo Verde-born individuals, ASW, and ACB populations with respect to European and Western, Central, South-Western, and Southern African populations (*Figure 1—source data 1*), using ADMIXTURE (*Alexander et al., 2009*). Indeed, ASD-MDS decompositions allow to efficiently identify major genetic pairwise dissimilarities among numerous samples, but exploring multiple combinations of higher order axes remains extremely difficult with this multivariate method. Instead, ADMIXTURE results recapitulate the major axes of genetic variation with increasing values of the number of clusters *K*, which allows to explore individual pairwise genetic resemblances for numerous major axes of variation at once. Extended descriptions of the results are presented in **Appendix 3**.

At *K*=2, the orange genetic-cluster in *Figure 3A* is maximized in African individuals while the blue alternative cluster is maximized in Europeans. Cabo Verdean, ASW and ACB individuals exhibit intermediate genotype-membership proportions between the two clusters, consistently with patterns expected for European-African admixed individuals. Among the Cabo Verdean, ASW, and ACB populations, ACB individuals show, on average, the highest membership to the orange 'African' cluster (88.23%, SD = 7.33%), followed by the ASW (78.00%, SD = 10.88%), and Cabo Verdeans (59.01%, SD = 11.97%). Membership proportions for this cluster are highly variable across Cabo Verdean islands, with highest average memberships for Santiago-born individuals (71.45%, SD = 10.39%) and

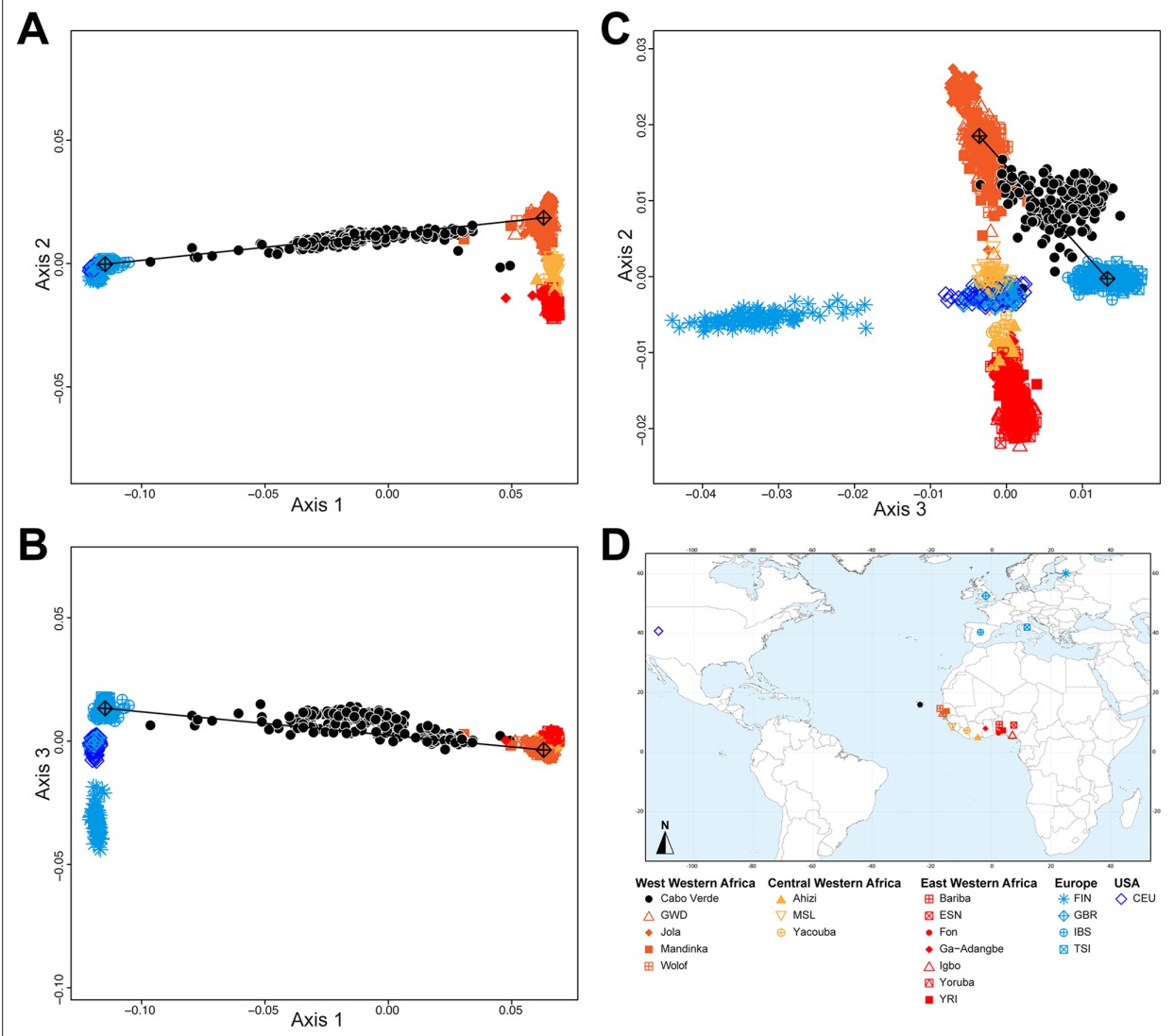

**Figure 2.** Multidimensional scaling projections of pairwise allele sharing dissimilarities in Cabo Verdeans and continental African and European populations. (**A–C**) Three-dimensional MDS projection of ASD computed among 233 unrelated Cabo Verdeans and other continental African and European populations using 445,705 autosomal SNPs. Cabo Verdean patterns in panels **A–C** can be compared to results obtained considering instead the USA African-Americans ASW, the Barbadians-ACB, and the Puerto Ricans-PUR in the same African and European contexts and presented in **Figure 2—figure supplement 1**. We computed the Spearman correlation between the matrix of inter-individual three-dimensional Euclidean distances computed from the first three axes of the MDS projection and the original ASD matrix, to evaluate the precision of the dimensionality reduction. We find significant ($p<2.2 \times 10^{-16}$) Spearman $\rho$=0.9635 for the Cabo Verde analysis (**A–C**). See **Figure 1—source data 1** for the populations used in these analyses. Sample locations and symbols are provided in panel **D**.

The online version of this article includes the following video and figure supplement(s) for figure 2:

**Figure supplement 1.** Multidimensional scaling three-dimensional projection of allele sharing pairwise dissimilarities, for the closest subsets of West African and European populations to the African American ASW, Barbadian ACB, and Puerto Rican PUR populations, separately.

**Figure 2—animation 1.** 3D animated MDS of pairwise allele sharing dissimilarities in Cabo Verdeans and continental African and European populations.

**Figure 2—animation 2.** 3D animated MDS of pairwise allele sharing dissimilarities in Barbadian ACB and continental African and European populations.

**Figure 2—animation 3.** 3D animated MDS of pairwise allele sharing dissimilarities in Puerto Rican PUR and continental African and European populations.

**Figure 2—animation 4.** 3D animated MDS of pairwise allele sharing dissimilarities in Afro American ASW and continental African and European populations.

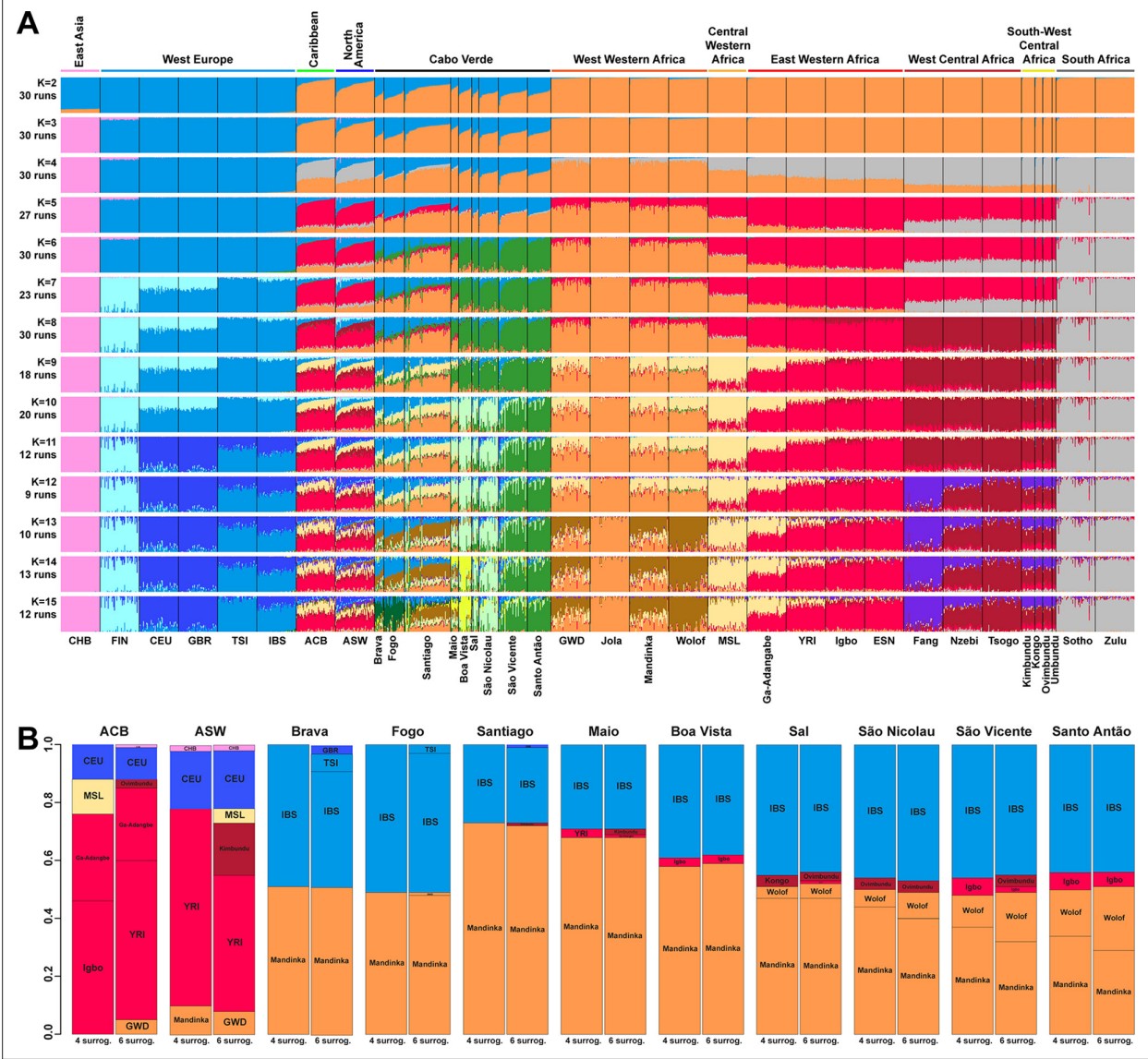

**Figure 3.** Individual genetic structure and haplotypic local ancestry inference among Cabo Verdean, Barbadian-ACB and African-American ASW populations. (**A**) Unsupervised ADMIXTURE analyses using randomly resampled individual sets for populations originally containing more than 50 individuals (*Figure 1—source data 1*). 225 unrelated Cabo Verdean-born individuals in the analysis are grouped by birth island. Numbers of runs highly resembling one another using CLUMPP are indicated below each *K*-value. All other modes are presented in *Appendix 3—figure 1*. (**B**) SOURCEFIND results for each eleven target admixed populations (ASW, ACB, each of the nine Cabo Verde birth islands), considering respectively 4 or 6 possible source surrogate populations (abbreviated 'surrog.') among the 24 possible European, African, and East Asian populations considered in the ADMIXTURE analyses. The cumulated average African admixture levels in each admixed population was highly consistent between SOURCEFIND estimates and ADMIXTURE results at $K=2$ (Spearman $\rho=0.98861$, $p<2 \times 10^{-8}$ and $0.99772$, $p<8 \times 10^{-12}$, for 4 or 6 surrogates, respectively). Furthermore, individual admixture levels estimated using an ASD-MDS-based approach (**Material and Methods** and *Appendix 1—figure 2*), were highly consistent with individual admixture estimates based on ADMIXTURE results at $K=2$ ($\rho=0.99725$; $p<2.2 \times 10^{-16}$ for Cabo Verde; $\rho=0.99568$; $p<2.2 \times 10^{-16}$ for ASW; $\rho=0.99366$; $p<2.2 \times 10^{-16}$ for ACB).

The online version of this article includes the following figure supplement(s) for figure 3:

**Figure supplement 1.** Population $F_{ST}/F_{ST}^{max}$ values for the ASW, ACB, and each Cabo Verdean birth-island separately considering the ADMIXTURE mode result at $K=2$ in *Figure 3A*.

**Figure supplement 2.** $f_3$-admixture tests of admixture for each Cabo Verdean birth-island, the Barbadian-ACB, and the African-American ASW populations related to the TAST.

Maio (70.39%, SD = 5.26%), and lowest for Fogo (48.10%, SD = 6.89%) and Brava (50.61%, SD = 5.80%). Inter-individual membership variation within Cabo Verde islands, captured as $F_{ST}/F_{ST}^{max}$ values (*Morrison et al., 2022*), are significantly different across pairs of islands for 32 out of 36 comparisons (Wilcoxon rank-sum test p<$3.96 \times 10^{-8}$), with variability across islands ranging from a lowest value of 0.010 in individuals from Santo Antão to a highest value of 0.0519 in Santiago (*Figure 3—figure supplement 1*).

At *K*=5, the new red cluster is maximized in the YRI, Igbo, and ESN populations, distinct from Western and Southern African orange and grey clusters, respectively. Note that the former orange cluster is almost completely replaced with red membership in the ACB and ASW populations, while it remains large for all Cabo Verdean-born individuals. Moreover, Cabo Verde-born individuals' patterns of membership proportions to the orange, red or grey 'African' clusters differ here between individuals born on Santiago, Fogo, Brava, and Maio, and individuals born on Sal, Boa Vista, São Nicolau, São Vicente, and Santo Antão, respectively. The former group of islands exhibit an 'African' component resembling patterns of membership proportions found in West Western African individuals, with a majority of membership to the orange cluster and a minority to the red cluster. Instead, the 'African' component in individuals born in the latter islands is almost exclusively orange. This potentially indicates differences in shared ancestries with different continental African populations across islands in Cabo Verde, which remains to be formally tested (see the next Results section).

At *K*=6, these two groups of islands are now clearly differentiated, as the novel green cluster is maximized in numerous individuals born on Sal, Boa Vista, São Nicolau, São Vicente, and Santo Antão, but represented to a much lesser extent in Santiago, Fogo, and Brava. Instead, individuals in these latter islands retain a majority of membership to the orange 'West Western African' cluster, and Maio-born individuals are now found intermediately between the two groups with relatively even memberships to the orange and green clusters respectively. Interestingly, this new green cluster appears to be specific to Cabo Verdean genetic variation, as it is virtually absent from other populations in our dataset except for a small proportion in certain Wolof individuals from West Western Africa.

At *K*=10, the light-green cluster is maximized in Cabo Verdean individuals born on Maio, Boa Vista, Sal, and São Nicolau, distinct form the dark green cluster maximized in individuals born on Santo Antão and São Vicente, and hence producing three distinct ADMIXTURE patterns among Cabo Verdean birth-islands. Furthermore, an alternative mode at *K*=10 shows (*Appendix 3—figure 1*) that Cabo Verde-born individuals resemble more IBS and TSI patterns for their European-like membership than ASW and ACB individuals who, instead, resemble more CEU and GBR patterns, consistently with ASD-MDS results (*Figure 2* and *Figure 2—figure supplement 1*).

While the modal results comprising the most ADMIXTURE runs for increasing values of *K* from 11 to 13 differentiate novel clustering patterns among continental African and/or European populations (*Figure 3*), alternative, minority, modes here highlight novel possible clustering solutions in turn maximized in different groups of Cabo Verdean islands (*Appendix 3—figure 1*). Ultimately, these alternative ADMIXTURE results are resolved at *K*=14 (*Figure 3*), with the emergence of the new bright yellow cluster maximized in individuals from Boa Vista, and in part in individuals from Maio, while virtually absent from the rest of our data set.

Finally, at *K*=15, the novel dark green cluster is maximized in individuals born on Fogo and substantially present in Brava-born individuals' membership proportions, while virtually absent from all other populations in our data set. Note that alternative clustering solutions at *K*=15 disentangle resemblances across other West Central and South-West Central African populations, but do not further propose additional clusters specifically represented by Cabo Verdean variation (*Appendix 3—figure 1*).

Therefore, altogether, we identified at least five clustering patterns across Cabo Verdean islands of births nested in increasing values of *K*, where, respectively, individuals from Fogo and Brava, from Santiago, from Boa Vista, from Sal and São Nicolau, and from Santo Antão and São Vicente resembled more one another than other individuals from elsewhere in Cabo Verde. In this context, note that Maio individuals cluster intermediately between the Santiago, Boa Vista and São Vicente clusters.

Finally, we aim at describing potential genetic resemblances between the East Asian gene-pool, represented here by the Chinese CHB population, and the Cabo Verdean gene-pool, as a community from China is established in the archipelago since at least the 1950s. Note that for every value of *K* above 3, the light-pink cluster mainly represented by Chinese CHB individuals is found in three ASW

and one ACB individuals, as previously identified (*Mathias et al., 2016*), but is virtually absent in the Cabo Verdean individuals that were included in our study without criteria of geographic origins nor community belonging (see also **Appendix 3**).

Altogether, these ADMIXTURE results, differentiating patterns of genetic resemblance across Cabo Verde and with respect to varied continental African and European populations, have been possible to uncover due to inclusion of varied reference populations from continental Atlantic Africa and Europe, treating all Cabo Verdean islands of birth as differentiated in the analyses (*Micheletti et al., 2020*; *Verdu et al., 2017*; *Beleza et al., 2013*).

## Local-ancestry in Cabo Verde and other TAST-related admixed populations

ASD-MDS and ADMIXTURE descriptive analyses do not formally test admixture and putative source populations of origins, they rather disentangle genetic resemblances among groups of individuals. The resulting ADMIXTURE patterns could be due either to admixture from populations represented in our dataset, to admixture from populations un-represented in our dataset, or to common origins and drift (*Alexander et al., 2009*; *Pritchard et al., 2000*; *Rosenberg et al., 2002*; *Falush et al., 2003*; *Lawson et al., 2018*). We further analyzed the observed ADMIXTURE results by computing $f_3$-admixture tests (*Patterson et al., 2012*). We considered as admixture targets each Cabo Verdean birth-island, the ASW, and the ACB separately, with, as admixture sources, in turn all 108 possible pairs of one continental African population and one continental European population, or one continental African population and the East Asian CHB, using the same individuals, population groupings, and genotyping dataset as in the previous ADMIXTURE analyses.

For each Cabo Verdean birth island as a separate target population and for all pairs of possible sources tested, we obtain negative values of $f_3$-admixture (*Figure 3—figure supplement 2*), indicative of possible admixture signals (50). Altogether for the admixture of each Cabo Verdean birth-island, $f_3$-admixture tests do not allow us to clearly discriminate among possible African sources, nor among possible European sources, due to largely overlapping $f_3$-admixture values across tests (*Figure 3— figure supplement 2*). Note that $f_3$ and $f_4$ statistics have been recently shown to be strongly geometrically related to MDS/PCA and that its results need not be due to admixture only (*Peter, 2022*), similarly to MDS/PCA or ADMIXTURE results (*Alexander et al., 2009*; *Pritchard et al., 2000*; *Rosenberg et al., 2002*; *Falush et al., 2003*; *Lawson et al., 2018*). Therefore, we conducted admixture-LD haplotypic local-ancestry inferences with the SHAPEIT2-CHROMOPAINTER-SOURCEFIND pipeline (*Lawson et al., 2012*; *Delaneau et al., 2012*; *Chacón-Duque et al., 2018*), to more precisely identify the possible European and African populations at the sources of genetic patterns observed in enslaved-African descendant populations and Cabo Verdeans in particular.

*Figure 3B* shows striking differences concerning both the European and the African source populations involved in the admixture history of ACB, ASW, and individuals born on different Cabo Verdean islands. We find that individuals from all Cabo Verdean islands share almost all their European haplotypic ancestry with the Iberian-IBS population rather than other European populations. Santiago-born individuals present the smallest (27%) average haplotypic ancestry shared with IBS, and Fogo-born the highest (51%). Conversely, the ASW and ACB both share haplotypic ancestries only with the USA-CEU of North-Western European origin (20% and 12% respectively).

Furthermore, we find that all Cabo Verdeans almost exclusively share African haplotypic ancestries with two Senegambian populations (Mandinka and Wolof) and very reduced to no shared ancestries with other regions of Africa. More specifically, we find that the Mandinka from Senegal are virtually the sole African population to share haplotypic ancestry with Cabo Verdeans born on Brava, Fogo, Santiago, Maio, and Boa Vista, and the majority of the African shared ancestry for Sal, São Nicolau, São Vicente, and Santo Antão. In individuals from these four latter islands, we find shared haplotypic ancestry with the Wolof population ranging from 4%–5% for individuals born on Sal (considering four or six possible sources, respectively), up to 16–22% for Santo Antão. Finally, we find limited (1–6%) shared haplotypic ancestry with East Western (Igbo, YRI, or Ga-Adangbe) or South-West Central (Kimbundu, Kongo, or Ovimbundu) African populations in all Cabo Verdean islands, except Fogo and Brava, and the specific populations identified and their relative proportions of shared haplotypic ancestries vary across analyses. Conversely, we find that the ASW and ACB populations share African haplotypic ancestries in majority with East Western African populations (YRI, Ga-Adangbe, and Igbo),

and substantial shared ancestries with Senegambian populations (5–10%), the MSL from Sierra Leone in Central Western Africa (5–12%), and South-West Central African populations (3–18%), albeit variable depending on the number of putative sources considered.

## Runs of homozygosity (ROH) and admixture patterns within Cabo Verde

Runs of homozygosity (ROH) are Identical-By-Descent haplotypes resulting from recent parental relatedness and present in individuals as long stretches of homozygous genotypes. Their length and abundance can reflect demographic events, such as population bottlenecks and founder events, natural selection, and cultural preferences for endogamy (*Thompson, 2013*; *Mooney et al., 2018*; *Szpiech et al., 2019*); and ROH have not been seen to depend strongly on recombination or mutation rate variation across the genome (*Pemberton et al., 2012*).

We find higher levels of long ROH (≥1 cM) in Cabo Verdeans compared to most other analyzed populations, including ASW and ACB (*Figure 4A* and *Appendix 4—figure 1*). We find the highest levels of long-ROH in individuals born on Maio, Boa Vista, Sal, São Nicolau, São Vincente, and Santo Antão, with a mean individual length of long-ROHs around 3 cM (*Figure 4B*), and the lowest levels of long-ROH in Santiago and Brava-born individuals. Among long ROH (*Appendix 4—figure 2C*), we find little to no correlation with total non-ROH levels for African local-ancestry segments (Pearson $\rho=-0.06689$, $p=0.3179$), European ($\rho=0.1551$, $p=0.01989$), or East Asian ($\rho=0.06239$, $p=0.3516$). Of all ROH identified, the mean proportion of ROH that were long ranged from 0.065 to 0.280 (*Figure 4—source data 1*).

In admixed populations, we expected that some of the long ROH spanned local ancestry breakpoint switches (see **Material and Methods**), indicating that the most recent common ancestor existed after the initial admixture event having generated local-ancestry patterns. Furthermore, we expected that these 'spanning' ROH would be among the longest ROH observed if admixture occurred only in the past few generations. We find that (*Figure 4C*), almost uniformly across Cabo Verde, the longest ROH identified indeed spanned at least one ancestry breakpoint, excluding the very few East Asian ancestry regions identified. Furthermore, correcting ancestry-specific long-ROH sizes (*Figure 4—figure supplement 1*) for individuals' total ancestry fraction of that ancestry, we find that individuals born in Fogo have, on average, an overrepresentation of European ancestry (and a corresponding underrepresentation of African ancestry) in long-ROH (*Figure 4D*; permutation $p<10^{-4}$; *Figure 4—figure supplement 2*, *Figure 4—source data 2*), and that individuals from Santo Antão and São Vincente have, conversely, an apparent overrepresentation of African ancestry and underrepresentation of European ancestry in long ROH (permutation $p=10^{-4}$ and $p<10^{-4}$, respectively; *Figure 4—figure supplement 2*, *Figure 4—source data 2*). Finally, we find that individuals from Brava, Santiago, Maio, Boa Vista, São Nicolau, and Sal have relatively similar long-ROH levels in African and European segments (permutation $p>0.01$; *Figure 4—figure supplement 2*, *Figure 4—source data 2*). This latter pattern may be consistent with these populations being founded by admixed individuals, while the former patterns could indicate, in addition to such admixture founding effects, more recent or recurring contributions from the sources (*Mooney et al., 2018*).

## Genetic and linguistic isolation-by-distance within Cabo Verde

The above ASD-MDS, ADMIXTURE, local-ancestry inferences, and ROH results suggest substantial genetic differentiation at a very reduced geographical scale within the archipelago across Cabo Verdean birth-islands of individuals. Following previous linguistic investigations highlighting Kriolu qualitative linguistic variation across islands within the archipelago (*Quint, 2000*; *Baptista, 2015*; *Baptista, 2003*; *Lang, 2009*), we aim at further characterizing possible patterns of joint genetic and linguistic isolation (*Rousset, 1997*) at very reduced geographical scale across islands as well as within islands. Indeed, while the geographic distribution of genetic diversity has been previously extensively explored across human populations to reveal population migration routes, in particular in island and archipelago contexts (e.g. *Arauna et al., 2022*; *Hunley et al., 2008*), the underlying parent-offspring genetic and linguistic dispersal mechanisms have been seldom explored in humans at extremely local scales to our knowledge (*Rousset, 1997*; *Barbujani and Sokal, 1991*; *Cavalli-Sforza and Moroni, 2004*; *Verdu et al., 2010*). Nevertheless, knowledge about such dispersal mechanisms can be built by exploring the influence of isolation-by-distance mechanisms on genetic and linguistic diversity

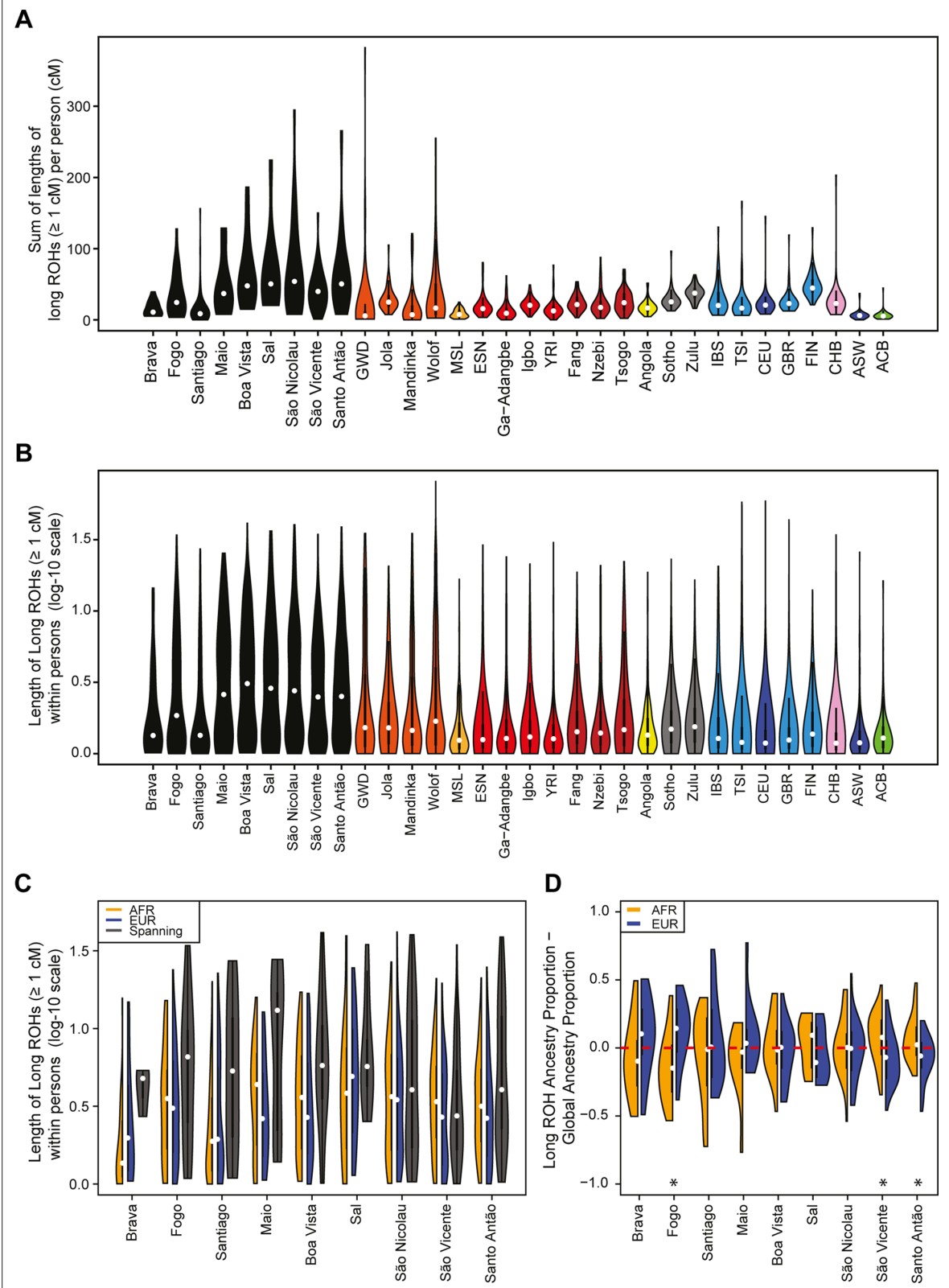

**Figure 4.** Distributions of long ROHs (≥ 1 cM) in Cabo Verde. (**A**) The distribution of the sum of long-ROH (≥1 cM) lengths per person for each Cabo Verdean birth-island and other populations. (**B**) The length distribution (log-10 scale) of individual long-ROHs identified within samples for each Cabo Verdean birth-island and other populations (e.g. for a distribution with mass at 1.0, this suggests individual ROHs of length 10 cM were identified among samples from that group). (**C**) The length distribution of ancestry-specific and ancestry-spanning individual long-ROHs for each Cabo Verdean

*Figure 4 continued on next page*

*Figure 4 continued*

birth-island. (**D**) The distribution of differences between individuals' long-ROH ancestry proportion and their global ancestry proportion, for African and European ancestries separately and for each Cabo Verdean birth-island. * indicates significantly ($a < 1\%$) different proportions of ancestry-specific long-ROH, based on non-parametric permutation tests, see Material and Methods, *Figure 4—source data 2*, and *Figure 4—figure supplement 2*.

The online version of this article includes the following source data and figure supplement(s) for figure 4:

**Source data 1.** Mean proportion of total length of ROH that are classified as long (cM ≥1) for each Cabo Verdean island of birth.

**Source data 2.** Permutation tests' p-values for over/under representation of ancestry in long ROH (cM ≥1) for each Cabo Verdean island of birth.

**Source data 3.** Mean proportion of total length of long ROH (cM ≥1) that have heterozygous ancestry (AFR and EUR), for each Cabo Verdean island of birth.

**Figure supplement 1.** The distribution of total ancestry in long ROH per individual for each Cabo Verdean birth-island.

**Figure supplement 2.** Permutation distributions for over/under representation of ancestry in long ROH (≥1 cM) for each Cabo Verdean island of birth.

distributions at very reduced geographical scales (~50 km) within a population (*Rousset, 1997*; *Barbujani, 1987*; *Malécot, 1975*). We thus explored both pairwise ASD and inter-individual variation in manners of speaking Kriolu (characterized as differences in the frequencies of use of Kriolu utterances among individual discourses; see **Material and Methods**), in the same set of 225 Cabo Verde-born individuals. To do so, we used MDS and Mantel testing of correlations between, respectively, genetic and linguistic pairwise differentiation, and socio-cultural and geographical covariates including age, duration of academic education, residence locations, birth-places, and parental birth-places (*Figure 5*, *Table 1*, and *Table 1—source data 1*).

The first ASD-MDS axis differentiates mainly individuals born on Brava and Fogo compared to Santiago (*Figure 5A–B*). The second axis mainly differentiates individuals from Santiago, Fogo, and Brava from all other islands, while the third axis differentiates individuals from Boa Vista, São Nicolau, Sal, and Maio from all other birth-islands. Furthermore, we find a significant positive correlation between ASD and actual individual birth-locations across Cabo Verde (*Table 1*; Spearman $\rho$=0.2916, two-sided Mantel $p<2 \times 10^{-4}$). This correlation increases when considering only within-islands pairwise comparisons and excluding all inter-island comparisons (Spearman $\rho$=0.3460, two-sided Mantel $p<2 \times 10^{-4}$), thus illustrating the strong signal of genetic isolation-by-distance (*Rousset, 1997*) within Cabo Verde at very reduced geographical scales.

Furthermore, the first utterance-MDS axis of pairwise inter-individual Euclidean distances between utterance frequencies mainly differentiates Santiago and Santo Antão/São Vicente-born individuals' speech-varieties; all other Cabo Verdeans cluster intermediately (*Figure 5C–D*). The third axis further separates speech-varieties recorded in individuals from Fogo, Maio, and Brava. Analogously to genetic differentiation patterns, we find a positive correlation between differences in utterance frequencies and actual birth-places' distances (Spearman $\rho$=0.2794, two-sided Mantel $p<2 \times 10^{-4}$), as well as paternal and maternal birth-places respectively (*Table 1*). However, unlike for to ASD, we find that utterance-frequencies differences stem from inter-birth-islands' distances, rather than shorter distances within islands only. Extending previous results from Santiago only (*Verdu et al., 2017*), these results altogether show that speech-varieties are significantly transmitted from one generation to the next throughout Cabo Verde, anchored in individuals' birth-places. Importantly, note, however, that this vertical transmission of manners of speaking Kriolu does not account for the majority of observed linguistic variation across individuals in our dataset. Indeed, we find that age-differences also substantially correlate with utterance-frequency differences even when correcting for individual birth-places (Spearman $\rho$=0.2294, two-sided partial-Mantel $p<2 \times 10^{-4}$). Finally, while we might intuitively expect that academic education influences idiolects, we find instead that differences in education-duration do not correlate with Kriolu utterance-frequencies differences, whether correcting for residence or birth-places distances, or not (*Table 1*). This shows the modest influence of academic education on Kriolu variation. Altogether, our results highlight strong genetic and linguistic isolation-by-distance patterns at reduced geographic distances within Cabo Verde.

Altogether, we find here genetic and linguistic isolation-by-distance anchored in inter-individual birth-places distances across, and sometimes within, Cabo Verdean islands. These results demonstrate the reduced dispersal of Cabo Verdeans at very local scales within the archipelago, both genetically and linguistically, a fundamental mobility-behavior mechanism likely explaining genetic and linguistic

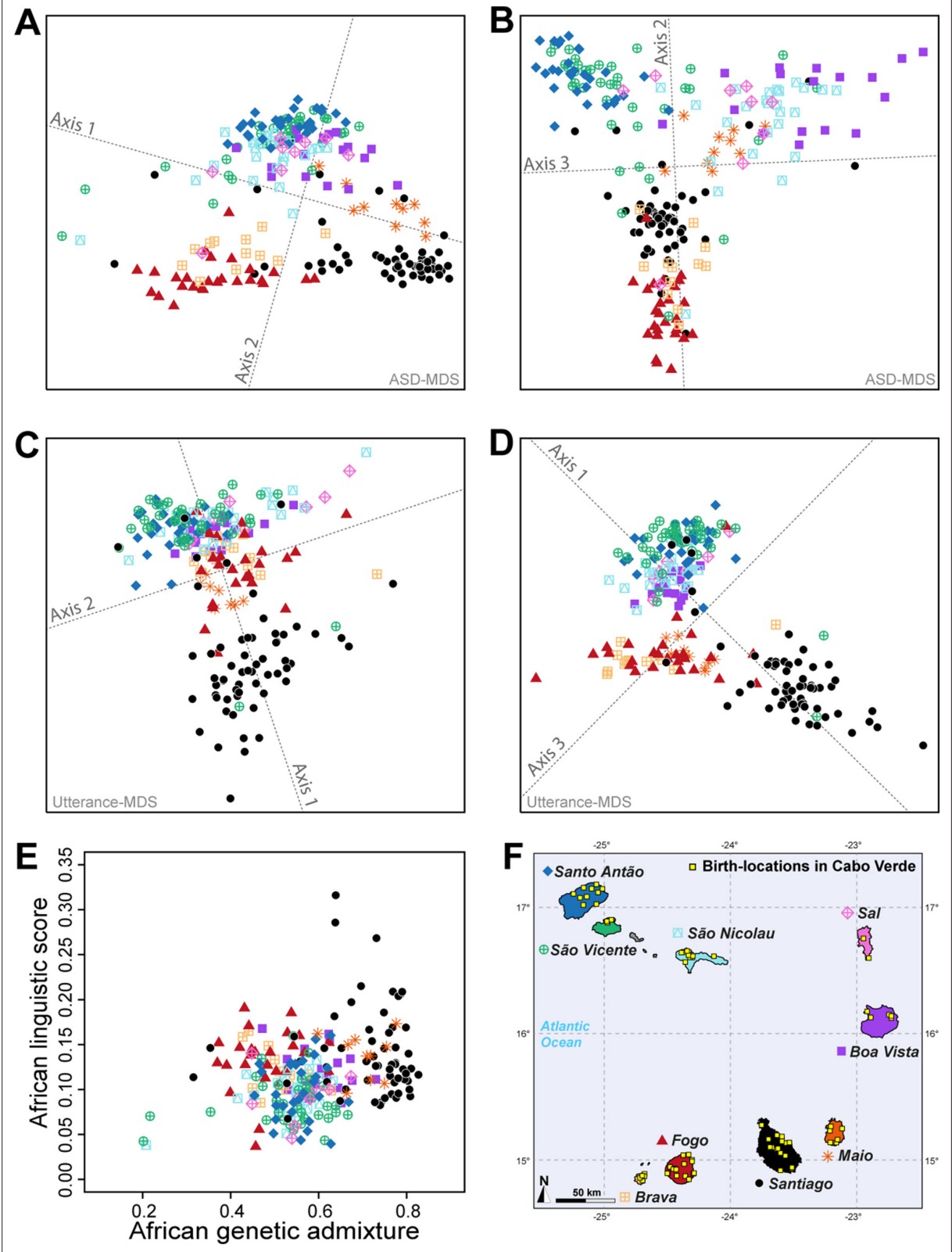

**Figure 5.** Utterance and genetic diversity and admixture within Cabo Verde. (**A–B**) 3D MDS projection of Allele Sharing Dissimilarities computed among 225 unrelated Cabo-Verde-born individuals using 1,899,878 autosomal SNPs. Three-dimensional Euclidean distances between pairs of individuals in this MDS significantly correlated with ASD (Spearman $\rho$=0.6863; $p<2.2 \times 10^{-16}$). (**C–D**) 3D MDS projection of individual pairwise Euclidean distances between uttered linguistic items frequencies based on the 4831 unique uttered items obtained from semi-spontaneous discourses. Three-dimensional Euclidean

*Figure 5 continued on next page*

*Figure 5 continued*

distances between pairs of individuals in this MDS significantly correlated with the utterance-frequencies distances (Spearman *ρ*=0.8647; *p*<2.2 × 10$^{-16}$). (**E**) Spearman correlation *ρ*=0.2070 (*p*=0.0018) between individual African utterance scores and individual genetic African admixture rates obtained with ADMIXTURE at *K*=2. (**F**) Birth-locations of 225 individuals in Cabo Verde. Symbols for individuals' birth-island in panels **A–E** are shown in panel **F**. Panel **A–D** were Procrustes-transformed according to individual actual birth-places' geographical locations in panel **F** (*Wang et al., 2010*).

isolation across islands and sometimes even within islands despite the large self-reported exploration mobility of Cabo Verdeans.

## Geographic distribution of genetic and linguistic admixture within Cabo Verde

Based on these results of genetic and linguistic diversity isolation-by-distance patterns anchored in individual's birth-places, we aim at investigating whether individual genetic and/or linguistic admixture levels also exhibit isolation-by-distance patterns across and within islands, beyond the qualitative observation that genetic and linguistic admixture patterns vary across different islands of Cabo Verde obtained above and in previous results (*Verdu et al., 2017*; *Beleza et al., 2013*). Interestingly, we find that absolute differences in inter-individual genetic admixture levels from Africa, estimated with ADMIXTURE or ASD-MDS, significantly correlate with actual birth-places distance across islands (Spearman *ρ*=0.1865, two-sided Mantel *p*<2 × 10$^{-4}$ and *ρ*=0.1813, *p*<2 × 10$^{-4}$, respectively), but not within-islands only (*ρ*=0.0342 *p*=0.3094 and *ρ*=0.0282 *p*=0.3385, respectively). This shows that two individuals born on far-away islands are likely to differ more in African genetic admixture levels, than two individuals born on close-by islands, a form of isolation-by-distance pattern for genetic admixture across Cabo Verdean islands.

We explored inter-individual variation in Kriolu utterance frequencies specifically for uttered items of clearly African and dual European-African origins (utterance categories A and B; see **Material and Methods**) providing an estimate of individual African linguistic-admixture scores (*Verdu et al., 2017*). We find that African linguistic-admixture score differences significantly correlate with actual birth-places' distances throughout Cabo Verde (Spearman *ρ*=0.1297, two-sided Mantel *p*<2 × 10$^{-4}$), and even marginally significantly correlate with birth-places' distances at short distances within birth-islands (Spearman *ρ*=0.1209, two-sided Mantel *p*=0.0419).

Finally, we find a significant positive correlation (Spearman *ρ*=0.2070, *p*=0.0018) between genetic and linguistic admixture in Cabo Verde (*Figure 5E*), indicating that individuals who frequently use African-related utterances in their manner of speaking Kriolu are more likely to exhibit higher levels of African genetic-admixture. This correlation remains, respectively, marginally significant and significant when considering utterances of strictly African-origin (Category A) or utterances with a dual European-African etymology (Category B) separately (Spearman *ρ*=0.1631, *p*=0.0143, and *ρ*=0.1829, *p*=0.0059, respectively). These positive correlations between genetic and linguistic admixture generalize to the whole archipelago our previous results obtained in Santiago only (*Verdu et al., 2017*), and further suggest that genetic and linguistic admixture histories may have occurred in parallel all throughout Cabo Verde.

Therefore, not only we identify isolation-by-distance patterns within Cabo Verdean islands for genetic and linguistic diversities, but also identify a form of isolation-by-distance for genetic and linguistic admixture levels at very reduced geographical scales. This suggests that processes of reduced dispersal of individuals can also be identified in the genetic and linguistic admixture patterns, which has never been previously observed in human admixed populations to our knowledge, nor previously suspected whether genetically or linguistically in Cabo Verde (*Micheletti et al., 2020*; *Quint, 2000*; *Baptista, 2015*; *Verdu et al., 2017*; *Lang, 2009*; *Beleza et al., 2012*).

Together with the above LAI and ROH results, the various isolation-by-distance patterns here identified suggest that different founding events followed by local isolation due to reduced genetic and linguistic dispersal ranges, as well as different admixture histories, are at the root of patterns of genetic and linguistic diversity and admixture throughout Cabo Verde, anchored in individual birth places across islands, and even sometimes within islands.

**Table 1.** Mantel and partial-Mantel correlations between utterance frequency differences and covariables, and between genetic ASD and the same covariables, in 225 genetically unrelated Cabo Verde-born Kriolu-speaking individuals.

Spearman correlations $\rho$ are indicated in bold when significant at $\alpha<0.001$, and in italics otherwise. Spearman correlations and Mantel-tests among covariables are provided in *Table 1—source data 1*.

| Mantel variable | Partial-Mantel control | n | Geographic scale | Genetic ASD - 1,899,978 SNPs | | Utterance-frequency Euclidean distances - 4831 uttered items | |
|---|---|---|---|---|---|---|---|
| | | | | Spearman rho | 10,000 Mantel two-sided permutation p | Spearman rho | 10,000 Mantel two-sided permutation p |
| abs(Age difference) | -- | 225 | within and between islands | 0.1303 | <2.10⁻⁴ | 0.2215 | <2.10⁻⁴ |
| abs(Age difference) | log(Birth-loc. dist.) | 225 | within and between islands | 0.1348 | <2.10⁻⁴ | 0.2294 | <2.10⁻⁴ |
| log(Birth-loc. dist.) | -- | 225 | within and between islands | 0.2916 | <2.10⁻⁴ | 0.2794 | <2.10⁻⁴ |
| log(Birth-loc. dist.) | abs(Age difference) | 225 | within and between islands | 0.2935 | <2.10⁻⁴ | 0.2855 | <2.10⁻⁴ |
| abs(Education duration difference) | -- | 186 | within and between islands | *0.0168* | *0.2730* | *0.0962* | *0.0024* |
| abs(Education duration difference) | log(Birth-loc. dist.) | 186 | within and between islands | *−0.0023* | *0.4900* | *0.0834* | *0.0071* |
| abs(Education duration difference) | -- | 185 | within and between islands | *0.0159* | *0.2825* | *0.1001* | *0.0014* |
| abs(Education duration difference) | log(Residence dist.) | 185 | within and between islands | *−0.0041* | *0.4651* | *0.0824* | *0.0068* |
| log(Residence dist.) | -- | 224 | within and between islands | 0.1658 | <2.10⁻⁴ | 0.2145 | <2.10⁻⁴ |
| log(Residence dist.) | log(Birth-loc. dist.) | 224 | within and between islands | *−0.0488* | *0.0005* | *0.0306* | *0.0682* |
| log(Birth-loc. dist.) | -- | 224 | within and between islands | 0.2889 | <2.10⁻⁴ | 0.2800 | <2.10⁻⁴ |
| log(Birth-loc. dist.) | log(Residence dist.) | 224 | within and between islands | 0.2445 | <2.10⁻⁴ | 0.1863 | <2.10⁻⁴ |
| log(Father Birth-loc. dist.) | -- | 222 | within and between islands | 0.2424 | <2.10⁻⁴ | 0.1704 | <2.10⁻⁴ |
| log(Father Birth-loc. dist.) | log(Birth-loc. dist.) | 222 | within and between islands | *0.0846* | *0.0014* | *0.0066* | *0.3915* |
| log(Mother Birth-loc. dist.) | -- | 224 | within and between islands | 0.2619 | <2.10⁻⁴ | 0.2634 | <2.10⁻⁴ |

*Table 1 continued on next page*

*Table 1 continued*

| Mantel variable | Partial-Mantel control | n | Geographic scale | Genetic ASD - 1,899,978 SNPs | | Utterance-frequency Euclidean distances - 4831 uttered items | |
|---|---|---|---|---|---|---|---|
| | | | | Spearman rho | 10,000 Mantel two-sided permutation p | Spearman rho | 10,000 Mantel two-sided permutation p |
| log(Mother Birth-loc. dist.) | log(Birth-loc. dist.) | 224 | within and between islands | *0.0748* | *0.0057* | *0.0853* | *0.0071* |
| abs(Age difference) | -- | 225 | within islands only | **0.2124** | **0.0006** | **0.2727** | **<2.10⁻⁴** |
| abs(Age difference) | log(Birth-loc. dist.) | 225 | within islands only | *0.1648* | *0.0041* | **0.2546** | **<2.10⁻⁴** |
| log(Birth-loc. dist.) | -- | 225 | within islands only | **0.3460** | **<2.10⁻⁴** | *0.1412* | *0.0401* |
| log(Birth-loc. dist.) | abs(Age difference) | 225 | within islands only | **0.3212** | **<2.10⁻⁴** | *0.0990* | *0.1030* |
| abs(Education duration difference) | -- | 186 | within islands only | *–0.0370* | *0.3077* | *0.1287* | *0.0440* |
| abs(Education duration difference) | log(Birth-loc. dist.) | 186 | within islands only | *–0.0537* | *0.2330* | *0.1239* | *0.0496* |
| abs(Education duration difference) | -- | 185 | within islands only | *–0.0382* | *0.3037* | *0.1421* | *0.0292* |
| abs(Education duration difference) | log(Residence dist.) | 185 | within islands only | *–0.0491* | *0.2566* | *0.1202* | *0.0546* |
| log(Residence dist.) | -- | 224 | within islands only | *–0.0667* | *0.1907* | *0.0982* | *0.0911* |
| log(Residence dist.) | log(Birth-loc. dist.) | 224 | within islands only | *–0.0549* | *0.2319* | *0.1063* | *0.0704* |
| log(Birth-loc. dist.) | -- | 224 | within islands only | **0.3465** | **<2.10⁻⁴** | *0.1537* | *0.0282* |
| log(Birth-loc. dist.) | log(Residence dist.) | 224 | within islands only | **0.3446** | **<2.10⁻⁴** | *0.1589* | *0.0230* |
| log(Father Birth-loc. dist.) | -- | 222 | within islands only | **0.2660** | **0.0006** | *0.0160* | *0.4123* |
| log(Father Birth-loc. dist.) | log(Birth-loc. dist.) | 222 | within islands only | *0.2187* | *0.0045* | *–0.0111* | *0.4546* |
| log(Mother Birth-loc. dist.) | -- | 224 | within islands only | *0.2240* | *0.0034* | *0.1283* | *0.0423* |
| log(Mother Birth-loc. dist.) | log(Birth-loc. dist.) | 224 | within islands only | *0.1563* | *0.0303* | *0.1000* | *0.0925* |

The online version of this article includes the following source data for table 1:

**Source data 1.** Mantel correlations among individual birth-places, residence-places, maternal and paternal birth places, age, and academic education duration.

## Genetic admixture histories in Cabo Verde inferred with *MetHis*-ABC

Highly complex admixture histories, with more than two separate pulses and/or periods of recurring admixture from each source population, are often impossible to infer from observed genetic data using maximum-likelihood approaches; whether the likelihood itself cannot be explicitly formulated

or whether its maximization is computationally intractable for such high levels of complexity (**Gravel, 2012**; **Hellenthal et al., 2014**; **Foll et al., 2015**; **Ni et al., 2019**). Instead, Approximate Bayesian Computation allows, in principle, formal comparison of competing scenarios underlying the observed data and estimation of the posterior distribution of the parameters under the winning model (**Tavaré et al., 1997**; **Beaumont et al., 2002**). The user simulates numerous genetic datasets under competing scenarios, drawing randomly the parameter values of each simulation in prior distributions. ABC then allows to formally compare a set of summary statistics calculated on the observed data with the same set of summary statistics calculated on each simulated genetic dataset separately, in order to identify which of the competing scenarios produces simulations for which summary-statistics are closest to the observed ones. Under the winning scenario, ABC then estimates the joint posterior distribution of parameter values which produced simulations whose summary statistics most resemble the observed ones. Therefore, ABC allows, in principle, to infer arbitrarily complex demographic models underlying the data, provided that data can be efficiently simulated under these scenarios drawing randomly parameter values from prior distributions explicitly set by the user, and provided that calculated summary statistics are indeed informative about the scenarios' parameters (**Sisson et al., 2018**).

We reconstruct the admixture histories of each Cabo Verde island separately using the *MetHis*-ABC framework (Fortes-Lima et al., 2021; **Pudlo et al., 2016**; **Csilléry et al., 2012**). It was recently developed to investigate highly complex admixture histories using machine-learning ABC, by simulating

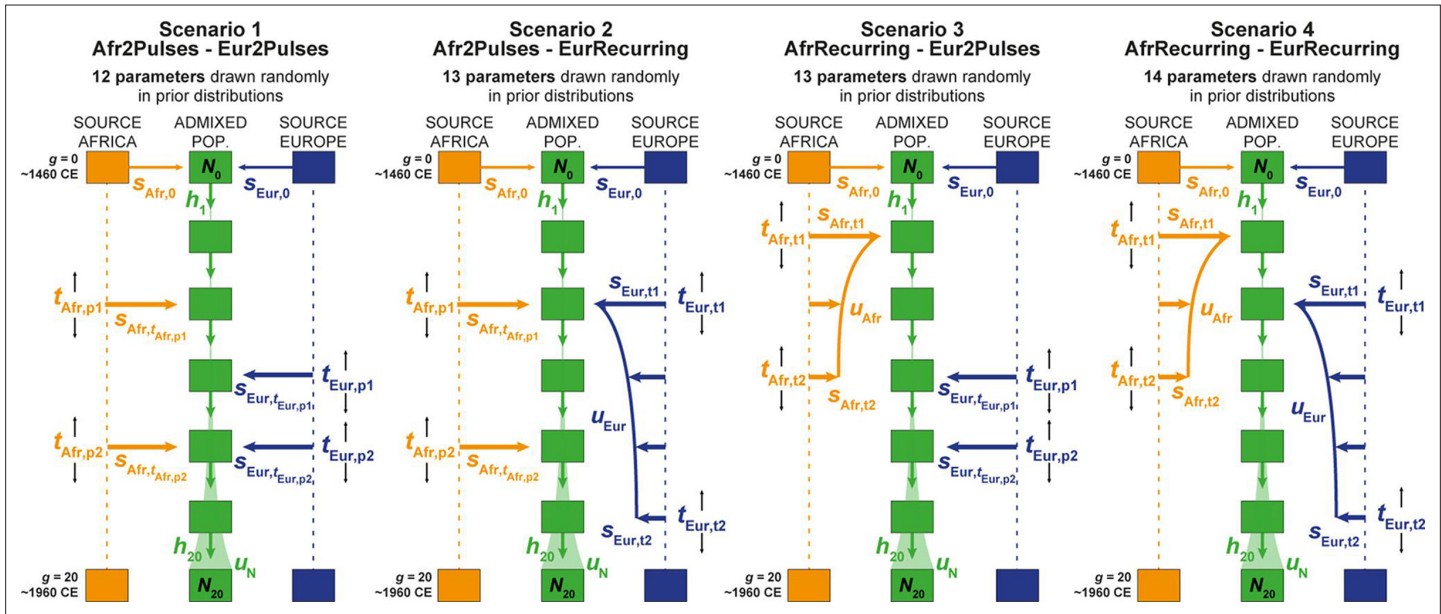

**Figure 6.** Four competing scenarios for the genetic admixture histories of each Cabo Verde island. For all scenarios, the duration of the admixture process is set to 20 generations after the initial founding admixture event occurring at generation 0, which corresponds roughly to the initial peopling of Cabo Verde in the 1460s, considering 25 years per generation and sampled individuals born on average between the 1960s and 1980s. *Scenario 1 Afr2Pulses-Eur2Pulses*: after the initial founding pulse of admixture, the admixed population receives two separate introgression pulses from the African and European sources, respectively. *Scenario 2 Afr2Pulses-EurRecurring*: after the initial founding pulse of admixture, the admixed population receives two separate introgression pulses from the African source, and a period of monotonically constant or decreasing recurring introgression from the European source. *Scenario 3 AfrRecurring-Eur2Pulses*: after the initial founding pulse of admixture, the admixed population receives a period of monotonically constant or decreasing recurring introgression from the African source, and two separate introgression pulses from the European source. *Scenario 4 AfrRecurring-EurRecurring*: after the initial founding pulse of admixture, the admixed population receives a period of monotonically constant or decreasing recurring introgression from the African source, and, separately, a period of monotonically constant or decreasing recurring introgression from the European source. For all scenarios, we consider demographic models corresponding to either a constant reproductive population size $N_g$ between the founding event and the present, or, instead, a linear or hyperbolic increase between $N_0$ and $N_{20}$, depending on the values of $N_0$, $N_{20}$, and $u_N$ used for each simulation respectively. Time for admixture pulses or time for the onset and offset of admixture periods are schematically represented as $t_{Source,g}$. We define (**Verdu and Rosenberg, 2011**), $s_{Afr,g}$, $s_{Eur,g}$, and $h_g$ as the proportion of parents of individuals in the admixed population at generation $g$ coming from, respectively, the African source population, the European one, and the admixed population itself at the previous generation. Thus, for $g=0$, $s_{Afr,0} + s_{Eur,0} = 1$, and for each value of $g$ in [1,20], $s_{Afr,g} + s_{Eur,g} + h_g = 1$. The number of 'free' scenario-parameters drawn randomly in prior distributions set by the user for simulations and subsequent Approximate Bayesian Computation inferences is indicated below the name of each scenario respectively. See *Table 2* for parameter prior distributions, and **Material and Methods** for detailed descriptions of scenario-parameters.

independent autosomal SNPs forward-in-time in an admixed population under any two source-population versions of a general admixture model (*Verdu and Rosenberg, 2011*), and calculating, for each simulation, sets of summary statistics shown to be informative about the underlying admixture models' parameters for ABC inferences (Fortes-Lima et al., 2021). See **Material and Methods** and **Appendix 1** for the detailed description of simulations and ABC machine-learning scenario-choice and posterior parameter estimation procedures.

**Table 2.** Prior distributions for the parameters of four competing scenarios for the admixture history of Cabo Verde islands.

Parameters are presented in *Figure 6* and described in Material and Methods.

| Description | Scenario | Model parameter | Prior | Conditions |
|---|---|---|---|---|
| African admixture-pulse times | 1, 2 | $t_{Afr,p1}$ <br> $t_{Afr,p2}$ | Uniform [1 , 20] in discrete generations, a range corresponding to between ~1485 and ~1960 in years CE | $t_{Afr,p1} > t_{Afr,p2}$ |
| European admixture-pulse times | 1, 3 | $t_{Eur,p1}$ <br> $t_{Eur,p2}$ | Uniform [1 , 20] in discrete generations, a range corresponding to between ~1485 and ~1960 in years CE | $t_{Eur,p1} > t_{Eur,p2}$ |
| African admixture period start and end times | 2, 4 | $t_{Afr,t1}$ <br> $t_{Afr,t2}$ | Uniform [1 , 20] in discrete generations, a range corresponding to between ~1485 and ~1960 in years CE | $t_{Afr,t1} > t_{Afr,t2}$ |
| European admixture period start and end times | 3, 4 | $t_{Eur,t1}$ <br> $t_{Eur,t2}$ | Uniform [1 , 20] in discrete generations, a range corresponding to between ~1485 and ~1960 in years CE | $t_{Eur,t1} > t_{Eur,t2}$ |
| African admixture-pulse intensities | 1, 2 | $s_{Afr,tAfr,p1}$ <br> $s_{Afr,tAfr,p2}$ | Uniform [0, 1] | |
| European admixture-pulse intensities | 1, 3 | $s_{Eur,tEur,p1}$ <br> $s_{Eur,tEur,p2}$ | Uniform [0, 1] | $s_{Afr,g} + s_{Eur,g} = 1 - h_g$, with $h_g$ in [0,1] |
| African admixture period intensity parameters | 2, 4 | $s_{Afr,tAfr,t1}$ | Uniform [0, 1] | $s_{Afr,tAfr,t1} \geq s_{Afr,tAfr,t2}$ |
| | | $s_{Afr,tAfr,t2}$ | Uniform [0, 1] | $s_{Afr,g} + s_{Eur,g} = 1 - h_g$, with $h_g$ in [0,1] |
| | | $u_{Afr}$ | Uniform [0, 0.5] | |
| European admixture period intensity parameters | 3, 4 | $s_{Eur,tEur,t1}$ | Uniform [0, 1] | $s_{Eur,tEur,t1} \geq s_{Eur,tEur,t2}$ |
| | | $s_{Eur,tEur,t2}$ | Uniform [0, 1] | $s_{Afr,g} + s_{Eur,g} = 1 - h_g$, with $h_g$ in [0,1] |
| | | $u_{Eur}$ | Uniform [0, 0.5] | |
| Admixture pulse at the foundation | 1, 2, 3, 4 | $s_{Afr,0}$ | Uniform [0, 1] | $s_{Eur,0} = 1 - s_{Afr,0}$ |
| Founding reproductive population size | 1, 2, 3, 4 | $N_0$ | Uniform [10, 1000] | $N_0 \leq N_{20}$ |
| Current reproductive population size | 1, 2, 3, 4 | $N_{20}$ | Uniform [100, 100,000] | |
| Steepness of the reproductive population size increase | 1, 2, 3, 4 | $u_N$ | Uniform [0, 0.5] | |

## *MetHis*–ABC prior checking

We considered four competing genetic-admixture scenarios described in *Figure 6* and *Table 2*, tested separately for individuals born on each Cabo Verdean island and for the 225 Cabo Verde-born unrelated individuals grouped altogether, with *MetHis*–ABC machine-learning scenario-choice and posterior parameter inferences (Fortes-Lima et al., 2021; *Pudlo et al., 2016*; *Csilléry et al., 2012*). ABC inferences are based on 42 summary statistics (*Table 3*), calculated for each simulation under each competing scenario separately using 60,000 independent autosomal SNPs in Cabo Verdean individuals, the African Mandinka and the European Iberian-IBS proxy source populations.

Note that we did not explicitly simulate genotype data in the European and African source-populations. Instead, we built gamete reservoirs at each 21 generation of the forward-in-time admixture process, matching in frequency the observed allele frequencies at the 60,000 independent SNPs for the Iberian IBS and Mandinka populations, respectively. As in our previous *MetHis*-ABC investigation of the admixture history of the African-American ASW and Barbadian ACB populations (Fortes-Lima et al., 2021), we therefore consider that the African and European proxy populations at the source of the admixture history of Cabo Verde are large and unaffected by mutation during the 21 generations of the admixture process; this assumption is reasonable provided that we consider only independent genotyped SNPs and the very recent demographic history of the archipelago, discovered un-inhabited and first settled in the 1460s. Therefore, although we cannot reconstruct the evolutionary history of the African and European source populations with our design, we nevertheless implicitly take the real demographic histories of these source populations into account in our simulations, as we use observed genetic patterns themselves, the product of this demographic history, to create the virtual source populations at the root of the admixture history of Cabo Verde.

We find that the summary-statistics calculated from the observed datasets fall well within the space of summary-statistics obtained from 10,000 simulated-datasets under each of the four competing scenarios (*Appendix 1—figure 3*, *Appendix 1—figure 3—figure supplements 1–10*), considering non-significant ($\alpha > 5\%$) goodness-of-fit, visual inspection of summary-statistics PCA-projections, and each summary-statistic's distribution, for each Cabo Verdean birth-island and for all Cabo Verde-born individuals grouped in a single population, separately. Prior-checks thus demonstrate that *MetHis* simulations are appropriate for further ABC scenario-choice and posterior parameter inferences using observed data in the African Mandinka and the European Iberian IBS source populations and each Cabo Verde islands separately or grouped altogether, as they allow to mimic the observed summary-statistics, despite the assumption that the European and African proxy source populations are at the drift-mutation equilibrium over the last 21 generations.

## *MetHis*–Random Forest (RF)-ABC scenario-choices

Overall (*Figure 7B*), *MetHis*-RF-ABC scenario-choices indicate that multiple pulses of admixture from the European and African source populations (after the founding admixture pulse, two independent admixture pulses from both Africa and Europe: 'Afr2Pulses-Eur2Pulses' scenarios, *Figure 6* – **Scenario 1**), best explain the genetic history of individuals born on six of nine Cabo Verdean islands. Furthermore, we find that even more complex scenarios involving a period of recurring admixture from either source best explain the history of the remaining three islands. Scenarios with periods of recurring admixture from both Africa and Europe ('AfrRecurring-EurRecurring', *Figure 6* – **Scenario 4**) are the least favored across Cabo Verde.

RF-ABC cross-validation prior-errors for each of the 40,000 simulations used, in-turn, as pseudo-observed data indicate a reasonably good, albeit not perfect, discriminatory power of the RF (*Appendix 1—figure 4A*). RF-ABC scenario-choices identify the correct scenario in the majority of cross-validations for most scenarios and most islands. Furthermore, asymmetrical scenarios are the least confused with one-another (AfrRecurring-Eur2Pulses vs Afr2Pulses-EurRecurring, or Afr2Pulses-Eur2Pulses vs AfrRecurring-EurRecurring). As expected and previously shown empirically with *MetHis*-RF-ABC scenario-choice (Fortes-Lima et al., 2021; *Robert et al., 2010*), these results are consistent with increased assignation-errors in the parts of the parameter-space where the different scenarios are highly nested and thus biologically equivalent. Finally (*Appendix 1—figure 4B*), the mean, variance, skewness, kurtosis, minimum, and maximum of individual admixture proportions' distributions are systematically among the most informative summary-statistics for RF-ABC scenario-choice in every

**Table 3.** Summary-statistics used for *MetHis*-machine-learning ABC inferences.

All 42 statistics were computed using the summary-statistics computation tool embedded in *MetHis* (Fortes-Lima et al., 2021).

| Summary Statistics for ABC inference | | Nunber of statistics | Reference |
|---|---|---|---|
| within population | Mean ASD within population H | 1 | ***Bowcock et al., 1994*** |
| | Mean Heterozygosity (SNP by SNP) within population H | 1 | ***Nei, 1978*** |
| | Variance Heterozygosity (SNP by SNP) within population H | 1 | ***Nei, 1978*** |
| | Mean inbreeding F within population H | 1 | ***Danecek et al., 2011*** |
| | Variance inbreeding F within population H | 1 | ***Danecek et al., 2011*** |
| admixture pattern | Mode ASD-MDS African admixture proportions in population H | 1 | Fortes-Lima et al., 2021; ***Verdu and Rosenberg, 2011*** |
| | Mean ASD-MDS African admixture proportions in population H | 1 | Fortes-Lima et al., 2021; ***Verdu and Rosenberg, 2011*** |
| | Variance ASD-MDS African admixture proportions in population H | 1 | Fortes-Lima et al., 2021; ***Verdu and Rosenberg, 2011*** |
| | Skewness ASD-MDS African admixture proportions in population H | 1 | Fortes-Lima et al., 2021; ***Verdu and Rosenberg, 2011*** |
| | Kurtosis ASD-MDS African admixture proportions in population H | 1 | Fortes-Lima et al., 2021; ***Verdu and Rosenberg, 2011*** |
| | Min ASD-MDS African admixture proportions in population H | 1 | Fortes-Lima et al., 2021; ***Verdu and Rosenberg, 2011*** |
| | Max ASD-MDS African admixture proportions in population H | 1 | Fortes-Lima et al., 2021; ***Verdu and Rosenberg, 2011*** |
| | Deciles of ASD-MDS African admixture proportions in population H | 9 | Fortes-Lima et al., 2021; ***Verdu and Rosenberg, 2011*** |
| | Mode ASD-MDS 'African-European angles' in population H | 1 | This study; ***Appendix 1—figure 2*** |
| | Mean ASD-MDS 'African-European angles' in population H | 1 | This study; ***Appendix 1—figure 2*** |
| | Variance ASD-MDS 'African-European angles' in population H | 1 | This study; ***Appendix 1—figure 2*** |
| | Skewness ASD-MDS 'African-European angles' in population H | 1 | This study; ***Appendix 1—figure 2*** |
| | Kurtosis ASD-MDS 'African-European angles' in population H | 1 | This study; ***Appendix 1—figure 2*** |
| | Min ASD-MDS 'African-European angles' in population H | 1 | This study; ***Appendix 1—figure 2*** |
| | Max ASD-MDS 'African-European angles' in population H | 1 | This study; ***Appendix 1—figure 2*** |
| | Deciles of ASD-MDS 'African-European angles' in population H | 9 | This study; ***Appendix 1—figure 2*** |
| between populations | $F_{ST}$ (African Source - Population H) | 1 | ***Weir and Cockerham, 1984*** |
| | $F_{ST}$ (European Source - Population H) | 1 | ***Weir and Cockerham, 1984*** |
| | Mean ASD (African Source - Population H) | 1 | ***Bowcock et al., 1994*** |
| | Mean ASD (European Source - Population H) | 1 | ***Bowcock et al., 1994*** |

*Table 3 continued on next page*

*Table 3 continued*

| Summary Statistics for ABC inference | Number of statistics | Reference |
|---|---|---|
| $f_3$ (Population H; European Source, African Source) | 1 | *Patterson et al., 2012* |

island or in Cabo Verde as a whole, consistently with theoretical expectations (Fortes-Lima et al., 2021; *Verdu and Rosenberg, 2011*).

Finally, when considering all Cabo Verde-born individuals as a single random-mating population without distinguishing birth-islands, our *MetHis*-RF-ABC scenario-choice identifies the Afr2Pulses-Eur2Pulses scenario as the winning scenario (*Appendix 1—figure 4B*), thus consistent with the scenario most often found as the winner among Cabo Verde islands considered as the target admixed population in nine separate *MetHis*-RF-ABC analyses.

## *MetHis*–Neural Network (NN)-ABC posterior parameter estimations

For individuals on each Cabo Verdean birth island separately, we performed NN-ABC joint posterior parameter estimation based on 100,000 *MetHis* simulations under the winning scenario (Fortes-Lima et al., 2021): Afr2Pulses-Eur2Pulses in Santiago, Fogo, Santo Antão São Nicolau, Brava, and Maio; Afr2Pulses-EurRecurring in Boa Vista and Sal; and AfrRecurring-Eur2Pulses in São Vicente (*Figure 7B*). For each island separately, detailed posterior parameters' distributions, Credibility Intervals (CI), and cross-validation errors are provided in *Figure 7—figure supplements 1–3* and *Appendix 5—Tables 1–10*. We synthesized our results considering median point-estimates and 50% CI for each scenario parameter in the admixture history of each island in *Figure 7C–D*. We detailed our results and discussion for admixture history inferences for each island separately in **Appendix 5,** in the light of historical data about the peopling of Cabo Verde (*Figure 7—source data 1*).

*Figure 7C* shows that the reproductive population size of Cabo Verde islands remained very low until a strong increase in the last three generations for all islands but Santiago and Brava. In Santiago the population expansion was more linear since the founding of Cabo Verde in the 1460s, while the reproductive size of Brava remained almost constant and low until today.

In summary, *Figure 7D* shows that European and African admixture events throughout the archipelago occurred first during the early peopling history of each island, before the mid-17th century massive expansion of the TAST due to the expansion of the plantation economy (*Eltis and Richardson, 2015*). We find that other admixture events from Europe or Africa, or both, likely occurred much later during, or immediately after, the 19th century abolition of the TAST and of slavery in European colonial empires. Altogether our *MetHis*-ABC results support limited historical admixture having occurred in Cabo Verde during the most intense periods of the TAST between the mid-17th and early 19th centuries. Furthermore, note that we find admixture events often earlier than, or concomitant with, the first perennial peopling of an island. For the islands of Santiago, Fogo, Santo Antão and, to a lesser extent, São Nicolau, initial historical admixture events occurred synchronously to the first perennial settlement of the island. For the islands of Brava, Maio, Boa Vista, and São Vicente, early admixture events occurred long before their first perennial peopling, thus showing that their founding was already largely composed of already admixed individuals. Importantly, note that our *MetHis*-NN-ABC posterior parameter inferences cannot infer all scenario-parameters accurately, as some parameters hardly depart from their respective priors (*Figure 7—figure supplements 1–3*), and the admixture history of the island of Sal remains overall poorly inferred.

Interestingly, *MetHis*-NN-ABC posterior parameter inference results obtained for the 225 Cabo Verde-born individuals grouped in a single random-mating population instead of separately for each island of birth, are largely undifferentiated from their prior distributions, and have very wide CI and large cross-validation errors, for all admixture-history parameters except for the two parameters associated with the most recent pulse of admixture from the African source (*Figure 7—figure supplements 1–3*; *Appendix 5—table 10*). This contrasts with the substantial number of informative posterior-parameter estimations obtained for all islands of birth separately except Sal (*Figure 7*, *Figure 7—figure supplements 1–3*; *Appendix 5—tables 1–9*), despite the much smaller sample sizes used in each one of these separate analyses compared to the analysis considering Cabo Verde as

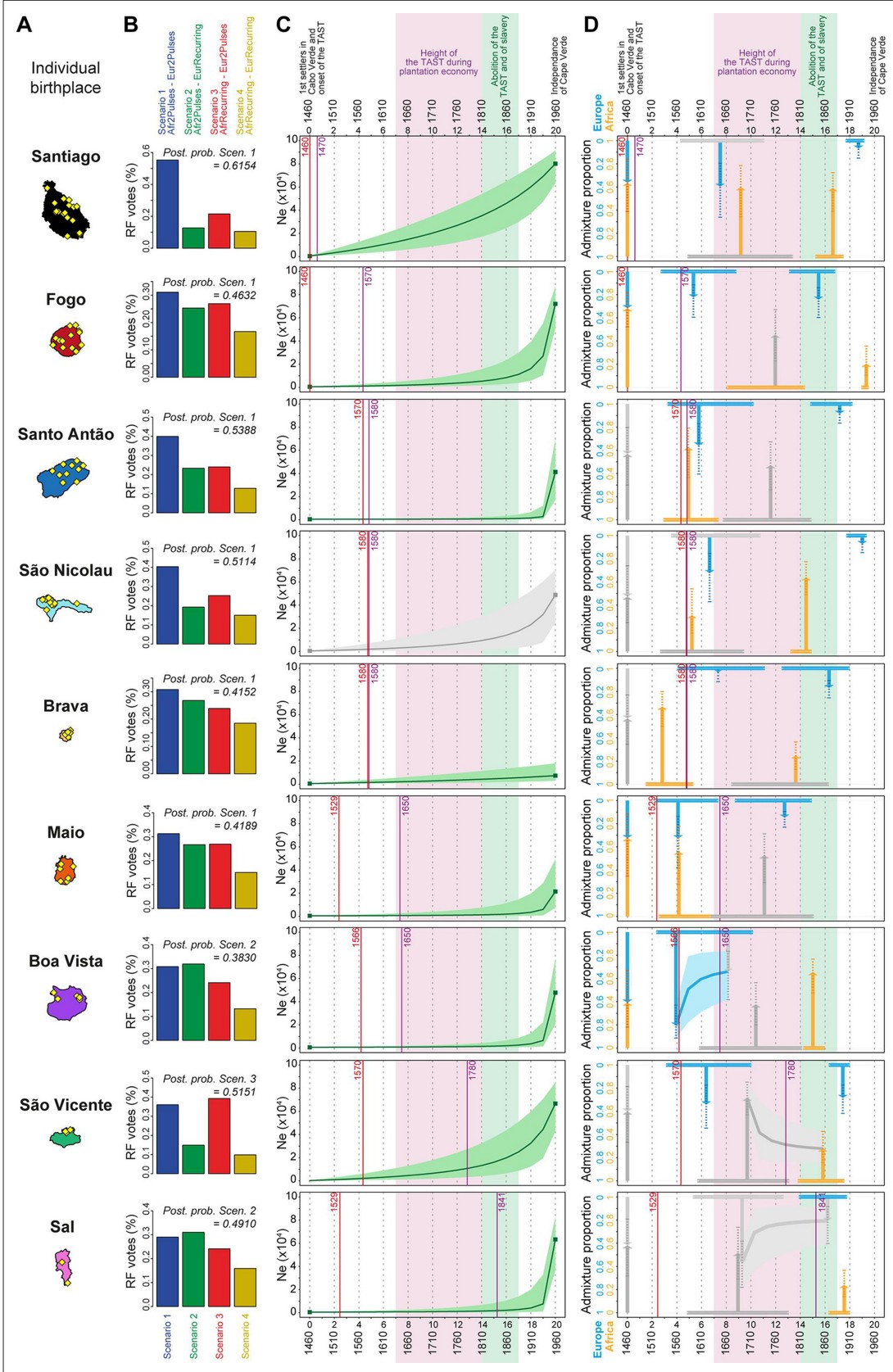

**Figure 7.** Genetic admixture histories of Cabo Verde islands inferred with *MetHis*-Approximate Bayesian Computation. Elements of the peopling-history of Cabo Verde islands are synthesized in *Figure 7—source data 1*, stemming from historical work cited therein. Islands are ordered from top

*Figure 7 continued on next page*

*Figure 7 continued*

to bottom in the chronological order of the first historical census perennially above 100 individuals within an island, indicated with the purple vertical lines. First historical records of the administrative, political, and religious, settlement of an island, are indicated with the red vertical lines. (**A**) Within-island birth-places of 225 Cabo-Verde-born individuals. (**B**) *MetHis*-Random Forest-ABC scenario-choice vote results for each island separately in histogram format. Posterior probabilities of correctly identifying a scenario if correct are indicated for the winning scenario as 'Post. prob. Scen.', above each histogram. (**C**) *MetHis*-Neural Network-ABC posterior parameter distributions with 50% Credibility Intervals for the reproductive population size history of each birth-island separately. (**D**) Synthesis of *MetHis*-NN-ABC posterior parameter median point-estimates and associated 50% CI, for the admixture history of each island under the winning scenario identified with RF-ABC in panel **B**. European admixture history appears in blue, African admixture history in orange. Horizontal bars indicate 50% CI for the admixture time parameters, vertical arrows correspond to median admixture intensity estimates with 50% CI in doted lines. For (**C**) and (**D**), posterior parameter distributions showing limited departure from their respective priors and large CI are greyed, as they were largely unidentifiable in our ABC procedures. Detailed parameter posterior distributions, 95% CI, and cross-validation errors are provided in *Figure 7—figure supplements 1–3* and *Appendix 5—Tables 1–9*. Detailed results description for each island are provided in **Appendix 5**. (**C–D**) The period between the 1630s and the abolition of the TAST in the 1810s, when most enslaved-Africans were deported from Africa by European empires concomitantly to the expansion of the plantation economy (*Eltis and Richardson, 2015*; *Fortes-Lima and Verdu, 2021*), is indicated in light-pink. The period between the abolition of the TAST in the 1810s and the abolition of slavery enacted between 1856 and 1878 throughout the Portuguese empire is indicated in light-green (*Carreira, 2000*). The independence of Cabo Verde occurred in 1975.

The online version of this article includes the following source data and figure supplement(s) for figure 7:

**Source data 1.** Historical landmark chronology for the peopling history of Cabo Verde as provided by previous historical work, respectively for each island.

**Figure supplement 1.** Reproductive-size posterior parameter distributions and associated priors obtained with Neural Network ABC inference for each island separately.

**Figure supplement 2.** African admixture histories posterior parameter distributions and associated priors obtained with Neural Network ABC inference for each island separately.

**Figure supplement 3.** European admixture histories posterior parameter distributions and associated priors obtained with Neural Network ABC inference for each island separately.

a single population. These results further show that the history of admixture substantially differs across Cabo Verde islands and that considering the Cabo Verde archipelago as a single random mating population is inadequate to successfully infer the parameters of its admixture history, consistently with our results from ADMIXTURE, LAI, ROH, and Isolation-By-Distance analyses.

## Discussion
### Which African and European populations contributed genetically to Cabo Verde?
#### The genetic heritage of continental Africa in Cabo Verde

Numerous enslaved-African populations from Western, Central, and South-Western Central Africa were forcibly deported during the TAST to both Cabo Verde and the Americas, as shown by historical demographic records (*Eltis and Richardson, 2015*; *Carreira, 2000*). There is still extensive debate about whether enslaved-Africans remained or more briefly transited in Cabo Verde during the most intense period of the TAST, in the 18th and 19th centuries, when the archipelago served as a slave-trade platform between continents (*Carreira, 2000*; *Patterson, 1988*; *Brooks, 2006*); the question of the duration of stay of enslaved individuals at a given location being also of major interest throughout the Americas during the TAST (*Eltis, 2002*; *Berlin, 2010*). In this context, previous genetic studies considering a relatively limited number of populations from mainland Europe and Africa, and/or limited numbers of Cabo Verdean islands of birth, suggested that mainly continental West Africans and South Europeans were at the root of Cabo Verde genetic landscape (*Micheletti et al., 2020*; *Verdu et al., 2017*; *Beleza et al., 2013*).

In this context, our genetic results favor scenarios where mostly certain West Western African Senegambian populations only (Mandinka and Wolof in our study) contribute to the genetic makeup of Cabo Verde (*Figures 2–3*). Other Western, Central, and South-Western African populations historically also forcibly deported during the TAST seem to have had very limited contributions to the genomic diversity of most Cabo Verde islands, and virtually no contribution to that of Brava, Fogo, and Santiago.

This could be due to Cabo Verde being only a temporarily waypoint for these latter enslaved-African populations between Africa, the Americas, and Europe, but would also be consistent with additional socio-historical processes (see below). Interestingly, and further echoing these genetic results, the Cabo Verdean Kriolu language carries specific signatures mainly from the Mande language-family, and Wolof and Temne languages from Western Africa, and largely more limited signatures of Kikongo and Kimbundu Bantu languages from Central and South-Western Africa (*Quint, 2000*; *Lang, 2009*).

These results contrast with the admixture patterns identified in other enslaved-African descendant populations in the Americas in our dataset (African-American and Barbadian, *Figures 2–3*), and in previous studies (*Micheletti et al., 2020*; *Ongaro et al., 2019*; *Mathias et al., 2016*; *Gouveia et al., 2020*; *Martin et al., 2017*). Indeed, the origins of African ancestries in numerous populations throughout the Caribbean and the Americas traced to varied continental African regions, from Western to South-Western Africa, thus qualitatively consistent with the known diversity of slave-trade routes used between continents and within the Americas after the Middle Passage.

## The genetic heritage of continental Europe in Cabo Verde

After the initial settlement of Cabo Verde by Portuguese migrants, temporary changes of European dominion in certain islands, newly developing commercial opportunities, and intense piracy during the 16th and 17th centuries have triggered different European populations to migrate to the archipelago (*Albuquerque and Santos, 1991*; *Albuquerque and Santos, 1995*; *Soares, 2011*).

Nevertheless, these latter historical events do not seem to have left a major signature in the genetic landscape of Cabo Verde (*Figures 2–3*). Instead, we find that Cabo Verdean-born individuals in our dataset share virtually all their European genetic ancestry with Iberian populations, with extremely limited evidence of contributions from other European regions, consistent with previous studies (*Micheletti et al., 2020*; *Verdu et al., 2017*). Interestingly, the reduced diversity of European populations' contributions to the genomic landscape of Cabo Verde is also identified in other enslaved-African descendant populations in our study, as well as in previous studies in Caribbean and American populations (*Moreno-Estrada et al., 2013*; *Baharian et al., 2016*; *Gouveia et al., 2020*). Our results thus show that European admixture in enslaved-African descendant populations on both sides of the Atlantic, as represented here by Cabo Verde, the Barbadians ACB and the African American ASW, mainly stem from the gene-pool of the European empires historically and socio-economically dominant locally, rather than subsequent European migrations (*Fortes-Lima and Verdu, 2021*).

Altogether, note that, in our local-ancestry inferences, we considered as reference source populations varied existing populations from continental mainland Africa and Europe, categorized as such from sampling information and geographic location only, prior to any genetic investigation. Therefore, in these analyses, we cannot disentangle the fraction of European admixture in Cabo Verdean genomes stemming directly from European migrations after the 1460s, from the fraction stemming from the European genetic legacy in continental African source populations acquired whether during more ancient migrations which occurred before the peopling of Cabo Verde (e.g. *Busby et al., 2016*), or since then during the European colonial expansion in Sub-Saharan Africa. Symmetrically, we cannot disentangle the fraction of African admixture in Cabo Verde stemming directly from continental Africa after the 1460s, from the fraction inherited from Africa-Europe admixture events that may have occurred in Europe prior to the peopling of Cabo Verde or during the colonial era until today. Disentangling both genetic heritages will require, in future studies, the explicit modelling of such possible admixture histories within African and European ancestral populations at the source of the Cabo Verde genetic landscape, and would also benefit from including data from North-African populations in our reference panels.

## Genetic and linguistic isolation-by-distance and recent demographic expansion in Cabo Verde

A scenario of island peopling via a series of founding events followed by slow-growing population sizes and local isolation due to reduced genetic and linguistic parent-offspring dispersal would consistently explain the increasing differentiation of island-specific patterns with increasing values of $K$

found with ADMIXTURE, ASD-MDS, and isolation-by-distance results (*Figures 2–5*, *Table 1*), as well as *MetHis*-ABC demographic inferences (*Figure 7C*).

Indeed, *MetHis*-ABC results (*Figure 7C*) show the long period of small relatively constant reproductive sizes until the very recent strong, hyperbolic, increases in most Cabo Verdean islands; with the notable exceptions of, (*i*) the linear increase in Santiago, the political and commercial center of Cabo Verde throughout the colonial history of the archipelago, and (*ii*) the relatively constant reduced reproductive sizes in Brava until today. Altogether, this result was expected as the dry Sahelian climate of Cabo Verde with scarcely accessible water resources, recurring famines and epidemic outbreaks, and the Portuguese crown maintaining a strong control over population movements within Cabo Verde, rendered difficult the perennial peopling of most islands (see *Figure 7—source data 1* and references therein, **Appendix 5**). Furthermore, such demographic scenarios are also consistent with long-ROH patterns reflecting isolation on each Cabo Verdean islands, whereas elevated shorter ancestry-specific ROH patterns likely stemmed from admixture (*Figure 4*), similarly to our results in the ASW and ACB populations and as previously identified (*Szpiech et al., 2019*). Note, however, that while we explored and found, a posteriori, a different demographic regime for each Cabo Verde island separately, with constant, hyperbolic, or linear increases of reproductive sizes, we did not consider possible demographic bottlenecks which may also have occurred as a result of the difficult settlement history of Cabo Verde described above. Such possible bottleneck events will need to be explored in the future, a particularly challenging task given the extensive number of competing models to be considered and given that bottleneck intensities and duration parameters have to be co-estimated with admixture parameters over a very short history of 21 generations.

Investigating isolation-by-distance anchored in individual birth-places at a very reduced geographical scale (*Rousset, 1997*; *Barbujani, 1987*; *Malécot, 1975*) within a population and a language allowed us to reveal that effective dispersal from one generation to the next across Cabo Verde islands, and sometimes even within-islands, was surprisingly reduced compared to the large mobility self-reported by participants (*Figure 5*, *Table 1*). Patterns of parent-offspring dispersal at a very local scale ~50 km within populations has seldom been tested using genetics, to our knowledge, in human populations (*Rousset, 1997*; *Barbujani and Sokal, 1991*; *Cavalli-Sforza and Moroni, 2004*; *Verdu et al., 2010*), although isolation-by-distance genetic patterns have been extensively explored to investigate serial founding events and migrations across human populations at varied geographical scales, including in archipelagos contexts (*Creanza et al., 2015*; *Arauna et al., 2022*; *Hunley et al., 2008*; *Ramachandran et al., 2005*). Furthermore, while the geographic distribution of genetic admixture patterns have been explored at much larger geographical scales (e.g. *Arauna et al., 2022*; *Bradburd et al., 2016*; *Gnecchi-Ruscone et al., 2017*), and in particular in enslaved-African descendant populations (*Moreno-Estrada et al., 2013*; *Baharian et al., 2016*), isolation-by-distance patterns for inter-individual differences of genetic admixture fractions at very reduced geographical scales have never been reported to our knowledge.

We also found substantial signals of isolation-by-distance among Kriolu idiolects (i.e. individual manners of uttering Cabo Verdean Kriolu), also anchored in individual birth-places (*Figure 5*, *Table 1*), thus showing striking parallels between the history of biological and cultural dispersal and isolation in Cabo Verde at a micro-geographical scale. Our results show that linguistic admixture patterns were isolated-by-distance within the archipelago, similarly to genetic admixture patterns, which was previously unsuspected in both genetics and linguistic studies of Cabo Verde (*Quint, 2000*; *Verdu et al., 2017*; *Baptista, 2003*; *Beleza et al., 2013*).

Altogether, these joint genetic-linguistic patterns highlight the limited effective genetic and linguistic dispersal from one generation to the next within Cabo Verde, including for genetic and linguistic admixture levels, despite extended mobility of individuals within the archipelago. Both mechanisms may thus underline individual linguistic identity construction processes and the genetic relative isolation across and within Cabo Verdean islands.

Importantly, we considered only random mating processes in our inferences and interpretations. However, the almost complete lack of identifiability of the admixture-history parameters obtained when considering Cabo Verde as a single random-mating population in our ABC inferences (*Figure 7—figure supplements 1–3*, *Appendix 5—Tables 1–9*), and our ROH results together with recent work (*Korunes et al., 2022*), altogether suggest that non-random matting processes significantly influenced Cabo Verdean genetic patterns. Therefore, future studies will need to evaluate how

possible deviations from random-mating in Cabo Verde, such as assortative mating (*Korunes et al., 2022*; *Zaitlen et al., 2017*), may have influenced our results and interpretations. Note that this is a conceptually particularly challenging task in a small census-size population with strong marital stratification where mate-choices have been, by definition, limited during most of the TAST (*Versluys et al., 2021*). Nevertheless, such complex processes may also underlie the joint genetic-linguistic isolation-by-distance patterns anchored in birth-place distances here observed for both diversity and admixture patterns; and would also explain that genetic and linguistic histories of admixture apparently occurred in parallel in Cabo Verde.

## Histories of genetic admixture in Cabo Verde

### Early admixture in Cabo Verde, limited admixture during the plantation economy era

While we expected recurring African admixture processes due to the known history of regular forced migrations from Africa during most of Cabo Verde history (*Carreira, 2000*), our *MetHis*–ABC scenario-choices indicate that, qualitatively, these demographic migrations did not necessarily translate into clearly recurring gene-flow processes to shape genetic patterns in most of Cabo Verde islands (*Figure 7B*). Indeed, African admixture processes in all islands, except São Vicente, seem to have occurred during much more punctual periods of Cabo Verdean history. Our *MetHis*–ABC posterior parameter estimations further highlighted often largely differing admixture histories across Cabo Verde islands (*Figure 7C–D* and **Appendix 5**).

We find that admixture from continental Europe and Africa occurred first early during the TAST history, concomitantly with the successive settlement of each Cabo Verdean island between the 15th and the early 17th centuries (*Figure 7D*). Furthermore, we find that the most intense period of enslaved-African deportations during the TAST via Cabo Verde, between the mid-17th and early-19th centuries during the expansion of the plantation economic system in the Americas and Africa (*Eltis and Richardson, 2015*; *Carreira, 2000*; *Albuquerque and Santos, 1991*; *Albuquerque and Santos, 1995*), seem to have left a limited signature in the admixture patterns of most Cabo Verdean islands today. Interestingly, previous studies also highlighted that admixture in enslaved-African descendants in the Caribbean may have occurred first early in the European colonial expansion in the region in the 16th century, and then much later towards the end of the TAST at the end of the 18th century, and had thus been relatively limited during most of the plantation economy era (*Moreno-Estrada et al., 2013*; *Fortes-Lima et al., 2018*). Together with our results, this illustrates the apparent discrepancy between intense demographic forced migrations during the TAST and genetic admixture signatures at least in certain populations on both sides of the Atlantic. Indeed, in contrast with these results, numerous other enslaved-African populations in the Americas have, instead, shown substantial historical admixture inferred to have occurred during the plantation economy era, hence exemplifying the diversity of admixture histories experienced locally by enslaved-Africans descendant populations during the TAST (*Ongaro et al., 2019*; *Baharian et al., 2016*).

The inferred lack of admixture events in Cabo Verde during the height of the TAST could be due to newly deported enslaved-Africans being only transiting via Cabo Verde before being massively re-deported to other European colonies during this era (*Carreira, 2000*; *Patterson, 1988*; *Brooks, 2006*). Furthermore, and not mutually exclusive with this latter hypothesis, historians reported, in Cabo Verde and other European colonies in the Americas, that relationships between enslaved and non-enslaved communities largely changed with the expansion of plantation economy at the end of the 17th century. These changes are often referred to as the shift from Societies-with-Slaves to Slave-Societies (*Eltis and Richardson, 2015*; *Berlin, 2009*; *Chaudenson, 2001*). Slave-Societies legally enforced the socio-marital and economic segregation between communities, and coercively controlled relationships between new enslaved-African migrants and pre-existing enslaved-African descendant communities more systematically and violently than Societies-with-Slaves (*Berlin, 2009*)[p.15-46,95-108],(*Carreira, 2000*)[p.281-319]. The high prevalence of segregation during the height of the plantation economy could have limited genetic admixture between enslaved-African descendants and non-enslaved communities of European origin, as well as admixture between new migrants, forced or voluntary, and pre-existing Cabo Verdeans; notwithstanding the known history of dramatic sexual abuses during and before this era. This could consistently explain our observations of a relative lack of diverse African or European origins in Cabo Verdean genomes despite the known geographical diversity of populations

deported and emigrated to the archipelago throughout the TAST. Furthermore, with legally enforced segregation, we might expect more marital pairings than before to occur among individuals with common origins; i.e. between two individuals with the same continental African or European origin. Such ancestry-specific marriages triggered by socio-cultural segregation would be consistent with our ROH patterns (*Figure 4*), also depending on how long such mate-choice behaviors persisted after the end of legal segregation. We note, however, that we have not formally tested this influence on ROH and ancestry patterns and that a careful consideration of alternate explanations, such as temporally varying admixture contributions over time or a severe bottleneck in one of the ancestral populations, would be important to consider in such future analyses.

In this context, the diversified African ancestries here found in the Americas (*Figures 2–3*), consistently with previous studies showing admixture events occurring before or during the height of plantation economy in the Americas (*Micheletti et al., 2020*; *Ongaro et al., 2019*; *Baharian et al., 2016*), would suggest that the gene-pool of enslaved-Africans communities admixing with Europeans in the Americas since the 16th century often involved, at a local scale, multiple African source populations, thus reflecting the multiple slave-trade routes between continents and within the Americas. Conversely in Cabo Verde, the early onset of the TAST during the 15th century likely privileged commercial routes with nearby Senegambia (*Carreira, 2000*)[p.31-54,281-319], thus favoring almost exclusively admixture events with individuals from this region and from certain populations only. Altogether, our results in Cabo Verde contrasting with certain other enslaved-African descendant populations in the Americas, highlight the importance of early admixture processes and socio-cultural constraints changes on inter-marriages throughout the TAST, which likely durably influenced genomic diversities in descendant populations locally, on both sides of the Atlantic.

## Admixture in Cabo Verde after the abolition of the TAST and of Slavery

Finally, we find that recent European and African admixture in Cabo Verde occurred mainly during the complex historical transition after the abolition of the TAST in European colonial empires in the 1800s and the subsequent abolition of slavery between the 1830s and the 1870s (*Figure 7D*). Historians have shown that these major historical shifts strongly disrupted pre-existing segregation systems between enslaved and non-enslaved communities (*Eltis and Richardson, 2015*)[p.271-290], (*Carreira, 2000*)[p.335-362], (*Cooper et al., 2000*). In addition, an illegal slave-trade flourished during this era, bringing numerous enslaved-Africans to Cabo Verde outside of the official routes (*Carreira, 2000*)[p.363-384] (*Conrad, 1969*). Altogether, our results indicate increased signals of European and African admixture events in almost every island of Cabo Verde during this period, and were thus consistent with a change of the social constraints regarding admixture and forced displacements of enslaved-descendants that had prevailed over the preceding 200 years of the TAST. These results were largely consistent with previous studies elsewhere in the Caribbean (*Moreno-Estrada et al., 2013*; *Fortes-Lima et al., 2018*), and Central and South America (*Ongaro et al., 2019*; *Gouveia et al., 2020*); showing, at a local scale, the major influence of this recent and global socio-historical change in inter-community relationships in European colonial empires on either sides of the ocean.

## Perspectives

Altogether, our results highlight both the unity and diversity of the genetic peopling and admixture histories of Cabo Verde islands, the first colony peopled by Europeans in Sub-Saharan Africa, resulting from the sociocultural and historical complexity of the Trans-Atlantic Slave Trade and European colonial expansions since the 15th century. Our results obtained at a micro-geographical scale reveal the fundamental importance of the early TAST history, before the expansion of the plantation economy, in durably shaping the genomic and cultural diversities of enslaved-African descendant populations in both Africa and the Americas.

Importantly, we considered only the genome-wide autosomal admixture history of Cabo Verde in this study, and therefore did not explore possible sex-biased admixture processes. However, previous studies demonstrated the strong sex-biased admixture processes involved in Cabo Verde using sex-specific genetic markers (*Gonçalves et al., 2003*; *Beleza et al., 2012*), similarly as in other enslaved-African descendant populations in the Americas (*Fortes-Lima and Verdu, 2021*). Furthermore, previous theoretical work considering sex-specific mechanistic admixture models showed that sex-biased admixture processes may possibly bias historical inferences based only on autosomal data

(*Goldberg et al., 2014*). It will thus be important, in future studies, to explore how this sex-biased admixture history may have influenced the ABC inferences here conducted; for instance, via sex-specific developments of *MetHis*-ABC.

Future work will need to formally test the serial-founder hypothesis here proposed to be at the root of the observed genetic and linguistic patterns within Cabo Verde, and thus compare the numerous possible routes for such a peopling history across islands within the archipelago. In particular, it will be of interest to investigate, then, the series of bottlenecks concomitant to each genetic and linguistic founding event; a much needed but challenging task considering the very recent history of the archipelago founded only 21 generations ago, and the historically-known small census sizes echoed in the relatively small reproductive sizes here identified in almost every island until the 20th century.

Our novel results together with their methodological limitations massively beg for future work further complexifying the admixture models here considered, as well as incorporating other summary-statistics such as admixture-LD and sex-specific statistics. This will allow to further dissect the admixture processes that gave birth to enslaved-African descendant populations, on both sides of the Atlantic.

## Materials and methods
### Cabo Verde genetic and linguistic datasets

We conducted joint sampling of anthropological, genetic, and linguistic data in Cabo Verde with the only inclusion criteria that volunteer-participants be healthy adults with Cabo Verdean citizenship and self-reporting speaking Kriolu (*Verdu et al., 2017*). Between 2010 and 2018, six interdisciplinary sampling-trips were conducted to interview 261 participants from more than thirty interview-locations throughout the archipelago (*Figure 1*).

### Familial anthropology and geography data

The 261 Cabo Verdean individuals were each interviewed to record a classical familial anthropology questionnaire on self-reported life-history mobility. In particular, we recorded primary residence location, birth location, parental and grand-parental birth and residence locations, and history of islands visited in Cabo Verde and foreign experiences (*Verdu et al., 2017*). Furthermore, we also recorded age, sex, marital status, and cumulative years of schooling and higher education (for 211 individuals only), and languages known and their contexts of use.

GPS coordinates for each reported location were acquired on site during field-work, supplemented by paper maps and Google Earth for non-Caboverdean locations and islands that we did not visit (Sal and Maio). While participants' birth-locations were often precise, increasing levels of uncertainty arose for the reported parental and grand-parental locations. We arbitrarily assigned the GPS coordinates of the main population center of an island when only the island of birth or residence could be assessed with some certainty by participants. All other uncertain locations where recorded as missing data. Figure maps were designed with the software *QuantumGIS* v3.10 'București' and using Natural Earth free vector and raster map data (https://www.naturalearthdata.com).

### Genome-wide genotyping data

The 261 participants each provided 2 mL saliva samples collected with DNAGenotek OG-250 kits, and complete DNA was extracted following manufacturer's recommendations. DNA samples were genotyped using an Illumina HumanOmni2.5Million-BeadChip genotyping array following manufacturer's instructions. We followed the quality-control procedures for genotypic data curation using Illumina's GenomeStudio Genotyping Module v2.0 described in *Verdu et al., 2017*. Genotype-calling, population-level quality-controls, and merging procedures are detailed in *Appendix 1—figure 1*.

In summary, we extracted a preliminary dataset of 259 Cabo Verdean Kriolu-speaking individuals, including relatives, genotyped at 2,118,835 polymorphic di-allelic autosomal SNPs genome-wide. We then merged this dataset with 2504 worldwide samples from the 1000 Genomes Project Phase 3 *Auton et al., 2015*; with 1307 continental African samples from the African Genome Variation Project (*Gurdasani et al., 2015*; *Network et al., 2019*) (EGA accession number EGAD00001000959); and with 1235 African samples (*Patin et al., 2017*; *Perry et al., 2014*) (EGA accession number EGAS00001002078). We retained only autosomal SNPs common to all data sets, and excluded

one individual for each pair of individuals related at the 2nd degree (at least one grand-parent in common) as inferred with KING (*Manichaikul et al., 2010*), following previous procedures (*Verdu et al., 2017*).

After merging all datasets (*Appendix 1—figure 1*), we considered a final working-dataset of 5157 worldwide unrelated samples, including 233 unrelated Cabo Verdean Kriolu-speaking individuals, of which 225 were Cabo-Verde-born, genotyped at 455,705 autosomal bi-allelic SNPs (*Figure 1*; *Figure 1—source data 1*). Note that, for this working-dataset, the fraction of missing genotypes was very low and equaled $7.0 \times 10^{-4}$ on average within Cabo Verdean islands of birth (SD = $3.0 \times 10^{-4}$ across islands).

## Individual utterances of Kriolu

We collected linguistic data for each of the 261 Cabo Verdean individuals using anthropological linguistics questionnaires, and semi-directed interviews. Each participant was shown a brief (~6 min) speech-less movie, '*The Pear Story'* (*Chafe, 1980*), after which they were asked to narrate the story as they wanted in 'the Kriolu they speak every day at home'. Discourses were fully recorded without interruption, whether individuals' discourses were related to the movie or not. Then each discourse was separately fully transcribed using the orthographic convention of the Cabo Verdean Kriolu official alphabet "Alfabeto Unificado para a Escrita da Língua Cabo-verdiana (ALUPEC)" (Decreto-Lei n° 67/98, 31 de Dezembro 1998, I Série n° 48, Sup. B. O. da República de Cabo Verde).

Building on the approach of *Verdu et al., 2017*, we were interested in inter-individual variation of "ways of speaking Kriolu" rather than in a prescriptivist approach to the Kriolu language. Thus, we considered each utterance as defined in *Croft, 1996*[p.107]: "a particular instance of actually-occurring language as it is pronounced, grammatically structured, and semantically interpreted in its context". Transcripts were parsed together and revealed 4831 (*L*=4831) unique uttered items among the 92,432 uttered items transcribed in total from the 225 discourses from the genetically-unrelated Cabo Verde-born individuals. To obtain these counts, we considered phonetic, morphological, and syntactic variation of the same lexical root items that were uttered/pronounced differently, and we excluded from the utterance-counts onomatopoeia, interjections, and names. Note that we were here interested in the diversity of realizations in the Kriolu lexicon, including within the same individuals. In other words, we are interested in both between-speaker and within-speaker variation. Also note that a very limited number of English words were pronounced by particular individuals (10 utterances each occurring only once), and were kept in utterance-counts.

## Individual Kriolu utterance frequencies

The list of unique uttered items was then used to compute individual's specific vectors of uttered items' relative frequencies as, for each genetically unrelated Cabo Verde-born individual *i* (in [1, 225]) and each unique uttered item *l* (in [1, *L*=4831]), $f_{i,l} = n_{i,l}/\sum_{j=1}^{L} n_{i,j}$, where $n_{i,l}$ is the absolute number of times individual *i* uttered the unique item *l* over her/his entire discourse, $f_i$ being the vector $(f_{i,1}, f_{i,2}, \ldots, f_{i,L})$. We compared vectors of individuals' utterance-frequencies by computing the inter-individual pairwise Euclidean distance matrix as, for all pairs of individuals *i* and *j*, $d\left(f_i, f_j\right) = \sqrt{\sum_{l=1}^{L}\left(f_{i,l} - f_{j,l}\right)^2}$, (Verdu et al., 2017).

## African origin of Kriolu utterances

We categorized each of the 4831 unique uttered items separately into five linguistic categories (*Verdu et al., 2017*). Category A included only unique utterances directly tracing to a known African language and comprised 88 unique items occurring 3803 times out of the 92,432 utterances. Category B included only items with a dual African and European etymology, that is items of a European linguistic origin strongly influenced in either meaning, syntax, grammar, or form by African languages or vice versa, attesting to the intense linguistic contacts at the origins of Cabo Verdean Kriolu, and comprised 254 unique items occurring 6960 times. Category C included 4432 items (occurring 73,799 times) with strictly Portuguese origin and not carrying identifiable traces of significant African linguistic origin or influence. Category D included 26 items occurring 6762 times with potential, not attested by linguists, traces of African languages' phonetic or morphologic influences. Finally, Category U included the 10

English utterances occurring 10 times and the 21 unique Kriolu utterances occurring 1089 times of unknown origin as they could not be traced to African or European languages.

Following (*Verdu et al., 2017*), we defined an 'African-utterances score' based, conservatively, on the utterance frequencies obtained separately for items in Category A, Category B, and merging Categories A and B, as, for individual $i$ and the set of utterances in each category denoted $Cat$ (in [A; B; A&B]), $Z_{i,Cat} = \sum_{l=1}^{L_{Cat}} f_{i,l}$, with $L_{Cat}$ the number of uttered items in the corresponding category, and $f_{i,l}$ defined as previously.

## Population genetics descriptions

### Allele Sharing Dissimilarities, Multidimensional Scaling, and ASD-MDS admixture estimates

We calculated pairwise Allele Sharing Dissimilarities (*Bowcock et al., 1994*) using the *asd* software (v1.1.0a; https://github.com/szpiech/asd; *Szpiech, 2020*), among the 5157 individuals in our world-wide dataset (*Figure 1*; *Figure 1—source data 1*), using 455,705 autosomal SNPs, considering, for a given pair of individuals, only the SNPs with no missing data. We then projected this dissimilarity matrix in three dimensions with metric Multidimensional Scaling using the *cmdscale* function in R (*R Development Core Team, 2020*). We conducted successive MDS analyses on different individual subsets of the original ASD matrix, by excluding, in turn, groups of individuals and recomputing each MDS separately (*Appendix 2—figures 1–4*; *Figure 2—figure supplement 1*). Lists of populations included in each analysis can be found in *Figure 1—source data 1*. 3D MDS animated plots in *gif* format for *Figure 2*, *Figure 2—figure supplement 1*, and *Appendix 2—figures 1–4* were obtained with the *plot3d* and *movie3d* functions from the R packages *rgl* and *magick*.

Recently admixed individuals are intuitively expected to be at intermediate distances between the clusters formed by their putative proxy source populations on ASD-MDS two-dimensional plots. A putative estimate of individual admixture rates can then be obtained by projecting admixed individuals orthogonally on the line joining the respective centroids of each proxy-source populations and, then, calculating the distance between the projected points and either centroid, scaled by the distance between the two centroids (*Appendix 1—figure 2*; Fortes-Lima et al., 2021; *Paschou et al., 2007*). We estimated such putative individual admixture rates in Cabo Verdean, ASW, and ACB individuals, considering sets of individuals for the putative proxy-source populations identified visually and resulting from our ASD-MDS decomposition (**Appendix 2**).

### ADMIXTURE-CLUMPP-DISTRUCT and $f_3$-admixture analyses

Based on ASD-MDS explorations, we focused on the genetic structure of individuals born in Cabo Verde compared to that of other admixed populations related to TAST migrations. Therefore, we conducted ADMIXTURE analyses (*Alexander et al., 2009*) using 1,100 individuals from 22 populations: four from Europe, 14 from Africa, the USA-CEU, the African-American ASW, the Barbadian-ACB populations, and the North Chinese-CHB population as an outgroup (*Figure 1—source data 1*). To limit clustering biases due to uneven sampling, we randomly resampled without replacement 50 individuals, once, for each one of these 22 populations. Furthermore, we also included all 44 Angolan individuals from four populations in the analyses, as no other samples from the region were available in our dataset. In addition to the 1100 individuals hence obtained, we included the 225 Cabo Verde-born individuals.

Following constructor recommendations, we pruned the initial 455,705 SNPs set for low LD using *plink* (*Purcell et al., 2007*) function *--indep-pairwise* for a 50 SNP-window moving in increments of 10 SNPs and a $r^2$ threshold of 0.1. We thus conducted all subsequent ADMIXTURE analyses considering 1,369 individuals genotyped at 102,543 independent autosomal SNPs.

We performed 30 independent runs of ADMIXTURE for values of $K$ ranging from 2 to 15. For each value of $K$ separately, we identified groups of runs providing highly similar results (ADMIXTURE 'modes'), with Symmetric Similarity Coefficient strictly above 99.9% for all pairs of runs within a mode, using CLUMPP (*Jakobsson and Rosenberg, 2007*). We plotted each modal result comprising two ADMIXTURE runs or more, for each value of $K$ separately, using DISTRUCT (*Rosenberg, 2003*). We

evaluated within-population variance of individual membership proportions as $F_{ST}/F_{ST}^{max}$ values using FSTruct (*Morrison et al., 2021*) with 1000 Bootstrap replicates per population, for the ADMIXTURE mode result at $K=2$ (*Figure 3—figure supplement 1*).

Finally, we computed, using ADMIXTOOLS (*Patterson et al., 2012*; *Maier et al., 2022*), $f_3$-admixture tests considering as admixture targets each Cabo Verdean birth-island, the ASW, and the ACB separately, with, as admixture sources, in turn all 108 possible pairs of one continental European population (Source 1) and one continental African population (Source 2), or the East Asian CHB (Source 1) and one continental African population (Source 2). For all tests, we used the same individuals, population groupings, and genotyping dataset as in the previous ADMIXTURE analyses (*Figure 3—figure supplement 2*), and considered the no-inbreeding option in ADMIXTOOLS.

## Local-ancestry inferences

To identify all populations sharing a likely common ancestry with the Cabo Verdean, ASW, or ACB individuals in our dataset using local-ancestry haplotypic inferences, we considered the same sample-set as for the ADMIXTURE analysis (*Figure 1—source data 1*), including all 455,705 SNPs from the merged dataset.

### Phasing with ShapeIT2

We first phased individual genotypes using SHAPEIT2 (*Delaneau et al., 2012*) for the 22 autosomal chromosomes separately using the joint Hap Map Phase 3 Build GRCh38 genetic recombination map (*The International HapMap 3 Consortium, 2010*). We considered default parameters using phasing windows of 2 Mb and 100 states per SNP. We considered by default 7 burn-in iterations, 8 pruning iterations, 20 main iterations, and missing SNPs were imputed as monomorphic. Finally, we considered a 'Ne' parameter of 30,000, and all individuals were considered unrelated.

### Chromosome painting with ChromoPainter2

We determined the possible origins of each Cabo Verdean, ASW, and ACB individual pair of phased haplotypes among European, African, USA-CEU, and Chinese-CHB populations using CHROMO-PAINTER v2 (*Lawson et al., 2012*) with the same recombination map used for phasing. Following authors' recommendations, we conducted a first set of 10 replicated analyses on a random subset of 10% of the individuals for chromosomes 2, 5, 12, and 19, which provided a posteriori Ne = 233.933 and $\theta$=0.0004755376, on average by chromosome weighted by chromosome sizes, to be used in the subsequent analysis. We then conducted a full CHROMOPAINTER analysis using these parameters to paint all individuals in our dataset, in turn set as Donor and Recipient, except for Cabo Verde, ACB, and ASW individuals set only as Recipient. Finally, we combined painted chromosomes for each individual in the Cabo Verdean, ASW, and ACB population, separately.

### Estimating possible source populations for the Cabo Verde gene-pool using SOURCEFIND

We used CHROMOPAINTER results aggregated for each Cabo Verdean, ACB, and ASW individual separately and conducted two SOURCEFIND (*Chacón-Duque et al., 2018*) analyses using all other populations in the dataset as a possible source, separately for four or six possible source populations ('surrogates'), to allow a priori for symmetric or asymmetric numbers of African and European source populations for each target admixed population. We considered 400,000 MCMC iteration steps (including 100,000 burn-in) and kept only one MCMC step every 10,000 steps for final likelihood estimation. Each individual genome was divided in 100 slots with possibly different ancestry, with an expected number of surrogates equal to 0.5 times the number of surrogates, for each SOURCEFIND analysis. We aggregated results obtained for all individuals in the ACB, ASW, and each Cabo Verdean birth-island, separately. We present the highest likelihood results across 20 separate iterations of the SOURCEFIND analysis in *Figure 3B*. The second-best results were highly consistent and thus not shown.

## Runs of homozygosity (ROH)

### Calling ROHs

Considering the same sample and SNP set as in the above local-ancestry analyses (*Figure 1—source data 1*), we called ROH with GARLIC (*Szpiech et al., 2017*). For each population separately, we ran GARLIC using the weighted logarithm of the odds (wLOD) score-computation scheme, with a genotyping-error rate of $10^{-3}$ (a likely overestimate), and using the same recombination map as for phasing, window sizes ranging from 30 to 90 SNPs in increments of 10 SNPs, 100 resampling to estimate allele frequencies, and all other GARLIC parameters set to default values. We only considered results obtained with a window size of 40 SNPs, which was the largest window size associated with a bimodal wLOD score distribution and a wLOD score cutoff between the two modes, for all populations.

For each population and Cabo Verdean island separately, we considered three classes of ROH that correspond to the approximate time to the most recent common ancestor of the IBD haplotypes, which can be estimated from the equation $g=100/2l$, where $l$ is the ROH length in cM and $g$ is the number of generations to the most recent common ancestor of the haplotypes (*Thompson, 2013*). Short ROH are less than 0.25 cM, reflecting homozygosity of haplotypes from more than 200 generations ago; medium ROH are between 0.25 cM and 1 cM reflecting demographic events having occurred between approximately 200 and 50 generations ago; and long ROH are longer than 1 cM, reflecting haplotypes with a recent common ancestor less than 50 generations ago. In *Figure 4A–B* and *Appendix 4—figure 1*, we plot the distribution of the summed length of ROH per individual per size-classes.

### Intersecting ROH and local ancestry painting

Using the same phasing results as described above, we conducted 10 EM iterations of the RFMIX2 (*Maples et al., 2013*) algorithm to assign, for each Cabo Verdean individual and each SNP on each chromosome, separately, its putative source population of origin among the 24 African, European, Chinese-CHB, and USA-CEU populations. We collapsed the local ancestry assignments for each SNP in each Cabo Verdean individual hence obtained into three continental regions, representing broadly African, broadly European, and broadly East Asian ancestries respectively. Any ancestry call that was assigned a population from the African continent was assigned a category of AFR, any ancestry call that was assigned a population from the European continent was assigned a category of EUR, and any ancestry call that was assigned a population from the East Asian continent was assigned a category of ASN. We considered an approach similar to previous work (*Szpiech et al., 2019*), and intersected local ancestry calls with ROH calls (*Figure 4C–D*).

RFMIX2 only makes local ancestry calls at loci that are present in the dataset. Therefore, a gap of unclassified ancestry exists where inferred ancestry switches between two successive genotyped loci as called by RFMIX2. These gaps necessarily each contain an odd number of ancestry switch points (≥1) absent from our marker set. Therefore, when computing the total ancestry content of an ROH that overlaps one of these ancestry breakpoints, we assign half of the length of this gap to each ancestry classification, effectively extending each local ancestry segment to meet at the midpoint.

We then plotted the length distribution of long ROH for each island and we break out the distributions by ancestral haplotype background: those with only African ancestry, those with only European ancestry, and those that span at least one ancestry breakpoint (*Figure 4C*). We excluded long ROH in East Asian ancestry segments from this and the following analyses, as we found such ancestry overall very limited in the samples (*Appendix 4—figure 2*). We also excluded ROH called with heterozygous ancestry calls (e.g. called with one haplotype called as AFR and the other as EUR). These regions were also rare (*Figure 4—source data 3*).

Finally, we explored how total African/European ancestry in long ROH varies between islands. For each individual, we summed the total amount of each ancestry in long-ROH and plot the distributions across islands (*Figure 4—figure supplement 1*). High levels of a given ancestry in long ROH could stem from an overall high level of that ancestry in that individual. Therefore, for each individual, we computed their global ancestry proportions by summing up the total length of the genome inferred as a given ancestry and dividing by the length of the genome. We then plotted (*Figure 4D*), the difference of an individual's long-ROH ancestry proportion and their global ancestry proportion, for African

and European ancestries separately. Values above zero indicated that a given ancestry is overrepresented in long-ROH relative to genome-wide proportions of that ancestry.

To assess the significance of these deviations, we performed a non-parametric permutation test. For each individual in each island, we randomly permuted the location of all long ROH (ensuring that no permuted ROH overlap), re-computed the local AFR ancestry proportion falling within these permuted ROH, and then subtracted the global ancestry proportion. We then took the mean of this difference across all individuals for each island and repeated the process 10,000 times. As there is negligible ASN ancestry across these individuals, the AFR and EUR proportions essentially add to 1, and therefore we consider an over/under representation of AFR ancestry in long ROH to be equivalent to an under/over representation of EUR ancestry in long ROH. Permutation distributions with observed values are plotted in *Figure 4—figure supplement 2* and permutation p-values are given in *Figure 4—source data 2*.

## Isolation-by-distance: genetic and linguistic diversity within Cabo Verde

We explored genetic pairwise levels of differentiation calculated with ASD as previously, considering the 1,899,878 non-monomorphic SNPs within Cabo Verde obtained after QC Stage 3 (*Appendix 1—figure 1*). We projected the matrix for the 225 unrelated Cabo Verde-born individuals using metric MDS as above. Note that pruning this data set for very low LD using *plink* function *--indep-pairwise* for a 50 SNPs window moving in increments of 10 SNPs and a $r^2$ threshold of 0.025 resulted in 85,425 SNPs for which the ASD matrix was highly correlated with the one using all SNPs (Spearman $\rho$=0.8745, p<2.2 × 10$^{-16}$). In parallel, we described the diversity of Kriolu idiolects (i.e. individuals' manners of speaking Kriolu) among the 225 genetically unrelated Cabo Verde-born individuals by projecting, with metric MDS, the matrix of pairwise Euclidean distances between vectors of individuals' utterance-frequencies (see **Material and Methods** section "Individual Kriolu utterance frequencies"), considering the 4831 unique uttered items.

We then conducted a series of Mantel and partial Mantel correlation tests (*Legendre and Legendre, 1998*), using the *partial.mantel.test* function in the R package *ncf*, with Spearman correlation and 10,000 Mantel permutations, to explore possible isolation-by-distance (*Rousset, 1997*) patterns as well as correlations with other variables of interest. We conducted correlation tests between either the pairwise Euclidean distances between vectors of individuals' utterance-frequencies or genetic ASD separately, and individual pairwise matrices of (*i*) absolute age differences, (*ii*) absolute differences in academic education duration, (*iii*) geographic distances between residence-locations (logarithmic scale), (*iv*) between birth-locations, (*v*) between mothers' birth-locations, (*vi*) between fathers' birth-locations (*Table 1*, *Table 1—source data 1*). All pairwise geographic distances were calculated with the Haversine great-circle distance formulation (*Snyder, 1987*), taking 6371 km for the radius of the Earth, using the *rdist.earth* function in the R package *fields*. Before computing logarithmic distances, we added 1 km to all pairwise distances.

## Isolation-by-distance: genetic and linguistic admixture within Cabo Verde

We further explored isolation-by-distance patterns within Cabo Verde specifically for African genetic and linguistic individual admixture levels. We considered African genetic admixture levels estimated from ADMIXTURE at *K*=2 or ASD-MDS approaches (see **Material and Methods** section 'Population genetics descriptions', *Figure 3*, *Appendix 1—figure 2*), and individual African linguistic admixture as 'African-utterances scores' $Z_{i,Cat}$ as defined in **Material and Methods** section 'African origin of Kriolu utterances' for utterance lists contained in Category A, Category B, or Category A&B, respectively (*Verdu et al., 2017*). For genetic or linguistic admixture levels separately, we computed the pairwise matrix of individual absolute admixture levels differences, and conducted Mantel testing with the different geographical distance matrices as above. Finally, we compared African genetic and linguistic admixture scores using Spearman correlations, throughout Cabo Verde and within all birth-islands, separately.

## Inferring genetic admixture histories in Cabo Verde with *MetHis*-ABC

We aimed at reconstructing the detailed genetic admixture history of each Cabo Verde island separately. To do so, we first design *MetHis* v1.0 (Fortes-Lima et al., 2021) forward-in-time simulations of four competing complex admixture scenarios. We then couple *MetHis* simulation and summary-statistics calculation tools with ABC Random-Forest scenario-choice implemented in the R package

*abcrf* (**Pudlo et al., 2016**), followed by Neural-Network posterior parameter estimation with the R package *abc* (**Csilléry et al., 2012**), under the winning scenario for each island separately.

## Simulating four competing scenarios of complex historical admixture for each Cabo Verde island

ABC inference relies on simulations conducted with scenario-parameter values drawn randomly within prior distributions set explicitly by the user. We used *MetHis* v1.0 (Fortes-Lima et al., 2021) to simulate 60,000 independent autosomal SNP markers in the admixed population H, forward-in-time and individual centered, under the four competing scenarios presented in *Figure 6* and *Table 2* and explicated below. In all four scenarios (*Figure 6*; *Table 2*), we considered, forward-in-time, that the admixed population (Population H) is founded at generation 0, 21 generations before present. Generation 0 thus roughly corresponds to the 15th century when considering an average generation-time of 25 years and sampled individuals born on average between the 1960s and the 1980s and no later than 1990 in our dataset. This is reasonable as historical records showed that Cabo Verde was likely un-inhabited before its initial colonial settlement established in the 1460s on Santiago (**Albuquerque and Santos, 1991**). Due to the recent admixture history of Cabo Verde and as we considered only independent genotyped SNPs, we neglected mutation in our simulations for simplicity.

Following our descriptive analyses results, we considered scenarios with only one 'European' and one 'African' source population, and each Cabo Verde island, separately, as the 'Admixed' recipient population H. This corresponds to the 'two-source' version of the general admixture model from **Verdu and Rosenberg, 2011**, also explored with *MetHis* previously (Fortes-Lima et al., 2021).We therefore considered a single African and European population at the source of all admixture in Cabo Verde, and further considered that both source populations were very large and at the drift-mutation equilibrium during the 21 generations of the admixture process until present. Furthermore, we considered that these source populations were accurately represented, respectively, by the Mandinka from Senegambia and the Iberian-IBS populations in a random genotyping dataset of 60,000 independent autosomal SNPs (see **Results**).

In brief (see below), at each generation, *MetHis* performs simple Wright-Fisher (**Fisher, 1922**; **Wright, 1931**) forward-in-time discrete simulations, individual-centered, in a randomly mating (without selfing) admixed population of $N_g$ diploid individuals. Separately for each $N_g$ individual in the admixed population at generation *g*, *MetHis* draws parents randomly from the source populations and the admixed population itself at the previous generation according to given parameter-values drawn from prior distributions separately for each simulation.

### Hyperbolic increase, linear increase, or constant reproductive population size in the admixed population

We considered, for the admixed population H, a reproductive population size of $N_0$ diploid individuals at generation 0, with $N_0$ in [10,1000], and $N_{20}$ in [100,100,000] at generation 20, such that $N_0 < N_{20}$. In between these two values, we considered for the $N_g$ values at each generation the discrete numerical solutions of an increasing rectangular hyperbola function of parameter $u_N$ in [0,1/2] (Fortes-Lima et al., 2021). Therefore, values of the demographic parameters $N_0 \sim N_{20}$ correspond to simulations with a constant admixed-population H reproductive population size of $N_0$ diploid individuals during the entire 21 generations of the admixture process, whichever the value of $u_N$. Instead, parameter values $N_0 \neq N_{20}$ necessarily correspond to simulations with an increase in reproductive size between $N_0$ and $N_{20}$, steeper with values of $u_N$ closer to 0. Note, thus, that parameter values $N_0 \neq N_{20}$ and $u_N$ close to 0 correspond to simulations where the reproductive size of the admixed population H is roughly constant and equal to $N_0$ at each generation until a very sharp increase to reach $N_{20}$ at the last generation before present. Instead, parameter values $N_0 \neq N_{20}$ and $u_N$ close to 1/2 correspond to simulations with a linear increase in reproductive size between $N_0$ and $N_{20}$.

Therefore, while we do not formally compare competing scenarios with different demographic regimes in this work, each scenario comprises simulations whose parameter values correspond to a variety of constant, hyperbolic increase, or linear increase in reproductive size over the course of the admixture history of each Cabo Verdean island separately. Therefore, ABC parameter estimation of

the demographic parameters $N_0$, $N_{20}$, and $u_N$ should determine, a posteriori, which of the three demographic regimes best explain our data, whichever is the winning admixture scenario among the four in competition.

## Founding the admixed population

At generation 0 (*Figure 6*), the admixed population of size $N_0$ diploid individuals drawn in [10, 1000] is founded with a proportion $s_{Eur,0}$ of admixed individuals' parents originating from the European source drawn in [0,1], and a proportion of $s_{Afr,0}$ parents from the African source drawn in [0,1], with $s_{Eur,0}$ + $s_{Afr,0}$=1. Note that parameter-values of $s_{Afr,0}$, or $s_{Eur,0}$, close to 0 correspond to simulations where the 'admixed' population is initially founded by only one of the sources, and that genetic admixture per se may only occur at the following admixture event.

After the founding of the admixed population H, we considered two different admixture scenarios for either source population's contribution to the gene-pool of population H. In all cases, for all generations $g$ in [1,20] after the initial founding of the admixed population at $g$=0, *MetHis* randomly draws parents from the African source, the European source and the admixed population H respectively in proportions $s_{Afr,g}$, $s_{Eur,g}$, and $h_g$, each in [0,1] and satisfying $s_{Afr,g}$ + $s_{Eur,g}$ + $h_g$ = 1. Then, the software randomly builds gametes for each parent and randomly pairs them, without selfing, to produce $N_g$ diploid individuals in the admixed population.

## Two admixture pulses after foundation

For a given source population hereafter designated 'Source' ('European' or 'African' in our case), after founding at generation 0, we considered scenarios with two additional pulses of admixture ('Source'–2Pulses scenarios). The two pulses occur, respectively, at generation $t_{Source,p1}$ and $t_{Source,p2}$ in [1,20], with $t_{Source,p1} \neq t_{Source,p2}$; and with intensity $s_{Source,tSource,p1}$ and $s_{Source,tSource,p2}$ in [0,1], respectively.

Note that simulations considering values of parameters $t_{Source,p1}$ = $t_{Source,p2}$ +1, and simulations with either $s_{Source,tSource,p1}$ or $s_{Source,tSource,p2}$ close to 0, may strongly resemble those expected under scenarios with only one pulse of admixture after the founding of the admixed population H.

## A period of recurring admixture after foundation

For a given Source population, after founding at generation 0, we considered scenarios with a possible period of recurring admixture, where, during this period, admixture intensity followed a monotonically decreasing trend ("Source"-Recurring scenarios). The period of admixture occurs between times $t_{Source,t1}$ and $t_{Source,t2}$ in [1,20] with $t_{Source,t2} \geq t_{Source,t1}$ +1. The beginning of the admixture period at $t_{Source,t1}$ is associated with admixture intensity $s_{Source,tSource,t1}$ in [0,1]. The end of the admixture period at $t_{Source,t2}$ is associated with intensity $s_{Source,tSource,t2}$ in [0,1] such that $s_{Source,tSource,t1} \geq s_{Source,tSource,t2}$. In between, the admixture intensity values at each generation of the admixture period are the discrete numerical solutions of a decreasing rectangular hyperbola function of parameter $u_{Source}$ in [0,1/2] (Fortes-Lima et al., 2021).

Intuitively, a $u$ parameter value close to 0 corresponds to a sharp pulse of admixture occurring at the beginning of the admixture period of intensity $s_{Source,tSource,t1}$, followed at the next generation by constant recurring admixture of intensity $s_{Source,tSource,t2}$ until the end of the admixture period. Alternatively, a $u$ parameter value close to 1/2 corresponds to a linearly decreasing admixture at each generation of the admixture period, from $s_{Source,tSource,t1}$ to $s_{Source,tSource,t2}$.

Note that in the limit when $s_{Source,tSource,t1} \sim s_{Source,tSource,t2}$, Recurring scenarios correspond to constant recurring admixture of that intensity. Furthermore, simulations with $u$ and $s_{Source,tSource,t2}$ parameter values both close to 0 correspond to scenarios with a single pulse of admixture from a given source after the founding pulse, occurring at time $t_{Source,t1}$ and with intensity $s_{Source,tSource,t1}$.

## Four competing scenarios of admixture from two-source populations in each Cabo Verde island

Finally, we combined the 2Pulses and Recurring scenarios from either the African and European Source populations in order to produce four competing scenarios of admixture for the genetic history of Cabo Verde (*Figure 6*), with the only constraint that at each generation $g$ between 1 and 20, $s_{Afr,g}$ + $s_{Eur,g}$=1 – $h_g$, where $h_g$ is the contribution of the admixed population H to itself at the following generation in [0,1].

## Simulating the admixed population from source-populations for 60,000 independent SNPs with MetHis

As introduced previously, our results showed that the Mandinka from West Western Africa and the Iberian IBS from South West Europe were reasonable proxies of the main source populations for the gene-pool of Cabo Verde, at least when considering a relatively small number of independent autosomal SNPs. We decided to consider both populations as very large and at the drift-mutation equilibrium since the 1460s and the initial founding of Cabo Verde. We thus chose not to explicitly simulate the evolutionary history of the two European and African populations at the source of the genetic history of the Cabo Verde islands.

Instead, we first randomly drew 60,000 independent SNPs, avoiding singletons, from the LD-pruned 102,543 independent SNPs employed for the ADMIXTURE analyses. We then built a single reservoir of 'African' gametes comprising 20,000 haploid genomes of 60,000 independent SNP-sites each, where each allele at each site of a gamete was randomly drawn in the site frequency spectrum observed for the corresponding SNP in the Mandinka proxy source population. Separately, we proceeded similarly to build a reservoir of 20,000 European gametes matching instead the site-frequency spectrum observed in the Iberian IBS at the 60,000 SNPs.

For a given simulation with, at generation 0, given parameter values $s_{Afr,0}$ and $s_{Eur,0}$, each in [0,1] such that $s_{Afr,0} + s_{Eur,0} = 1$, and a given value (in [10, 1000]) of $N_0$ diploid individuals in the admixed population H, MetHis randomly draws two different gametes in the African gamete-reservoir and randomly pairs them to produce one parent from the African source and repeats the process $s_{Afr,0}$ x $2N_0$ times to obtain that number of African parents. In parallel and using the same procedure, MetHis randomly builds $s_{Eur,0}$ x $2N_0$ parents from the European gamete-reservoirs. For each of $N_0$ individuals in the admixed population separately, MetHis then draws randomly a pair of parents among the European and African parents hence obtained, and for each parent separately, builds one haploid gamete by randomly drawing one allele for each 60,000 genotypes. Finally, MetHis pairs both hence obtained gametes to create the novel individual in the admixed population at the following generation, and repeats the procedure (replacing the pair of parents after each random draw in the parental pool) for each of $N_0$ individuals separately.

Then, for the same given simulation, admixture from a source population is set to occur at a given generation $g$ (in [1,20]) associated with a given intensity $s_{source,g}$, keeping in mind that at all $g$ in [1,20], $s_{Afr,g} + s_{Eur,g} + h_g = 1$. MetHis then builds anew $s_{source,g}$ x $2N_g$ reproductive parents from this source population's gamete-reservoir as previously, and in parallel randomly draws $h_g$ x $2N_g$ parents from the admixed population H itself at the previous generation. Then, for each of $N_g$ individuals in the admixed population separately, MetHis randomly draws a pair of different parents (replacing the pair of parents after each random draw in the parental pool), randomly builds haploid gametes from each parent, and pairs them similarly as previously to obtain a new individual at the following generation.

Thus, note that while our source-populations' gamete reservoirs were fixed during the admixture process, the African or the European diploid parents possibly contributing to the gene-pool of the admixed population are randomly built anew, and each produce novel gametes, at each generation and in each simulation separately. Importantly, note that recombination is thus not a parameter in MetHis simulations as all SNPs are considered statistically independent.

## Sampling simulated source and admixed populations

At the end of each simulation, we randomly drew individual samples from each source and the admixed population H matching observed sample sizes. We randomly sampled 60 individuals in the African source gamete reservoirs, 60 individuals in the European source, and the number of admixed individuals corresponding to the sample size of each Cabo Verde island of birth of individuals or to all 225 Cabo Verde-born individuals grouped in a single random mating population, in turn (*Figure 1—source data 1*). We sampled individuals without grand-parents in common, as allowed in MetHis by explicit genealogical flagging of individuals during the last two generations of the admixture process, hence mimicking our observed family unrelated dataset within Cabo Verde.

## Number of simulations and scenario-parameter priors for ABC inference

We performed 10,000 such *MetHis* simulations under each of four admixture scenarios for each of the nine birth-islands of Cabo Verde and for all Cabo Verde-born individuals grouped in a single population, separately. Each simulation was performed under a vector of scenario parameters drawn from prior distributions described above and in *Table 2*, using *MetHis parameter generator* tools. Separately for each island and for Cabo Verde as a single population, we used this set of simulations to determine which scenario best explained the observed data using Random-Forest ABC (see below). Under the winning scenario for each birth-island and separately for Cabo Verde as a whole, we then performed an additional 90,000 simulations each corresponding to a different vector of parameter values drawn randomly from the priors set by the user for this scenario, to produce a total of 100,000 simulations to be used for Neural-Network ABC posterior parameter estimation, for each ten – 9 islands and Cabo Verde as a whole, separately – analysis separately.

### *MetHis*-ABC Random Forest scenario-choice and Neural Network posterior parameter estimations

To reconstruct highly complex admixture histories in Cabo Verde using genetic data, we conducted machine-learning Approximate Bayesian Computation inferences based on the simulations produced with *MetHis* as described above under the four competing admixture history scenarios. We performed Random-Forest ABC scenario-choice (*Pudlo et al., 2016*), and Neural Network ABC posterior parameter inferences (*Csilléry et al., 2012*), for each Cabo Verde island and for all Cabo Verde-born individuals grouped in a single population, separately. We followed the *MetHis*-ABC approach proposed in Fortes-Lima et al., 2021 for summary-statistics calculation, prior-checking, out-of-bag cross-validation, machine-learning ABC predictions and inferences parameterization, and posterior parameter cross-validation error calculations.

All the details and results of these ABC procedures can be found in corresponding **Appendix 1**, *Appendix 1—table 1*, *Appendix 5—tables 1–10*, *Appendix 1—figures 3 and 4*, *Appendix 1—figure 3—Figure supplements 1–10*, and *Figure 7—figure supplements 1–3*. Briefly, we performed 10,000 simulations of 60,000 independent SNPs under each of the four competing scenarios described above, drawing parameter values in prior distributions detailed in *Table 2*. As listed in *Table 3* and described in details in **Appendix 1**, we then computed 42 summary-statistics separately for each simulated dataset comprising 60,000 independent SNPs, by drawing randomly 60 parents in the African and European source populations, and randomly drawing sample-sets matching the observed sample-sizes of each Cabo Verde birth-island or all Cabo Verde-born individuals as a single population, separately. Note that we considered summary-statistics specifically aiming at describing the distribution of individual admixture fractions in the sample set, known theoretically and empirically to be highly informative about complex admixture history parameters (Fortes-Lima et al., 2021; *Verdu and Rosenberg, 2011*). We thus performed RF-ABC scenario-choices using 40,000 vectors of 42 summary-statistics (10,000 under each four competing scenarios), each corresponding to a vector of parameter values randomly drawn in prior distributions and used for *MetHis* simulations, for each nine birth-island and for Cabo Verde as a whole, separately. We then performed NN-ABC joint posterior parameter inferences of all scenario-parameters under the winning scenarios obtained with RF-ABC, using 100,000 vectors of 42 summary statistics obtained from additional *MetHis* simulations under the winning scenarios respectively for each nine birth-island and for Cabo Verde as a whole, separately.

## Software availability

MetHis is an open source C software available with user manual on GitHub at https://github.com/romain-laurent/MetHis (Fortes-Lima et al., 2021).

## Acknowledgements

The authors would like to warmly thank all Cabo Verdean participants to this study as well as the UniCV for facilitating the project. This project was funded in part by the French "Agence Nationale

pour la Recherche" grant ANR METHIS 15-CE32-0009-1, and by a grant from the France-Stanford Center for Interdisciplinary Studies. MB was supported in part by the Linguistics Department at the University of Michigan (MI, USA). ZAS was supported in part by startup funds from the Department of Biology at the Pennsylvania State University (PA, USA), and by the NIH grant R35 GM146926. CAFL was supported in part by the Marcus Borgströms Foundation for Genetic Research and the Bertil Lundman Foundation for Anthropological Studies (Sweden). We thank the platform "Paléogénomique et génétique moléculaire" (P2GM) of the French Muséum National d'Histoire Naturelle at the Musée de l'Homme for support handling biological samples and generating genetic data. We thank the BIOMICS platform from the Pasteur institute for performing genotyping analyses. Finally, the authors warmly thank two independent reviewers and the editors of *eLife*, as well as Frank Alvarez-Pereyre, Erkan O Buzbas, Marta Ciccarella, Pierre Darlu, Evelyne Heyer, Ethan M Jewett, Marie-France Mifune, Etienne Patin, Jorge M Rocha, Lara Rubio Arauna, and Bruno Toupance, for useful comments and discussions about this work.

## Additional information

### Funding

| Funder | Grant reference number | Author |
| --- | --- | --- |
| Agence Nationale de la Recherche | ANR METHIS 15-CE32-0009-1 | Romain Laurent<br>Sergio S da Costa<br>Valentin Thouzeau<br>Cesar A Fortes-Lima<br>Françoise Dessarps-Freichey<br>José Utgé<br>Paul Verdu |
| France-Stanford Center for Interdisciplinary Studies | | Noah A Rosenberg |
| National Institutes of Health | R35 GM146926 | Zachary A Szpiech |
| Marcus Borgströms Foundation for Genetic Research | | Cesar A Fortes-Lima |
| Bertil Lundman Foundation for Anthropological Studies | | Cesar A Fortes-Lima |
| University of Michigan Linguistics Department Faculty Research Funds | | Marlyse Baptista |

The funders had no role in study design, data collection and interpretation, or the decision to submit the work for publication.

### Author contributions

Romain Laurent, Data curation, Software, Formal analysis, Validation, Investigation, Visualization, Methodology, Writing – original draft, Writing – review and editing, Conducted genetic data quality control and merging; Conducted linguistic data parsing; Conducted genetic and linguistic statistical descriptions; Conducted genetic inferences; Wrote the paper with the help of all authors; Zachary A Szpiech, Data curation, Software, Formal analysis, Validation, Investigation, Visualization, Methodology, Writing – original draft, Writing – review and editing, Conducted genetic and linguistic statistical descriptions; Wrote the paper with the help of all authors; Sergio S da Costa, Resources, Data curation, Investigation, Methodology, Writing – review and editing, Conducted sampling fieldwork; Conducted linguistic transcriptions; Conducted linguistic data categorization; Conducted genetic and linguistic statistical descriptions; Valentin Thouzeau, Resources, Data curation, Formal

analysis, Validation, Investigation, Methodology, Writing – review and editing, Conducted sampling field-work; Conducted genetic and linguistic statistical descriptions; Cesar A Fortes-Lima, Resources, Methodology, Writing – review and editing, Conducted sampling field-work; Françoise Dessarps-Freichey, Resources, Data curation, Validation, Investigation, Writing – review and editing, Conducted molecular-genetics experiments; Laure Lémée, Data curation, Validation, Investigation, Writing – review and editing, Conducted molecular-genetics experiments; José Utgé, Resources, Data curation, Validation, Investigation, Writing – review and editing, Conducted molecular-genetics experiments; Noah A Rosenberg, Conceptualization, Funding acquisition, Validation, Investigation, Writing – review and editing, Designed sampling strategy; Supervised the research; Marlyse Baptista, Conceptualization, Resources, Data curation, Funding acquisition, Validation, Investigation, Methodology, Writing – review and editing, Designed sampling strategy; Conducted sampling field-work; Conducted linguistic data categorization; Conducted genetic and linguistic statistical descriptions; Wrote the paper with the help of all authors; Supervised the research; Paul Verdu, Conceptualization, Resources, Data curation, Software, Formal analysis, Supervision, Funding acquisition, Validation, Investigation, Visualization, Methodology, Writing – original draft, Project administration, Writing – review and editing, Designed sampling strategy; Conducted sampling field-work; Conducted molecular-genetics experiments; Conducted genetic data quality control and merging; Conducted linguistic data categorization; Conducted linguistic data parsing; Conducted genetic and linguistic statistical descriptions; Conducted genetic inferences; Wrote the original draft with help from all authors; wrote the revised version with help from all authors; Supervised the research

### Author ORCIDs
Romain Laurent http://orcid.org/0000-0003-0363-2954
Zachary A Szpiech http://orcid.org/0000-0001-6372-8224
Cesar A Fortes-Lima http://orcid.org/0000-0002-9310-5009
Paul Verdu http://orcid.org/0000-0001-6828-268X

### Ethics
Research sampling protocols followed the Declaration of Helsinki guidelines and the French laws of scientific research deontology (Loi n° 2016-483 du 20 avril 2016). Research and ethics authorizations were provided by the Ministério da Saúde de Cabo Verde (228/DGS/11), Stanford University IRB (Protocol ID n°23194-IRB n°349), University of Michigan IRB (n°HUM00079335), and the French ethics committees and CNIL (Declaration n°1972648). All volunteer participants provided written and video-recorded informed consent.

### Decision letter and Author response
Decision letter https://doi.org/10.7554/eLife.79827.sa1
Author response https://doi.org/10.7554/eLife.79827.sa2

## Additional files

### Supplementary files
• MDAR checklist

### Data availability
The novel genome-wide genotype data, the linguistic utterance counts, and the self-reported anthropological data presented here can be accessed and downloaded via the European Genome-Phenome Archive (EGA) database accession numbers EGAD00001008976, EGAD00001008977, EGAD00001008978, and EGAD00001008979. All datasets can be shared provided that future envisioned studies comply with the informed consents provided by the participants, and in agreement with institutional ethics committee's recommendations applying to this data.

The following datasets were generated:

| Author(s) | Year | Dataset title | Dataset URL | Database and Identifier |
|---|---|---|---|---|
| Verdu P | 2022 | The admixture histories of Cabo Verde | https://ega-archive.org/studies/EGAD00001008977 | European Genome-Phenome Ar-chive (EGA), EGAD00001008977 |
| Verdu P | 2022 | The admixture histories of Cabo Verde | https://ega-archive.org/studies/EGAD00001008976 | European Genome-Phenome Archive (EGA), EGAD00001008976 |
| Verdu P | 2022 | The admixture histories of Cabo Verde | https://ega-archive.org/studies/EGAD00001008978 | European Genome-Phenome Archive (EGA), EGAD00001008978 |
| Verdu P | 2022 | The admixture histories of Cabo Verde | https://ega-archive.org/studies/EGAD00001008979 | European Genome-Phenome Archive (EGA), EGAD00001008979 |

The following previously published datasets were used:

| Author(s) | Year | Dataset title | Dataset URL | Database and Identifier |
|---|---|---|---|---|
| The 1000 Genomes Project Consortium | 2015 | 1000 Genomes Project Phase 3 | https://www.internationalgenome.org/data | International Genome Sample Resource (IGSR), IGSR |
| Gurdasani D | 2015 | African Genome Variation Project | https://ega-archive.org/studies/EGAS00001000959 | eGA, EGAD00001000959 |
| Patin E | 2017 | Dispersals and genetic adaptation of Bantu-speaking populations in Africa and North America. Science; 356(6337):543-6 | https://ega-archive.org/studies/EGAS00001002078 | eGA, EGAS00001002078 |

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

# Appendix 1

## 1a. Quality Control and genomic datasets merging procedures

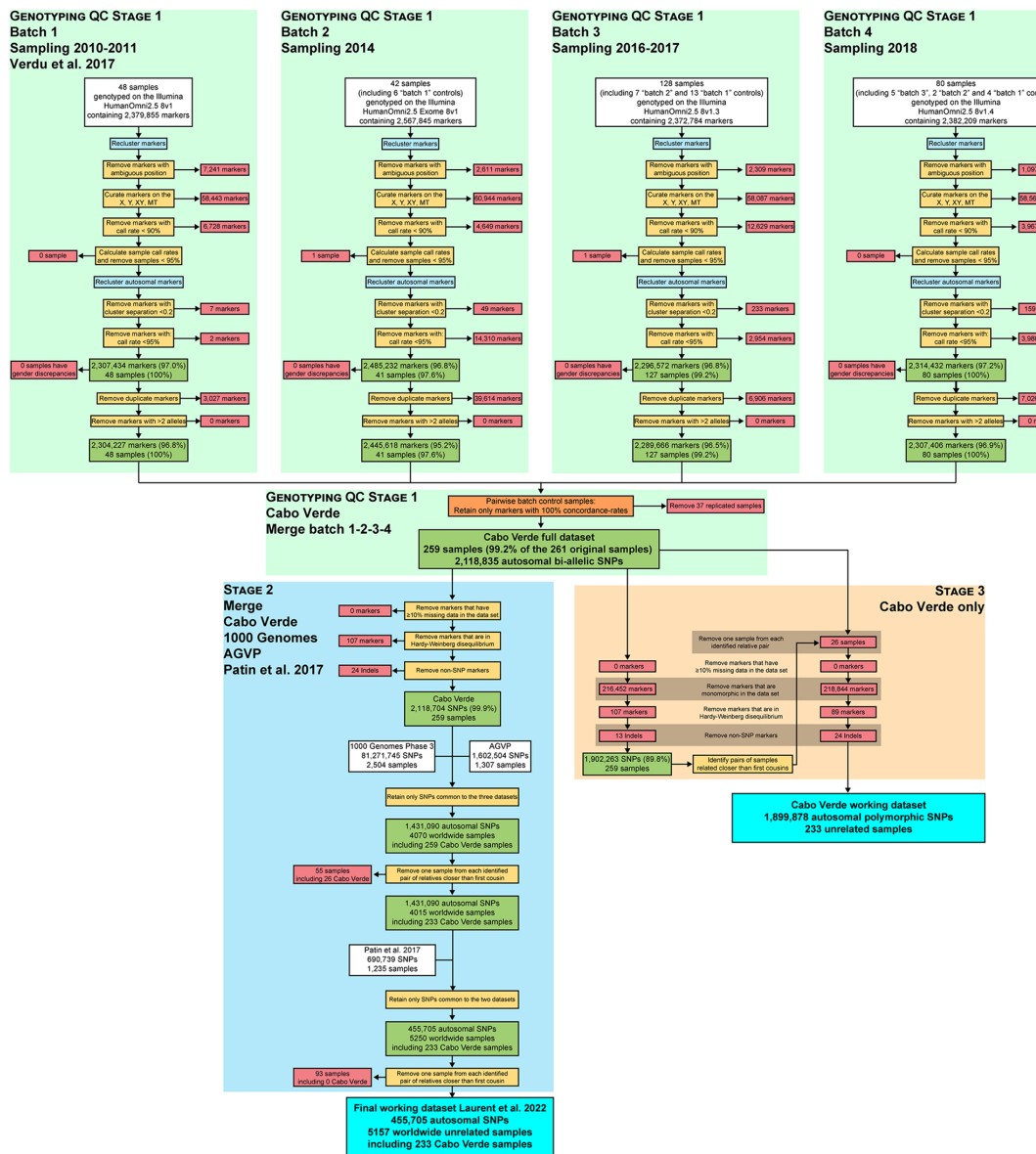

**Appendix 1—figure 1.** Quality-control and datasets merging procedures. Quality controls at the genotyping call level (Stage 1) were conducted with Illumina's GenomeStudio software Genotyping Module. Cabo Verde original DNA samples have been collected during six separate field-trips between 2010 and 2018, genotyped in four batches using four different versions of the Illumina Omni2.5Million Beadchip genotyping array. The resulting dataset was merged with 2,504 worldwide samples from the 1000 Genomes Project Phase 3 *Auton et al., 2015*; with 1,307 continental African samples from the African Genome Variation Project (EGA accession number EGAD00001000959, *Gurdasani et al., 2015*); and with 1,235 African samples from *Patin et al., 2017* (EGA accession number EGAS00001002078). We retained only autosomal SNPs common to all datasets, and excluded one individual for each pair of individuals related at the 2nd degree (at least one grand-parent in common) as inferred with KING (*Manichaikul et al., 2010*), following previous procedures (*Verdu et al., 2017*).

## 1b. Summary statistics for ABC inference

We used *MetHis* (Fortes-Lima et al., 2021) summary statistics calculation tools to calculate, for each simulated dataset, a vector of 42 summary statistics listed with references in main-text *Table 3*.

We computed the same 42 summary-statistics using the real data set for each Cabo Verdean island of birth and for all Cabo Verde-born individuals grouped in a single population, respectively. Henceforth, Population H refers to the admixed population simulated with *MetHis* corresponding, in turn, to each Cabo Verde island of birth and the 225 Cabo Verde-born individuals grouped in a single population.

## Within-population summary statistics

We computed the mean and variance of interindividual ASD (*Bowcock et al., 1994*) within Population H. Furthermore, we computed the mean and variance of average heterozygosities (where the average is taken over independent SNPs for each individual and then over individuals within an island) as in *Nei, 1978*. We also calculated the mean and variance of inbreeding coefficient F (*Danecek et al., 2011*). Intuitively, we a priori expected these statistics to be particularly sensitive to genetic-drift and possibly informative specifically about reproductive population size $N_e$ parameters in our scenarios.

## Admixture patterns summary statistics

We analytically showed previously that the distribution of admixture fractions across individuals within admixed populations carried identifiable information about the history of admixture (*Verdu and Rosenberg, 2011*; *Buzbas and Verdu, 2018*). Furthermore, we showed with *MetHis* (Fortes-Lima et al., 2021) that summary statistics describing this distribution could be successfully used in ABC inferences.

Therefore, we considered the 16 admixture-related summary statistics describing the mode, the first four moments (as mean, variance, skewness, and kurtosis), and the minimum, maximum, and all deciles of the ASD-MDS admixture estimates of individual admixture fractions from the African source population (or one minus that from the European source in a two-source admixture scenario). ASD-MDS admixture estimates were obtained as described in **Material and Methods**, represented schematically in *Appendix 1—figure 2*, considering the 2D-MDS centroids, respectively, of the African and European sources, and the projection on the line joining these two points of each Population H individual. Furthermore, to further capture the ASD-MDS admixture patterns, we calculated the distribution of the angles between population H's individuals and the source populations' centroids on the 2D ASD-MDS. We then considered as summary-statistics for ABC inference the mode, the first four moments (as mean, variance, skewness, and kurtosis), the minimum and maximum, and the deciles of this distribution of angles in radian (*Appendix 1—figure 2*).

## Among-populations summary statistics

Using *MetHis* summary-statistic calculation tools, we computed $F_{ST}$ values between either Source population (African or European) and Population H following *Weir and Cockerham, 1984*. We also computed the mean ASD between the Source Population and Population H. Finally, we computed the $f_3$ statistic (*Patterson et al., 2012*) with African and European populations as the two sources and Population H as the targeted admixed population.

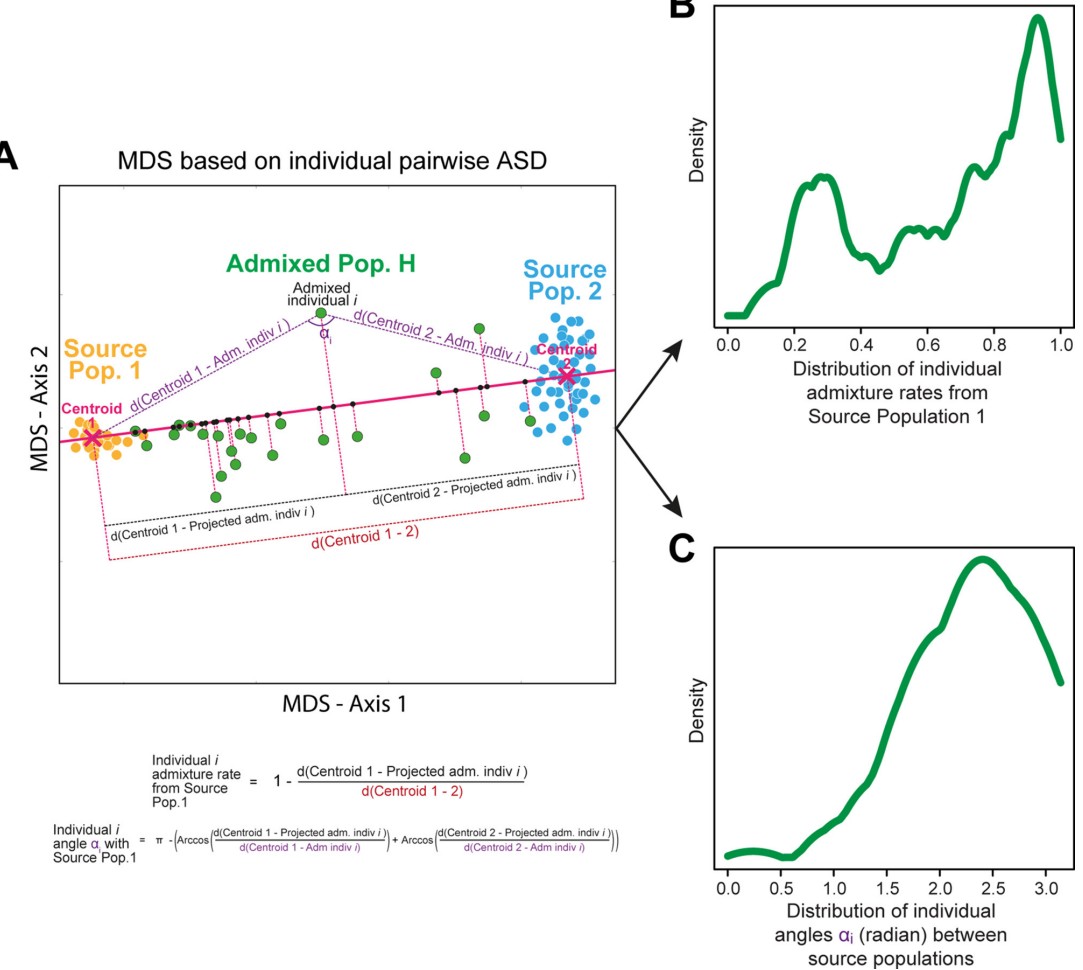

**Appendix 1—figure 2.** Schematic representation of the ASD-MDS estimates of individual admixture proportions and angles, also used as summary statistics for ABC inference and implemented in *MetHis* summary-statistic calculation tools. All the panels of the figure are schematic.

## 1c. Prior-ABC checking

Before conducting any ABC inference, we needed to statistically evaluate whether the 42 summary-statistics calculated on the observed datasets were within the range of the sets of summary statistics computed on the simulated datasets with *MetHis*. To do so, we considered each Cabo Verdean birth-island and Cabo Verde as a whole, in turn as population H, and conducted the following three-step procedure whose results are provided in *Appendix 1—figure 3*, *Appendix 1—figure 3—Figure supplements 1–10*.

1. first, we conducted a goodness-of-fit test between the 40,000 vectors of summary statistics computed for each simulation under the four competing scenarios, and the vector of observed summary statistics with the *gfit* function of the R package *abc* (*Csilléry et al., 2012*), with 1000 random repetitions and a 1% tolerance level (*Appendix 1—figure 3A*).

2. second, we computed a Principal Component Analysis considering each simulation as an individual and all the summary statistics as observed variables. By projecting the observed summary statistics onto this PCA along the first three PCA axes of variation (with the *princomp* function in R), we visually evaluated whether the observed data fell into the range of simulated statistics along the major PCA axes (*Appendix 1—figure 3B–C*).

3. finally, we present the distribution of each summary statistic obtained for the 40,000 simulations and the observed value, for each summary statistic and each Cabo Verdean birth-island and for Cabo Verde as a whole, separately (*Appendix 1—figure 3—Figure supplements 1–10*).

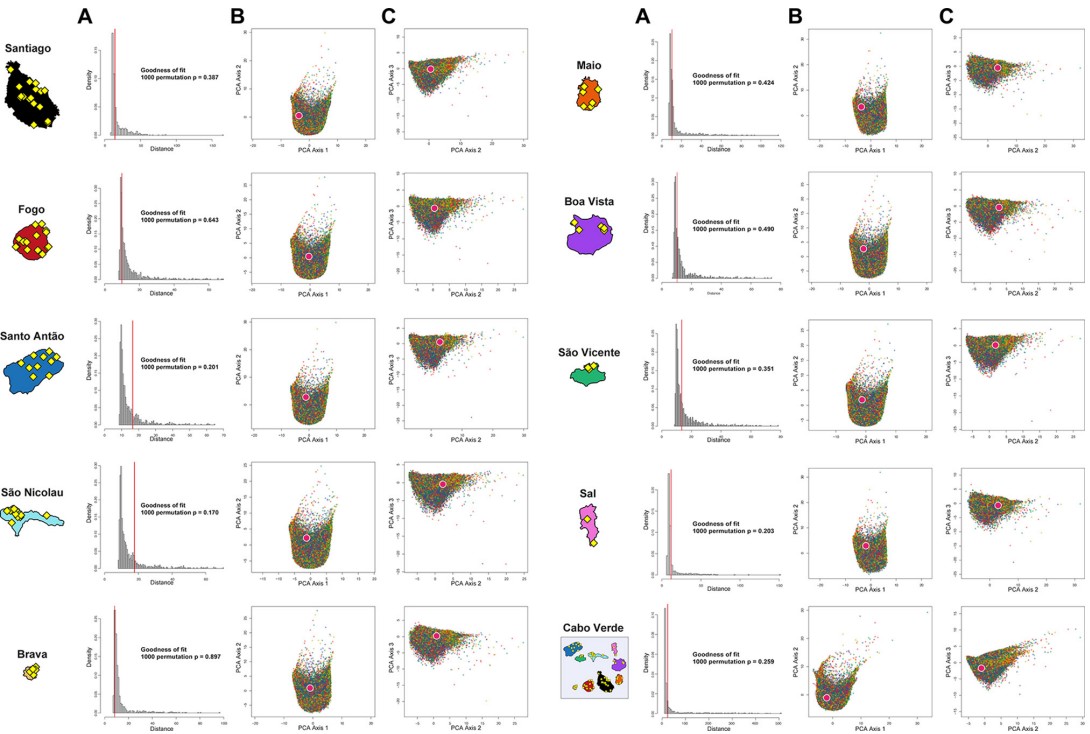

**Appendix 1—figure 3.** ABC Prior-checking for *MetHis* simulations for each island separately. 10,000 simulations are conducted under each four competing scenarios of historical admixture considered in Random-Forest ABC scenario-choice (see *Figure 6*). (**A**) Goodness-of-Fit tests: we use as goodness-of-fit statistic the median of the distance between one target vector of 42 summary-statistics and the vectors of 42 statistics obtained for the 1% simulations in the 40,000 simulations reference table that are closest to the target (as identified by simple rejection *Pritchard et al., 1999*). Results obtained with the observed data as target are indicated in the vertical red line. Null-distribution of the goodness-of-fit statistics as histograms are obtained by considering as target in-turn 1000 random simulations as pseudo-observed data, for each island and for the 225 Cabo Verde-born individuals grouped in a single random-mating population, separately. (**B**) First two axes of a principal component analysis performed on the 42 summary-statistics obtained for 40,000 simulations per island (10,000 simulations for each of four competing-scenarios). Each point corresponds to a single simulation colored per scenario: simulations under Scenario 1 are in blue; Scenario 2 in green; Scenario 3 in red; and Scenario 4 in yellow (*Figures 6–7*). The pink white-circled dot corresponds to the vector of summary-statistics from the observed dataset. (**C**) Axes 1 and 3 of the same PCA projection as in panel **B**.

The online version of this article includes the following figure supplement(s) for appendix 1—figure 3:

**Appendix 1—Figure 3 supplement 1.** Density distributions (black line) of each of 42 summary-statistics obtained from 40,000 simulations, 10,000 under each of four competing scenarios for Santiago, compared to the observed statistic obtained in this island (red vertical line).

**Appendix 1—Figure 3 supplement 2.** Density distributions (black line) of each of 42 summary-statistics obtained from 40,000 simulations, 10,000 under each of four competing scenarios for Fogo, compared to the observed statistic obtained in this island (red vertical line).

**Appendix 1—Figure 3 supplement 3.** Density distributions (black line) of each of 42 summary-statistics obtained from 40,000 simulations, 10,000 under each of four competing scenarios for Santo Antão, compared to the observed statistic obtained in this island (red vertical line).The four scenarios are synthetically described in *Figure 6* and the 42 summary-statistics in *Table 3*.

**Appendix 1—Figure 3 supplement 4.** Density distributions (black line) of each of 42 summary-statistics obtained from 40,000 simulations, 10,000 under each of four competing scenarios for São Nicolau, compared to the observed statistic obtained in this island (red vertical line).

**Appendix 1—Figure 3 supplement 5.** Density distributions (black line) of each of 42 summary-statistics obtained from 40,000 simulations, 10,000 under each of four competing scenarios for Brava, compared to the observed statistic obtained in this island (red vertical line).

**Appendix 1—Figure 3 supplement 6.** Density distributions (black line) of each of 42 summary-statistics obtained from 40,000 simulations, 10,000 under each of four competing scenarios for Maio, compared to the observed statistic obtained in this island (red vertical line).

**Appendix 1—Figure 3 supplement 7.** Density distributions (black line) of each of 42 summary-statistics obtained from 40,000 simulations, 10,000 under each of four competing scenarios for Boa Vista, compared to the observed statistic obtained in this island (red vertical line).

**Appendix 1—Figure 3 supplement 8.** Density distributions (black line) of each of 42 summary-statistics obtained from 40,000 simulations, 10,000 under each of four competing scenarios for São Vicente, compared to the observed statistic obtained in this island (red vertical line).

**Appendix 1—Figure 3 supplement 9.** Density distributions (black line) of each of 42 summary-statistics obtained from 40,000 simulations, 10,000 under each of four competing scenarios for Sal, compared to the observed statistic obtained in this island (red vertical line).

**Appendix 1—Figure 3 supplement 10.** Density distributions (black line) of each of 42 summary-statistics obtained from 40,000 simulations, 10,000 under each of four competing scenarios for the 225 Cabo Verde-born individuals grouped in a single random-mating population, compared to the observed statistic obtained in this dataset (red vertical line).

The four scenarios are synthetically described in *Figure 6* and the 42 summary-statistics in *Table 3*.

## 1d. Random-Forest (RF) ABC scenario-choice cross-validation

We used the Random-Forest ABC algorithm implemented in the *abcrf* package in R (*Pudlo et al., 2016*; *Raynal et al., 2019*) for scenario-choice, as in the *MetHis*-ABC pipeline previously described (Fortes-Lima et al., 2021). Random-Forest classification has been shown to allow for robust ABC scenario-choice with relatively small numbers of simulations and, most importantly, to be unaffected by correlations among summary statistics; thus outperforming local linear regression classically used in ABC scenario-choice (*Beaumont et al., 2002*). Here, we considered the same prior probability (25%) for the four competing scenarios.

First, for each island separately, we used the *abcrf* function in this R package to conduct ABC scenario-choice cross-validation. Each of 40,000 simulations served in-turn as pseudo-observed data and the remaining 39,999 simulations as training (*Appendix 1—figure 4A*), using 1000 decision trees in the Random-Forest. We visually checked that prior error rates (the rates of erroneously assigned scenario in the cross-validation) were minimized for this number of decision trees, using the *err.abcrf* function in the same R package. Each summary-statistic's importance to the cross-validation accurate decision was calculated and plotted using the *abcrf* function (*Appendix 1—figure 4B*).

Second, for each birth-island and for Cabo Verde as a whole, separately, we used the *predict.abcrf* function on the trained Random Forest with all simulations in the reference table, to determine which competing scenario produced simulations whose summary statistics most resembled those from the observed data. We then estimate the posterior probability of accurately finding such winning scenario in our framework (indicated as "Post. prob. Scen." in *Figure 7B*).

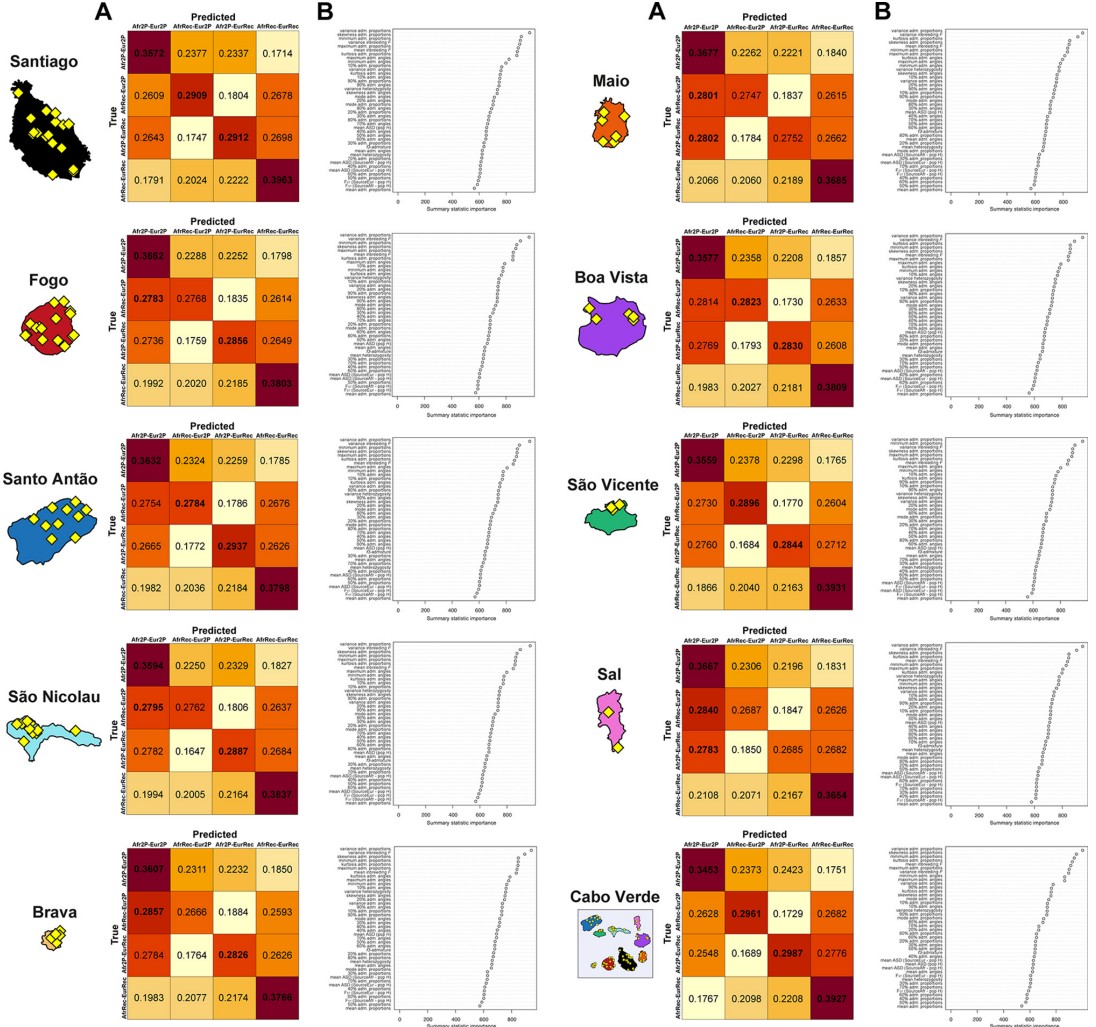

**Appendix 1—figure 4.** Random-Forest ABC scenario-choice cross validation results and summary-statistics' importance. We considered four competing scenarios as described in Genetic admixture histories in Cabo Verde inferred with MetHis-ABC and in *Figure 6*, with associated prior parameter distributions in *Table 2* and 42 summary-statistics described in *Table 3*, for each Cabo Verdean birth-island and the 225 Cabo Verde-born individuals grouped in a single random mating population, separately. Cross-validation results are obtained by conducting RF scenario-choices using, in-turn, each 40,000 simulations (10,000 per competing scenario) as pseudo-observed data and the remaining 39,999 simulations as the reference table. We considered 1,000 trees in the random-forest for each analysis. Cross-validation results and associated summary-statistics' importance for the RF decisions are obtained from the function *abcrf* in the R package *abcrf*.

## 1e. Neural Network ABC posterior parameter inference procedures

While RF-ABC scenario-choice procedures are specifically designed to accurately work with a relatively small number of simulations compared to classical regression ABC (*Beaumont et al., 2002*; *Pudlo et al., 2016*), posterior joint-estimations of all scenario-parameters under the winning scenario remain difficult in Random-Forest (Fortes-Lima et al., 2021; *Raynal et al., 2019*). In this context, our previous benchmarking of various ABC posterior-parameter estimation under the *MetHis* framework showed that Neural Network ABC joint parameter estimations outperformed other approaches (Fortes-Lima et al., 2021). However, Neural Network ABC posterior parameter inferences have been shown in multiple previous investigations, including ours with *MetHis*, to require substantial amounts of information in the reference table to perform satisfactorily (Fortes-Lima et al., 2021; *Csilléry et al., 2012*; *Jay et al., 2019*). Therefore, we conducted anew 90,000 simulations with *MetHis*, for each birth-island and for Cabo Verde as a whole, separately, under the winning scenario determined with RF-ABC, thus obtaining a total 100,000 simulations under the winning scenario, for each birth-

island or for Cabo Verde as a whole, separately, for further Neural Network ABC posterior parameter estimation using the *abc* package in R (*Csilléry et al., 2012*).

## NN-ABC parameter inference settings

We followed the same empirical approach as in *MetHis*-ABC (Fortes-Lima et al., 2021) to determine the most suitable number of neurons in the hidden layer and associated number of simulations closest to the target data and retained for training the Neural Network (called "tolerance level") for further NN-ABC posterior parameter estimation for this target. Indeed, there is no rule-of-thumb to determine these parameters a priori in order to obtain the most informative posterior parameter inference while minimizing overfitting using a Neural-Network approach under a given scenario (*Csilléry et al., 2012*; *Jay et al., 2019*).

Therefore, we empirically tested three different tolerance-level values to be used in NN-ABC posterior parameter estimation: 1%, 5%, or 10% of the total 100,000 simulations generated under the winning scenario determined by our RF-ABC procedure for each Cabo Verdean island and for Cabo Verde as a whole, respectively. Furthermore, for each one of the three tolerance-level values, we tested associated numbers of neurons in the hidden layer between 5 and 11, or 12, the number of free parameters minus one in the winning scenario for each Cabo Verdean island and for Cabo Verde as a whole, respectively (*Figures 6–7*).

Then, for each pair of tolerance-level value and number of neurons for each targeted data separately, we performed 1,000 cross-validation NN-ABC posterior parameter inferences using the *cv4abc* function in the *abc* package (*Csilléry et al., 2012*). This procedure draws 1,000 simulations at random and considers them, in turn, as pseudo-observed data for posterior-parameter inference using the remaining 99,999 simulations as the reference table. For the 1,000 pseudo-observed cross-validations and each pair of tolerance level and number of hidden neurons, we calculated the error between the median posterior point-estimate of each parameter ($\hat{\theta}_i$) and its' known true-value used for simulation ($\theta_i$) , as the mean-square error (MSE) scaled by the parameter's variance across the 1,000 cross-validations (*Csilléry et al., 2012*):

$$Scaled\ MSE\ \left( \hat{\theta}_i \right) = \sum_1^{1000} \left( \hat{\theta}_i - \theta_i \right)^2 / \left( 1000 \times Variance\left( \theta_i \right) \right).$$

Therefore, we conducted, in total, 219,000 NN-ABC cross-validation parameter inference procedures to identify the NN-ABC settings most suitable for further posterior parameter estimations using the observed data for each Cabo Verdean birth-island and for Cabo Verde as a whole. For each Cabo Verdean island and for Cabo Verde as a whole, separately under the corresponding winning scenario, we chose the pair of tolerance-level values and number of hidden neurons that minimized the average error across all estimated parameters, as the Neural-Network setting used in further ABC posterior parameter inferences based on the observed data (*Appendix 1—table 1*).

## NN-ABC posterior parameter estimation

For each Cabo Verdean island and for all Cabo Verde-born individuals grouped in a single population, separately, under the winning scenario identified with RF-ABC, we jointly estimated the posterior distributions of all parameters using NN-ABC "*neuralnet*" method option in the function *abc* of the R package *abc* (*Csilléry et al., 2012*), with logit-transformed parameters ("*logit*" transformation option) using the tolerance level and number of neurons in the hidden layer identified previously in *Appendix 1—table 1*. For each birth-island and for Cabo Verde as a whole, separately, a synthetic schematic figure of the complex admixture processes identified using median point estimates of the posterior scenario-parameter distribution is provided in *Figure 7*, and full posterior parameter distributions with 95% Credibility-Intervals (CI) are provided in *Figure 7—figure supplements 1–3* and *Appendix 5—Tables 1–10*.

## NN-ABC posterior parameter errors

We evaluated posterior parameter error rates and 95% CI accuracies in the vicinity of the observed data, for each island and for Cabo Verde as a whole, separately, and for each corresponding NN-ABC posterior parameter joint estimation.

**Appendix 1—table 1.** Average parameter posterior random cross-validation error across all model parameters as a function of the number of neurons in the hidden layer and the rejection tolerance level (number of simulations retained for training the Neural-Network) under the winning scenario for each island respectively. For each island separately, and for the 225 Cabo Verde-born individuals grouped as a single "Cabo Verde" population, we considered, 1,000 random simulations in-turn as pseudo-observed data to estimate posterior parameter distributions and 100,000 total simulations in the reference table. For each cross-validation procedure, we considered between 5 and 11 neurons in the hidden layer ("NN-HL") for the winning scenarios with 12 original parameters, and between 5 and 12 NN-HL for the winning scenarios with 13 original parameters. We considered three different tolerance levels of 0.01, 0.03, and 0.1 ("Tol.") corresponding, respectively, to 1,000, 3,000, and 10,000 simulations, in turn closest to each one of the 1,000 cross-validation pseudo-observed simulation retained for training the NN. The median values of posterior parameter distributions were used as point estimates for the calculation of the error of each parameter (*Csilléry et al., 2012*). For each birth-island and Cabo Verde as a whole, and corresponding winning scenario, separately, we considered, for further posterior parameter estimation using the observed "real" data, only the pair of tolerance level and number of hidden neurons that minimized the average error on posterior parameter estimations across all parameters (indicated in bold in the table).

| Individual birth-island | | SANTIAGO | FOGO | SANTO ANTAO | SAO NICOLAU | BRAVA | MAIO | BOA VISTA | SAO VICENTE | SAL | CABO VERDE |
|---|---|---|---|---|---|---|---|---|---|---|---|
| Winning scenario | | Scenario 1: Afr2pulses-Eur2pulses | Scenario 1: Afr2pulses-Eur2pulses | Scenario 1: Afr2pulses-Eur2pulses | Scenario 1: Afr2pulses-Eur2pulses | Scenario 1: Afr2pulses-Eur2pulses | Scenario 1: Afr2pulses-Eur2pulses | Scenario 2: Afr2pulses-EurReccurring | Scenario 3: AfrReccurring-Eur2pulses | Scenario 2: Afr2pulses-EurReccurring | Scenario 1: Afr2pulses-Eur2pulses |
| Number of scenario parameters | | 12 | 12 | 12 | 12 | 12 | 12 | 13 | 13 | 13 | 12 |
| NN-HL=5 | Tol.=0.01 | 0.79264 | 0.79943 | 0.80170 | 0.79380 | 0.81251 | 0.83232 | 0.79972 | 0.77795 | 0.82888 | **0.79532** |
| NN-HL=5 | Tol.=0.03 | 0.80866 | 0.81252 | 0.81257 | 0.80711 | 0.81703 | 0.83470 | 0.79708 | 0.80208 | 0.81783 | 0.83438 |
| NN-HL=5 | Tol.=0.1 | 0.84002 | 0.84001 | 0.84662 | 0.84970 | 0.84526 | 0.85366 | 0.83374 | 0.82312 | 0.83264 | 0.85271 |
| NN-HL=6 | Tol.=0.01 | **0.77139** | 0.80186 | 0.79800 | 0.80765 | 0.81085 | 0.82516 | 0.78239 | 0.77743 | 0.81601 | 0.82118 |
| NN-HL=6 | Tol.=0.03 | 0.79501 | 0.81397 | 0.80533 | 0.80483 | 0.83318 | 0.83673 | 0.80492 | 0.80127 | 0.81325 | 0.82982 |
| NN-HL=6 | Tol.=0.1 | 0.84684 | 0.84072 | 0.84028 | 0.84484 | 0.85753 | 0.85428 | 0.82136 | 0.81617 | 0.83773 | 0.84745 |
| NN-HL=7 | Tol.=0.01 | 0.78525 | 0.79836 | 0.79075 | 0.81595 | 0.82657 | 0.82662 | 0.79304 | 0.77898 | 0.80415 | 0.80413 |
| NN-HL=7 | Tol.=0.03 | 0.80899 | 0.81151 | 0.81000 | 0.81035 | 0.83069 | 0.83474 | 0.80060 | 0.79491 | 0.81855 | 0.83622 |
| NN-HL=7 | Tol.=0.1 | 0.85600 | 0.84844 | 0.83704 | 0.84737 | 0.85071 | 0.85582 | 0.81083 | 0.81619 | 0.83779 | 0.85601 |
| NN-HL=8 | Tol.=0.01 | 0.78512 | 0.79805 | 0.79127 | 0.80886 | 0.82241 | 0.83397 | **0.77998** | 0.78077 | 0.80992 | 0.80783 |
| NN-HL=8 | Tol.=0.03 | 0.80595 | 0.81077 | 0.81353 | 0.82455 | 0.82334 | 0.83718 | 0.79490 | 0.78639 | 0.80666 | 0.83396 |
| NN-HL=8 | Tol.=0.1 | 0.83938 | 0.84904 | 0.83681 | 0.85231 | 0.84982 | 0.84614 | 0.83499 | 0.81671 | 0.82696 | 0.84774 |
| NN-HL=9 | Tol.=0.01 | 0.77910 | 0.79899 | 0.81027 | 0.80103 | 0.81898 | 0.82483 | 0.79744 | 0.78619 | 0.81770 | 0.80542 |
| NN-HL=9 | Tol.=0.03 | 0.81192 | 0.81046 | 0.81474 | 0.81642 | 0.82593 | 0.82631 | 0.80830 | 0.78640 | 0.82212 | 0.84165 |
| NN-HL=9 | Tol.=0.1 | 0.84379 | 0.84721 | 0.83676 | 0.84813 | 0.85218 | 0.85223 | 0.83254 | 0.81696 | 0.81723 | 0.85381 |
| NN-HL=10 | Tol.=0.01 | 0.78593 | **0.79615** | 0.80275 | 0.79444 | **0.80692** | **0.82095** | 0.80121 | 0.77644 | 0.81801 | 0.80485 |
| NN-HL=10 | Tol.=0.03 | 0.80159 | 0.80818 | 0.81143 | 0.80949 | 0.82496 | 0.82787 | 0.79994 | 0.78673 | 0.81565 | 0.82157 |
| NN-HL=10 | Tol.=0.1 | 0.85245 | 0.84242 | 0.84222 | 0.83444 | 0.85410 | 0.85424 | 0.83049 | 0.81742 | 0.84950 | 0.84183 |
| NN-HL=11 | Tol.=0.01 | 0.79994 | 0.81073 | **0.78479** | **0.78709** | 0.82527 | 0.84091 | 0.79414 | **0.77365** | **0.79984** | 0.80239 |
| NN-HL=11 | Tol.=0.03 | 0.82611 | 0.80802 | 0.80825 | 0.81653 | 0.82511 | 0.82334 | 0.80234 | 0.80333 | 0.81750 | 0.82959 |
| NN-HL=11 | Tol.=0.1 | 0.84914 | 0.83930 | 0.84011 | 0.84941 | 0.85471 | 0.85756 | 0.82120 | 0.81919 | 0.83044 | 0.85282 |
| NN-HL=12 | Tol.=0.01 | na | na | na | na | na | na | 0.79915 | 0.78138 | 0.81418 | na |

*Appendix 1—table 1 continued on next page*

*Appendix 1—table 1 continued*

| Individual birth-island | | SANTIAGO | FOGO | SANTO ANTAO | SAO NICOLAU | BRAVA | MAIO | BOA VISTA | SAO VICENTE | SAL | CABO VERDE |
|---|---|---|---|---|---|---|---|---|---|---|---|
| Winning scenario | | Scenario 1: Afr2pulses-Eur2pulses | Scenario 1: Afr2pulses-Eur2pulses | Scenario 1: Afr2pulses-Eur2pulses | Scenario 1: Afr2pulses-Eur2pulses | Scenario 1: Afr2pulses-Eur2pulses | Scenario 1: Afr2pulses-Eur2pulses | Scenario 2: Afr2pulses-EurReccurring | Scenario 3: AfrReccurring-Eur2pulses | Scenario 2: Afr2pulses-EurReccurring | Scenario 1: Afr2pulses-Eur2pulses |
| Number of scenario parameters | | 12 | 12 | 12 | 12 | 12 | 12 | 13 | 13 | 13 | 12 |
| NN-HL=12 | Tol.=0.03 | na | na | na | na | na | na | 0.79959 | 0.79306 | 0.81551 | na |
| NN-HL=12 | Tol.=0.1 | na | na | na | na | na | na | 0.81731 | 0.82090 | 0.82284 | na |

First, we identified the 1,000 simulations closest to the observed "real" data for each target data separately using a 1% tolerance level in the "*neuralnet*" option from the *abc* function. We then used each one of these 1,000 specific simulations, in turn, as pseudo-observed target data for cross-validation NN-ABC posterior parameter estimation using the same NN settings as previously, logit-transformed parameters, and the 99,999 remaining simulations in the reference table. We then compared the median posterior estimate of each scenario parameter $\left(\hat{\theta}_i\right)$, with the original parameter value used for the simulation $(\theta_i)$, respectively for each 1,000 cross-validation posterior parameter estimation; and calculated the cross-validation mean absolute error (MAE), for each scenario parameter (***Appendix 5—Tables 1–10***):

$$MAE\left(\hat{\theta}_i\right) = \sum_1^{1000} \left|\hat{\theta}_i - \theta_i\right| / 1000.$$

Second, for each island and for Cabo Verde as a whole, and for each scenario-parameter separately, we calculated how many times the true parameter values $(\theta_i)$, used for simulating the 1,000 simulations closest to the observed data, fell within the 95% CI [2.5% quantile $\left(\hat{\theta}_i\right)$; 97.5% quantile $\left(\hat{\theta}_i\right)$] estimated using the observed data. Thus, if the 1,000 cross-validation true parameter values fell more than 95% of the time within the estimated 95% CI, the length of the estimated 95% CI was considered over-estimated and thus excessively conservative; and, alternatively, if it was the case less than 95% of the time, the length of the 95% CI was considered under-estimated thus indicating a less conservative behavior of our CI estimation procedure (***Appendix 5—Tables 1–10***).

# Appendix 2

## Genetic diversity patterns with ASD-MDS at different geographical scales

At the worldwide scale (*Appendix 2—figure 1*), the first MDS axis is mainly driven by genetic differentiation between African individuals and non-African individuals, the second MDS axis by differentiation between European and East Asian individuals, and the third axis by differentiation between South Asian and American individuals.

We explored successive subsamples of this worldwide dataset in order to decompose apparent preferred relationships between each admixed population related to the TAST with respect to African and European populations only. We find (*Appendix 2—figure 2*) that the first MDS axis differentiates African and European individuals; the second axis differentiates mainly Senegambian West Western African individuals from Congo Basin hunter-gatherer individuals (namely the Baka, Ba.Bongo, Ba.Koya, Ba.Mbuti, Ba.Twa and Bezan); and the third axis differentiates further these latter populations from Eastern African individuals. In this context, we find Cabo Verdean individuals clustering along an axis going from the European cluster to a cluster formed mainly by West Western African individuals, with the sole exception of two individuals with fathers born on Angola as per our familial anthropology questionnaires and consistently clustering closer to our Angolan samples. Instead, ASW, ACB and PUR individuals are found on a slightly different trajectory going from European individuals towards East Western Africa mainly (*Appendix 2—figure 2*).

Further resampling our ASD matrix (*Appendix 2—figure 3*), we find no signs of particular genetic affinity, along the first three MDS axes, between enslaved-African descendants and Eastern and Southern African individuals.

Notably, we find Cabo Verdean individuals clustering similarly to Senegambian Fulani individuals in between European and other Senegambian populations, the Fulani being known to have historically received gene-flow from Northern African populations (*Busby et al., 2016*; *Gurdasani et al., 2015*; *Pereira et al., 2010*; *Henn et al., 2012*). In the absence of Northern African populations in our dataset, the South-Western European Iberian-IBS population represented the best such proxy-populations with known historical relationship with Northern Africa, hence resulting, at this geographical scale, in the similar clustering of Cabo Verdean and Fulani individuals. Note that Fulani and Cabo Verdean individuals substantially departed from each other along higher MDS axes, the Cabo Verdeans remaining on a European-African axis while the Fulani departed from this trajectory on the European side (*Appendix 2—figure 3*).

Finally, considering only populations related to the TAST with respect to European and East, Central, and West Western African populations (excluding the Fulani), we find (*Appendix 2—figure 4*) that Cabo Verdeans cluster separately from other TAST populations not only with respect to African populations, but also with respect to their relationships to European populations (*Figure 2*).

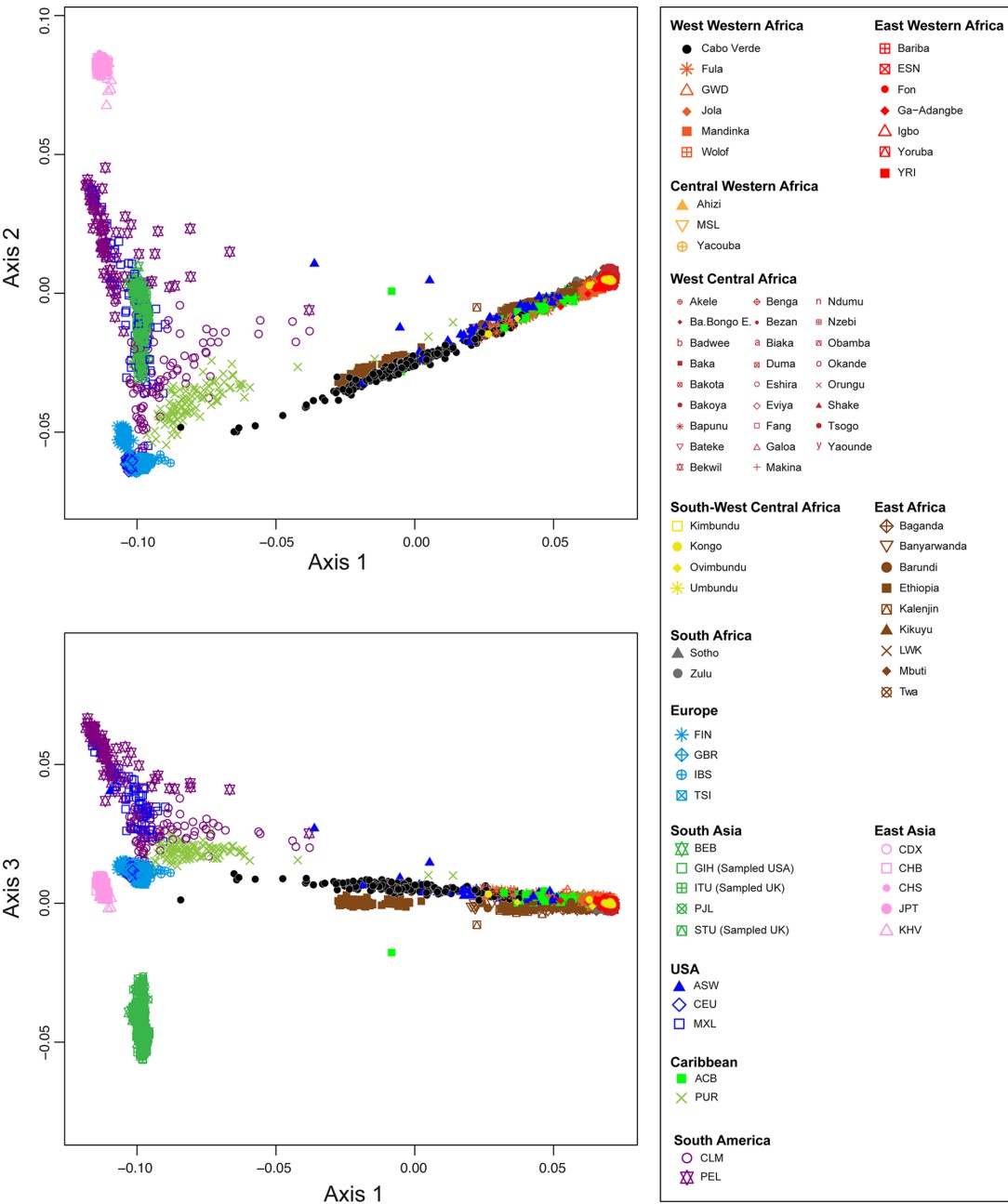

**Appendix 2—figure 1.** Multidimensional scaling three-dimensional projection of allele sharing pairwise dissimilarities, for all worldwide populations in our dataset. Each individual in the plot is represented by a single point. See *Figure 1—source data 1* for the population list used and *Figure 1* for sample locations and symbols. (**A**) Axis 1 and 2; (**B**) Axis 1 and 3. 3D animated plot is provided in.*gif* format. We evaluate the Spearman correlation between 3D MDS projections and the original ASD pairwise matrix to evaluate the precision of the dimensionality reduction along the first three axes of the MDS. We find significant Spearman $\rho$ of 0.9796 ($p<2.2 \times 10^{-16}$).

The online version of this article includes the following video for appendix 2—figure 1:

**Appendix 2—figure 1—animation 1.** 3D animated MDS of pairwise allele sharing dissimilarities for all worldwide populations in our dataset.

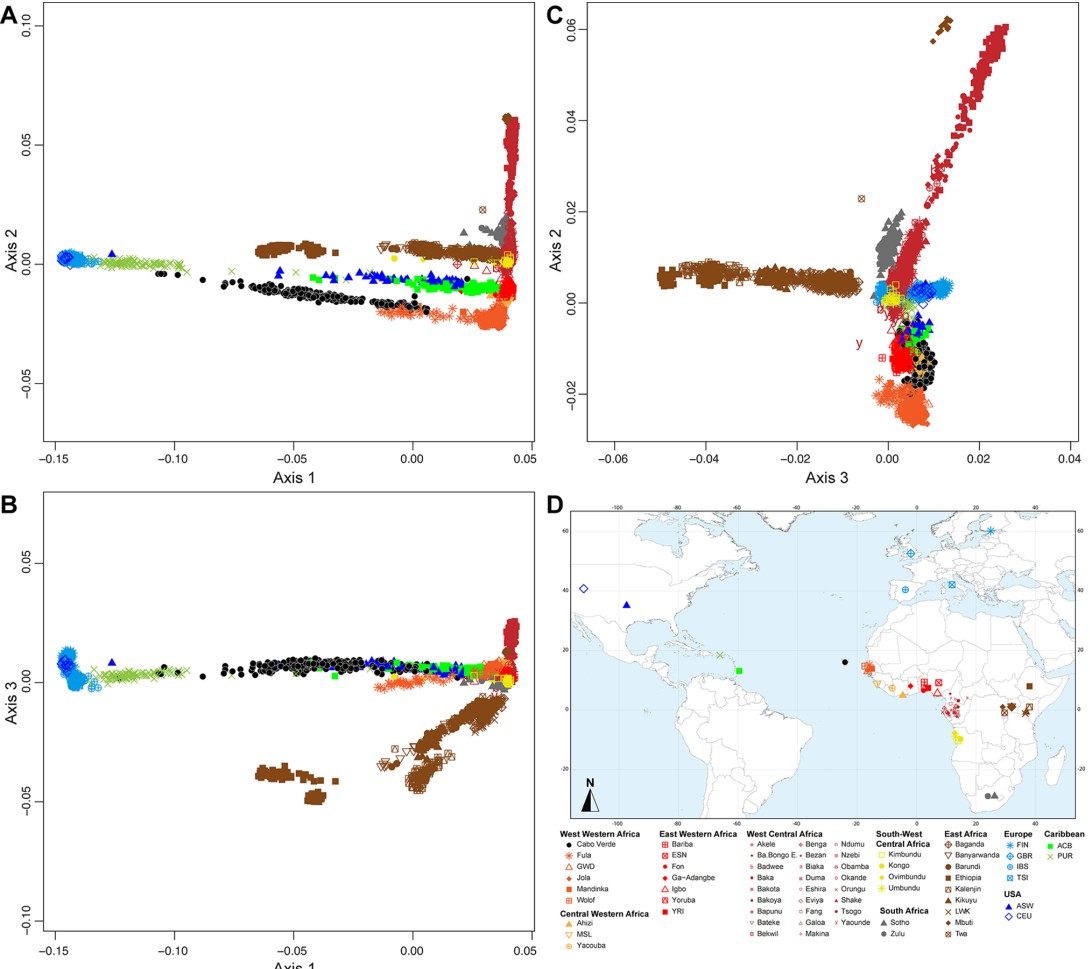

**Appendix 2—figure 2.** Multidimensional scaling three-dimensional projection of allele sharing pairwise dissimilarities, for all of the African, European and other admixed populations related to the TAST. Each individual is represented by a single point. We removed all Asian and South American samples compared to the sample set employed in *Appendix 2—figure 1*, and recomputed the MDS based on this reduced sample set. See *Figure 1—source data 1* for the population list used in these analyses. The sample map used in these analyses is extracted from *Figure 1* and provided in panel D. (**A**) Axis 1 and 2; (**B**) Axis 2 and 3; (**C**) Axis 1 and 3. 3D animated plot is provided in.*gif* format. We evaluate the Spearman correlation between 3D MDS projections and the original ASD pairwise matrix to evaluate the precision of the dimensionality reduction along the first three axes of the MDS. We find Spearman $\rho$ of 0.9392 ($p < 2.2 \times 10^{-16}$).

The online version of this article includes the following video for appendix 2—figure 2:

**Appendix 2—figure 2—animation 1.** 3D animated MDS of pairwise allele sharing dissimilarities for all of the African, European and other admixed populations related to the TAST.

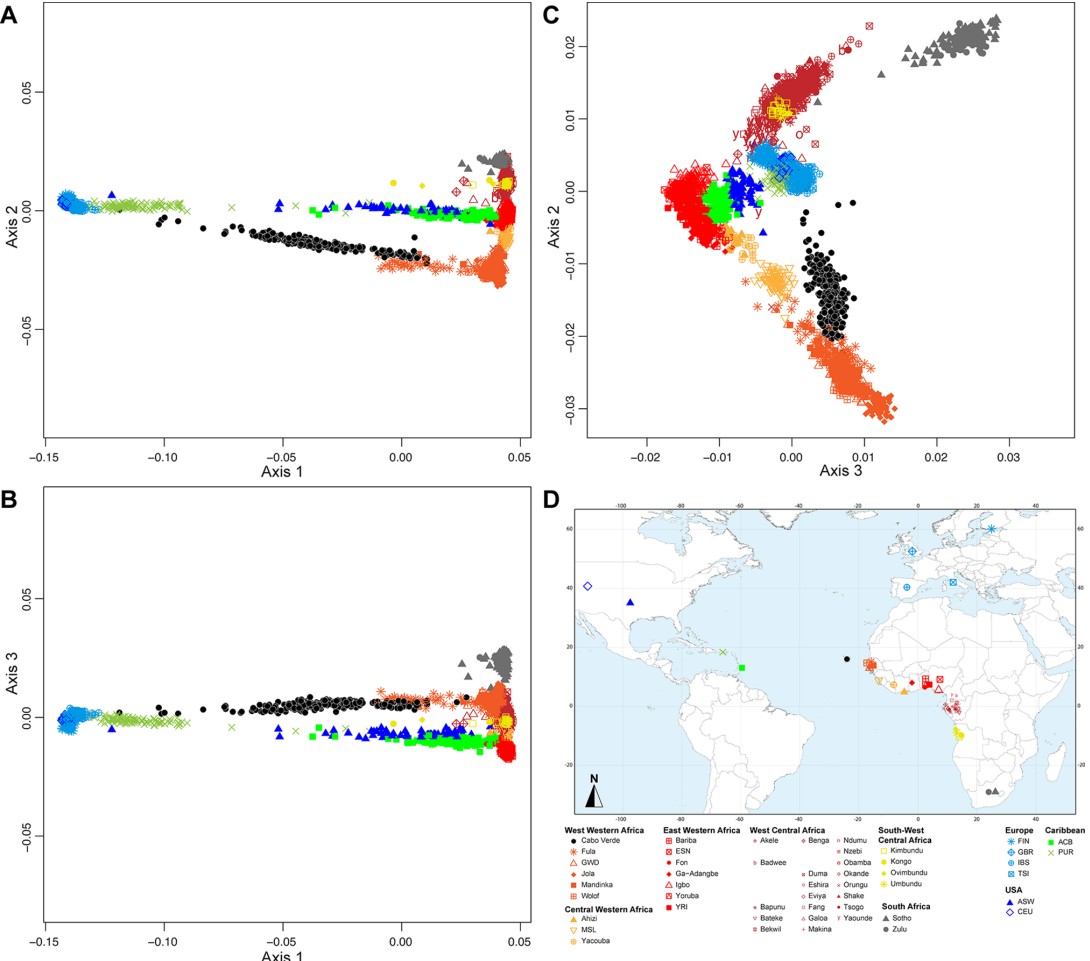

**Appendix 2—figure 3.** Multidimensional scaling three-dimensional projection of allele sharing pairwise dissimilarities, for a subset of African, European and other admixed populations related to the TAST. Each individual is represented by a single point. We removed all East African and Central African hunter-gatherer samples (Baka, Bezan, Ba.Bongo, Ba.Koya, Ba.Twa, Bi.Aka, Mbuti) compared to the sample set employed in *Appendix 2—figure 2*, and recomputed the MDS based on this reduced sample set. See *Figure 1—source data 1* for the population list used in these analyses. Sample map used in these analyses is extracted from *Figure 1* and provided in panel D. (**A**) Axis 1 and 2; (**B**) Axis 2 and 3; (**C**) Axis 1 and 3. 3D animated plot is provided in.*gif* format. We evaluate the Spearman correlation between 3D MDS projections and the original ASD pairwise matrix to evaluate the precision of the dimensionality reduction along the first three axes of the MDS. We find Spearman $\rho$ of 0.9450 ($p < 2.2 \times 10^{-16}$).

The online version of this article includes the following video for appendix 2—figure 3:

**Appendix 2—figure 3—animation 1.** 3D animated MDS of pairwise allele sharing dissimilarities for a subset of African, European and other admixed populations related to the TAST.

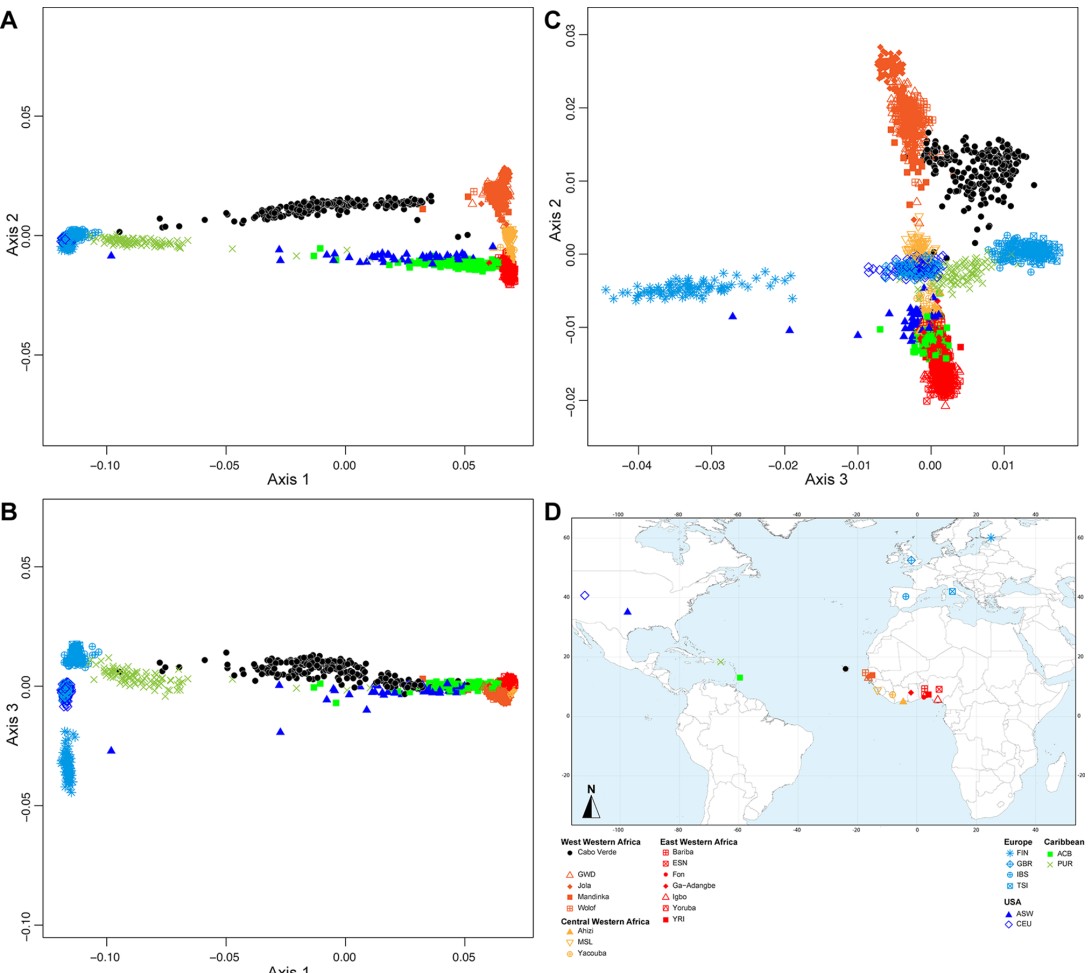

**Appendix 2—figure 4.** Multidimensional scaling three-dimensional projection of allele sharing pairwise dissimilarities, for the closest subsets of West African, European and other admixed populations related to the TAST. Each individual is represented by a single point. We removed all West and South-West Central African samples, as well as South African samples, compared to the sample set employed in *Appendix 2—figure 3*, and recomputed the MDS based on this reduced sample set. See *Figure 1—source data 1* for the population list used in these analyses. Sample map used in these analyses is extracted from *Figure 1* and provided in panel **D**. (**A**) Axis 1 and 2; (**B**) Axis 2 and 3; (**C**) Axis 1 and 3. 3D animated plot is provided in.*gif* format. We evaluate the Spearman correlation between 3D MDS projections and the original ASD pairwise matrix to evaluate the precision of the dimensionality reduction along the first three axes of the MDS. We find Spearman $\rho$ of 0.9636 ($p<2.2 \times 10^{-16}$).

The online version of this article includes the following video for appendix 2—figure 4:

**Appendix 2—figure 4—animation 1.** 3D animated MDS of pairwise allele sharing dissimilarities for the closest subsets of West African, European and other admixed populations related to the TAST.

## Appendix 3

### Alternative ADMIXTURE modes, complementary to Results *Figure 3A*

In *Figure 3A*, ADMIXTURE results at $K=3$ show that the new pink cluster is maximized in East Asian Chinese-CHB individuals, with a few African-American ASW and Barbadian-ACB individuals exhibiting relatively high membership to this cluster as visually identified in the ASD-MDS analysis (*Appendix 2—figure 1*), and previously reported (*Mathias et al., 2016*). While Finnish-FIN individuals all exhibit low membership to this cluster, such signal probably emerges from unresolved clustering for this population geographically between Western Europe and East Asia. This is further evidenced by the almost complete disappearing of this resemblance of FIN individuals with the pink cluster at higher values of $K$. No other individuals or populations in our analysis show signals of strong resemblance with the East Asian Chinese-CHB for higher values of $K$.

The fourth gray cluster at $K=4$ in *Figure 3A* is maximized in Southern African individuals from the Soto and Zulu populations. At this value of $K$, the original orange cluster is maximized in the West Western African Jola from Senegambia and all other African populations exhibit intermediate membership between the gray and the orange cluster. Interestingly, the average membership to the gray "Southern African" cluster decreases with increasing geographic distance from Southern Africa, along the Atlantic Ocean coast of the continent. This echoes the relatively continuous clustering observed in the ASD-MDS analysis for the Atlantic African populations in our dataset (*Appendix 2—figures 2–4*). Notably, the Cabo Verdean individuals born on Brava, Fogo, Santiago, and Maio, exhibit membership to both the orange and the gray clusters, albeit with a majority of membership to the orange cluster rather than the gray cluster. All other Cabo Verdean individuals show virtually no membership to the gray cluster. While indicating the stronger resemblance between all Cabo Verdean individuals with genetic patterns observed mostly in West Western Africa and Senegambian populations, this provides a first indication of differentiated genetic structure across islands within Cabo Verde. Conversely, the orange cluster previously identified in the ASW and ACB is, at $K=4$ in *Figure 3A*, divided into gray and orange with a slight excess of gray. A similar pattern is found among Central and East Western African populations, such as the Yoruba-YRI, the Nigerian-ESN, the Igbo, and the Ga-Adangbe populations, hence providing a first indication of a closer resemblance of the individuals of these two enslaved-African descendant populations with East Western Africa rather than with West Western Africa.

At $K=5$ in *Figure 3A*, note that West Central and South-West African populations exhibit a distinctive unique pattern with high membership to both the red and gray clusters. However, note that ADMIXTURE clustering is largely unresolved at this value of $K$ as no individuals exhibit close (<1%) to 100% membership to the red cluster, and as only 27 out of the 30 ADMIXTURE iterations provide similar results. Indeed, an alternative minor mode is identified at this value of $K$ (*Appendix 3—figure 1*).

At $K=7$ in *Figure 3A*, the novel light-blue cluster is maximized only in Finnish-FIN individuals, while the blue cluster is now maximized only in Tuscan-TSI individuals. British-GBR and USA-CEU individuals exhibit intermediate membership between the two clusters while Iberian-IBS individuals strongly resemble TSI individuals. This clustering patterns further echoes the observed ASD-MDS clustering of European populations identified in *Figure 2*. Importantly, the 'European' membership of all Cabo Verdean islands resembles, relatively, the patterns exhibited in TSI and IBS individuals (with high proportions of blue and minimal proportion of light-blue), while African-American ASW and Barbadian-ACB individuals show relatively much higher proportions of light-blue, hereby resembling, for this genotype membership, GBR and CEU individuals rather than other Europeans. Nevertheless, variable results across runs at this value of K indicate unresolved clustering and these interpretations should be considered cautiously at this point (see alternative ADMIXTURE modes at $K=7$, *Appendix 3—figure 1*).

At $K=8$ in *Figure 3A*, the novel dark-red cluster is maximized in West Central African Tsogho individuals and at very high proportions in all other individuals from West-Central and South-West Africa, hence resolving the unresolved clustering identified for these populations at $K=5$. Interestingly, substantial traces of this cluster are found in YRI, Igbo and ESN East Western African populations as well as ASW and ACB individuals, but virtually absent from Cabo Verdeans or any other African population. This further indicates the closer resemblances of ASW and ACB African genetic component to East Western and West Central Africa than to other African populations in our dataset, as echoed in our ASD-MDS analyses (*Figure 2*, *Figure 2—figure supplement 1*).

At *K*=9 in *Figure 3A*, the novel light-yellow cluster is maximized in Sierra Leone-MSL population from Central West Africa, albeit imperfectly. This further illustrates the intermediate clustering of Central Western African populations in between Senegambian and East Western African populations, largely overlapping on both sides, and observed in ASD-MDS analyses (*Appendix 2—figures 1–4*). Nevertheless, patterns should be interpreted with caution at this value of *K*, as clustering is largely unresolved, also indicated by the alternative ADMIXTURE solutions (*Appendix 3—figure 1*).

At *K*=11 in *Appendix 3—figure 1*, we find five alternative ADMIXTURE solutions to *Figure 3*. In one alternative solution, the novel dark blue cluster is maximized either in the Fang from West Central Africa hereby distinguished from the Tsogho from the same geographic region. Other modal results show this novel dark-blue cluster maximized instead in the Wolof from West Western Africa, hence distinguished from the Jola from the same geographic region; or instead in the British GBR, thus distinguishing three different ADMIXTURE patterns among Western European populations and hence resolving minoritarian modes observed previously at *K*=10; or, finally, among individuals from Fogo island, and Brava to a lesser extent, thus indicating further sub-structure among Cabo Verdean islands. Interestingly, note that for all ADMIXTURE solutions at this value of *K*, the light-yellow cluster is majoritarian in Sierra Leone MSL individuals, hence more clearly differentiated from either West Western and East Western African individuals compared to results obtained at previous values of *K*.

At *K*=12 in *Appendix 3—figure 1*, we find three alternative modes to the results presented in *Figure 3*. These results resolve alternative modes obtained at the previous values of *K*, thus further differentiating populations within Africans and Europeans sub-regions, respectively.

At *K*=13 in *Appendix 3—figure 1*, we find five alternative mode results compared to the one presented in *Figure 3*. These further resolve the various modal results obtained at previous values of *K*=10, 11, and 12. Furthermore, note that three minoritarian such modes maximize the novel light-brown cluster in Boa Vista individuals, while memberships to this specific cluster are virtually absent from all other individuals in our data set, except for a substantial proportion in individuals born on Maio.

At *K*=14 in *Appendix 3—figure 1*, we find three alternative solutions compared to *Figure 3*. Note that some such solutions resolve separate modal solutions obtained at previous values of *K*. Furthermore, interestingly, in one modal solution, the novel bright yellow cluster is maximized in Ga-Adangabe individuals, hence differentiating this population from others in the East Western African sub-region.

Finally, at *K*=15 in *Appendix 3—figure 1*, we obtain four alternative solutions compared to *Figure 3* results. Interestingly, in two such solutions, the novel dark-green cluster is maximized in Nzebi West Central Africans thus further differentiating the three populations from Gabon (the Fang, the Tsogho, and the Nzebi) in three different cluster. Note that in this cases, geographically close South-West Central African individuals from Angola exhibit membership proportion patterns resembling more the Nzebi than the two other Gabonese populations. An alternative mode interestingly differentiates some individuals within the CHB East Asian population, and is virtually absent from all other individuals in our data set. Finally, the last alternative mode encompasses the smallest number of runs and resolves some of the alternative solutions previously obtained at *K*=14 (*Figure 3A*, *Appendix 3—figure 1*).

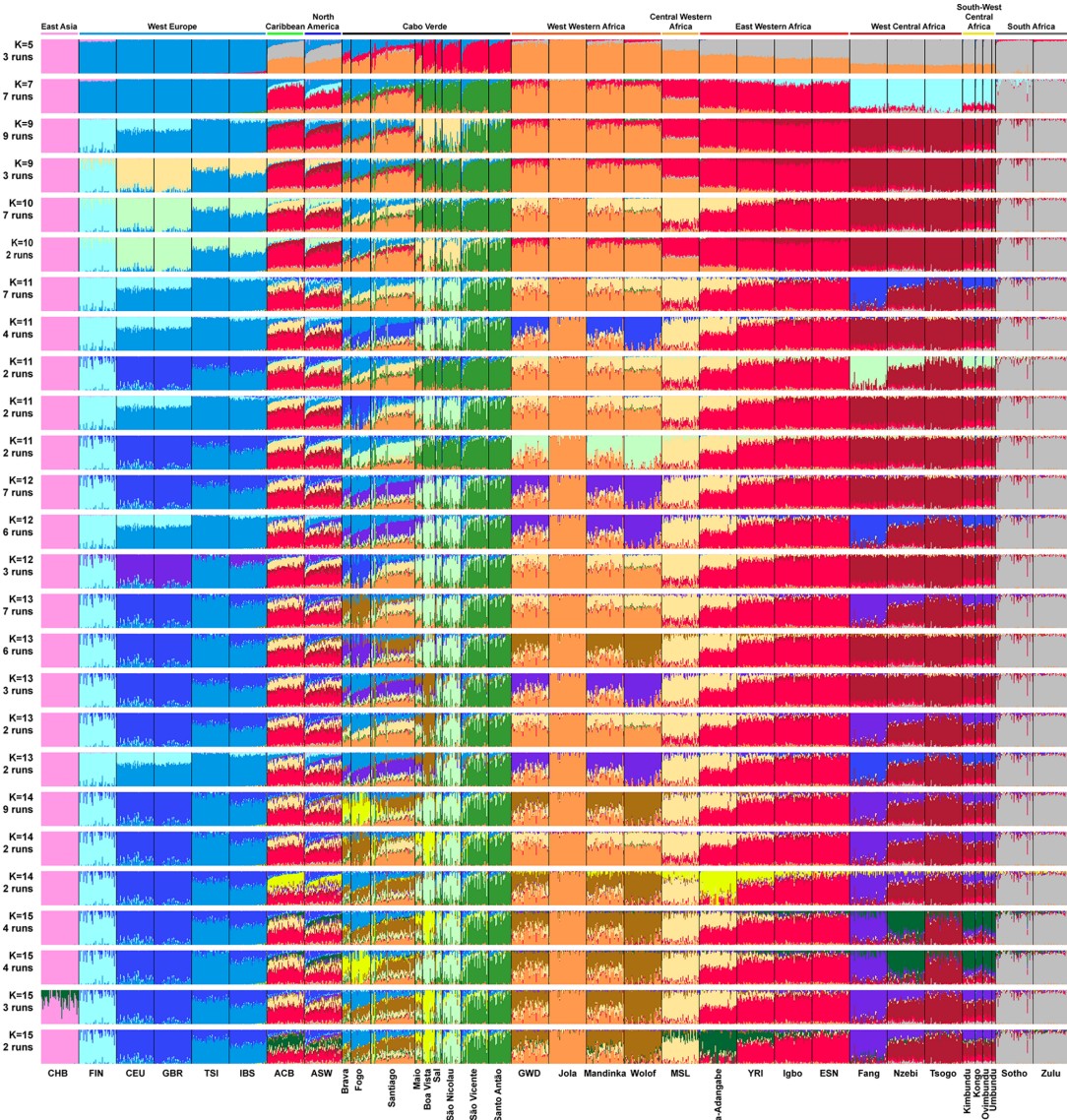

**Appendix 3—figure 1.** Alternative ADMIXTURE mode results for the individual genetic structure among Cabo Verdean, Barbadian-ACB and African-American ASW populations related to the TAST. ADMIXTURE (*Alexander et al., 2009*) analyses using resampled individual sets for the population sets originally containing more than 50 individuals (*Figure 1—source data 1*). 225 unrelated Cabo Verdean-born individuals are grouped by island of birth (*Figure 1—source data 1*). All analyses considered 102,543 independent autosomal SNPs, for values of *K* between 2 and 15. 30 independent ADMIXTURE runs were performed for each value of *K*, and groups of runs (>2) providing similar results (all pairwise SSC >99.9%) were averaged in a single 'mode' result using CLUMPP (*Jakobsson and Rosenberg, 2007*), and plotted with DISTRUCT (*Rosenberg, 2003*). Number of runs in the presented modes are indicated below the value of *K*. All other modes are presented in *Figure 3*.

# Appendix 4

## Runs of homozygosity patterns in Cabo Verde and other admixed populations related to the TAST, complementary to *Figure 4*

Examining patterns of ROH less than 1 cM long, we find generally low levels of short and medium ROH in continental Africa (*Appendix 4—figure 1*), with the lowest levels in Southern Africa. Conversely, we found high levels of such short and medium-size ROH among European and East Asian populations, likely reflecting ancient migration bottlenecks during the Out-of-Africa expansion (*Szpiech et al., 2019*). We note that the Cabo Verdean individuals have slightly elevated levels of short and medium ROH compared to continental Africa, similarly to the ACB and ASW populations. This may stem from the known admixture with European populations that occurred recently during the TAST. Admixture can be intuitively expected to decrease ROH levels compared to the source populations in general. However, recent admixture between African and European populations, the latter who exhibit much higher levels of short and medium ROH than the former (*Appendix 4—figure 1*), may inflate levels of such ROHs in the admixed population compared to continental African sources. Consistently with our findings for the African American ASW and Barbadian ACB (*Appendix 4—figure 1*), such phenomenon had been previously reported (*Mooney et al., 2018*). Finally, overall total ROH levels reflected the generally higher levels of ROH in Cabo Verde compared to continental African and other admixed populations from the Americas (*Appendix 4—figure 1C*), patterns likely related to the relative isolation of populations within the archipelago (*Figure 5*, *Table 1*).

Exploring how local ancestry intersects with ROH less than 1 cM long, since short and medium ROH are comprised of old haplotypes that likely predate over 50 generations ago (*Thompson, 2013*), the colonization of the Cabo Verde islands and the subsequent admixture histories of their populations, we expected that the local ancestry content of these ROH classes should be correlated with total non-ROH local ancestry levels. *Appendix 4—figure 2* illustrates the relationship between total ancestry in a given ROH size class versus the total ancestry not in that ROH size class for all Cabo Verdean individuals. We find strong correlations among the short ROH and local ancestry (*Appendix 4—figure 2A*; African local ancestry: Pearson $\rho$=0.9372, $p<2.2 \times 10^{-16}$; European local ancestry: Pearson $\rho$=0.9496, $p<2.2 \times 10^{-16}$; Asian local ancestry: Pearson $\rho$=0.7516, $p<2.2 \times 10^{-16}$), and medium ROH and local ancestry (*Appendix 4—figure 2B*; African local ancestry: Pearson $\rho$=0.9071, $p<2.2 \times 10^{-16}$; European local ancestry: Pearson $\rho$=0.9379, $p<2.2 \times 10^{-16}$; Asian local ancestry: Pearson $\rho$=0.3905, $p<1.298 \times 10^{-9}$).

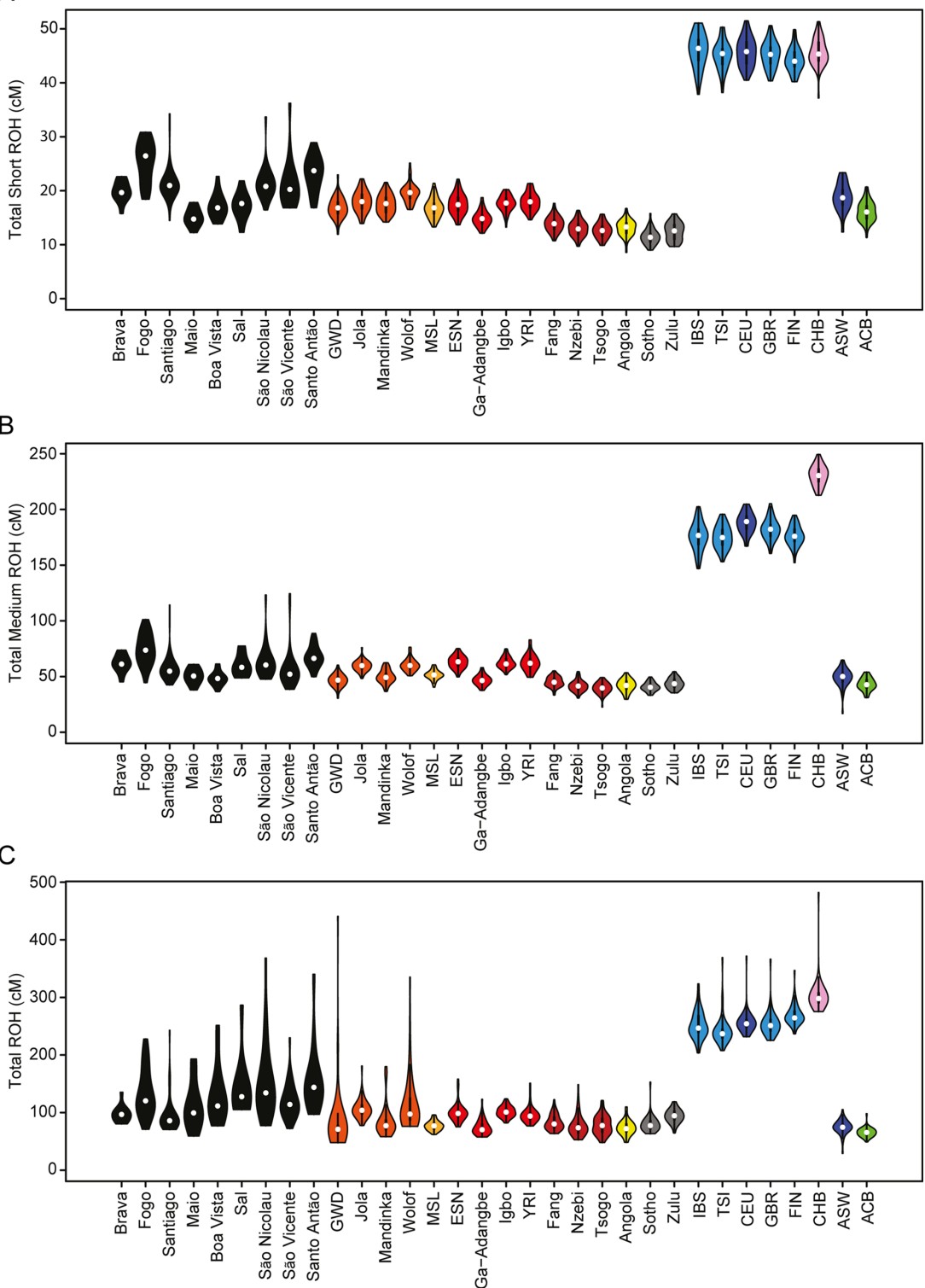

**Appendix 4—figure 1.** The distribution of (**A**) short, (**B**) medium, and (**C**) all ROH per individual, for each Cabo Verdean birth-island and other worldwide populations.

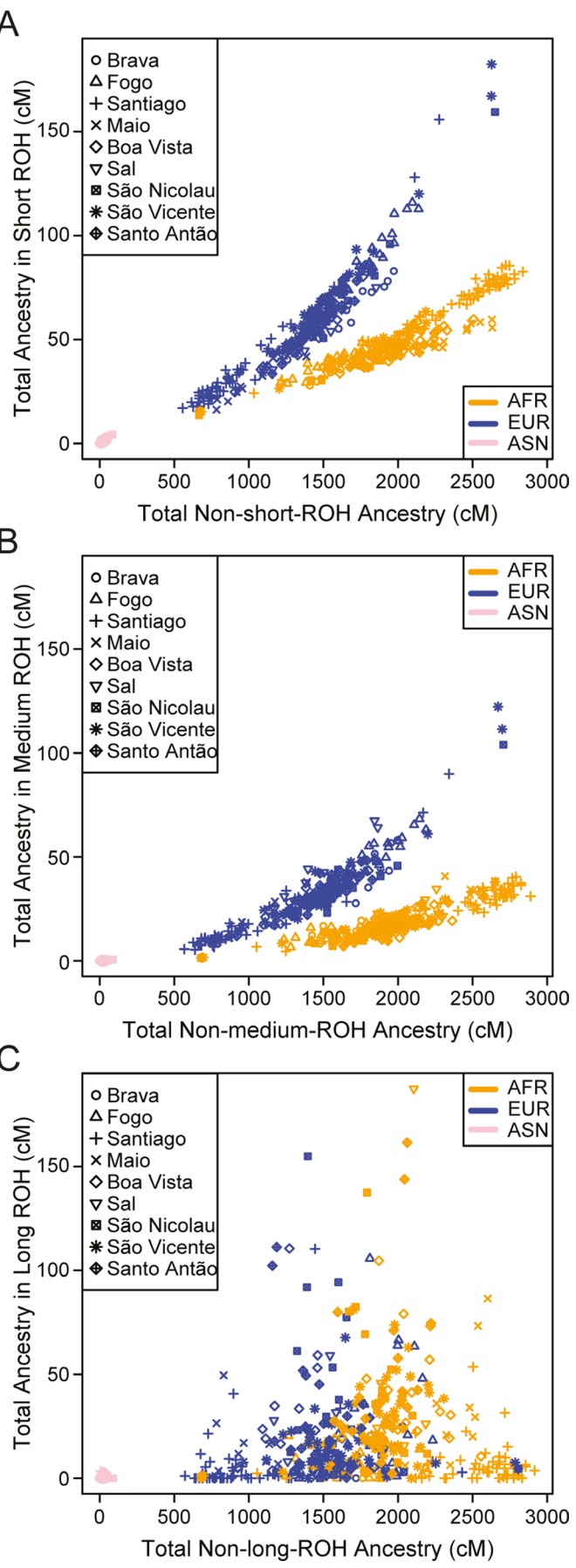

**Appendix 4—figure 2.** Individual total ancestry contained in (**A**) short, (**B**) medium, and (**C**) long ROH versus total ancestry not in that class ROH, for each Cabo Verdean birth-island and other worldwide populations.

# Appendix 5

## Genetic admixture histories of each Cabo Verdean island of birth inferred with *MetHis*-ABC, complementary to *Figure 7* and Discussion sections

Results for the detailed admixture history of each island separately are henceforth presented in the chronological order of the first time in an islands' history when more than 100 individuals are reported in historical census information, and after which further census reports always exceeded this number until today; hereafter called 'first perennial peopling' of an island (*Figure 7D*, *Figure 7—source data 1*). As in the main text of the article, note that this historical date can vastly differ from the date for the first official settlement of an island inferred via political, administrative and tax records, and witnessed by parishes foundations, illustrating the complex and versatile peopling history of each Cabo Verdean island after the initial establishment of the Portuguese dominion over the archipelago.

## Admixture history of Santiago (First official settlement 1460; Perennial peopling: 1470)

### *MetHis*-ABC inferences detailed results for Santiago

For individuals born on Santiago island, we find that the most recent pulses of admixture from Africa and Europe, respectively, occurred after the official abolition of the TAST in the Portuguese Empire in the 1830s, and after the effective abolition of slavery in the Portuguese empire in the 1860s (*Figure 7*). Both admixture pulses are characterized by posterior distributions with narrow 95%CIs clearly departing from their respective priors, both for the timing of each pulse and their respective intensities, in particular for the European admixture pulse (*Figure 7—figure supplements 1–3*, *Appendix 5—table 1*). For these recent events, observed genetic patterns are consistent with a relatively intense African admixture event (median = 0.5913, mode = 0.6837, 95%CI=[0.0642–0.9038]), occurring between generation sixteen and seventeen after the founding of Cabo Verde at generation 0 (generation median = 16.6, mode = 17.2, 95%CI=[10.0–18.9]). Furthermore, we find strong evidence for the most recent European admixture pulse being weak (median = 0.0684, mode = 0.0258, 95%CI=[0.0028–0.6021]), and having occurred between one and two generations before the sampled generation (generation median = 18.7, mode = 19.1, 95%CI=[11.7–19.6]).

NN-ABC posterior parameter estimations perform less accurately to infer the dates of the earlier admixture pulses and their associated intensities, both from Africa and Europe respectively, as shown by posterior distribution departing less clearly from their respective priors and overall large 95% CI. Nevertheless, results indicate that both the African and European pulses may have occurred in a more remote past; the African pulse was inferred at generation 9.2 (median) after founding at generation 0 (mode = 13.9, 95%CI=[1.7–16.8]); the European pulse at generation 7.5 (median, mode = 3.4, 95%CI=[2.3–15.9]), towards the beginning of the plantation economy era in the Americas and during the intensification of the TAST in the second half of the 17th century (*Eltis and Richardson, 2015*; *Fortes-Lima and Verdu, 2021*; *Figure 7D*). Conversely, the initial founding admixture pulse, set to occur at generation 0 at the beginning of Cabo Verde settlement history in the 1460s, is more accurately estimated a posteriori, as indicated by substantial departure from the prior distribution and relatively narrow CI. Results here indicate a stronger contribution from Africa than Europe (median = 0.6365, mode = 0.8816, 95%CI=[0.0538–0.9858]), at the root of the genetic make-up of Santiago-born individuals in our sample set.

Finally, effective population sizes in Santiago indicate a strong, relatively linear ($u_{Ne}$ median = 0.365, mode = 0.440, 95%CI=[0.150–0.494]), increase since the 1460s up to 80,077 effective individuals (median, mode = 92,109, 95%CI=[30,408–98,975]), in the sampled generation born on average between the 1960s and the 1980s.

### Discussion for the admixture history of Santiago

Altogether, our results indicate that admixture in Santiago occurred mainly between Senegambian and South Western European populations (*Figures 2–3*), at the very beginning of the Portuguese colonization of the archipelago in the 15th century. This early admixture history is likely due to an initial peopling of Cabo Verde strongly biased towards solitary males (familial permanent migrations from Portugal being a minority), who reproduced with African enslaved women, and whose admixed

**Appendix 5—table 1.** SantiagoNN-ABC posterior parameter estimations for Santiago island, cross-validation posterior parameter errors, and cross-validation 95% Credibility Interval accuracies. Cross-validation errors and 95% CI accuracies are based on 1000 NN-ABC posterior parameter inferences using, in turn, each one of the 1000 simulations closest to the observed data used as pseudo-observed data, and the remaining 99,999 simulations as the reference table. Error calculations are based on the mode and, separately, the median point estimate for each 1000 pseudo-observed simulations posterior parameter estimation, compared to the known parameter used for simulation. We considered six neurons in the hidden layer and a tolerance level of 0.01 (1000 simulations) for all posterior parameter estimation in this analysis as identified in *Appendix 1—table 1*. Plotted distributions for posterior parameter estimations can be found in *Figure 7—figure supplements 1–3*.

| SANTIAGO | Parameter posterior estimation | | | | 1,000 cross-validation errors | | |
|---|---|---|---|---|---|---|---|
| Afr2P-Eur2P scenario parameters | Mode | Median | 50% Credibility Interval | 95% Credibility Interval | Mode mean absolute error | Median mean absolute error | 95% CI length accuracy |
| Ne.0 | 142 | 431 | [194 - 692] | [28 - 974] | 305 | 255 | 0.059 |
| Ne.20 | 92109 | 80077 | [63220–91349] | [30408–98975] | 25426 | 21558 | 0.045 |
| u.Ne | 0.440 | 0.365 | [0.284–0.438] | [0.150–0.494] | 0.113 | 0.096 | 0.042 |
| sAfr,0 | 0.8816 | 0.6365 | [0.3884–0.8415] | [0.0538–0.9858] | 0.3147 | 0.2507 | 0.056 |
| tAfr,p1 | 13.9 | 9.2 | [4.9–13.4] | [1.7–16.8] | 5.3 | 4.2 | 0.101 |
| sAfr,tAfr,p1 | 0.7844 | 0.5986 | [0.3421–0.7861] | [0.0420–0.9479] | 0.2618 | 0.2204 | 0.054 |
| tAfr,p2 | 17.2 | 16.6 | [15.2–17.5] | [10.0–18.9] | 1.8 | 1.6 | 0.199 |
| sAfr,tAfr,p2 | 0.6837 | 0.5913 | [0.3894–0.7241] | [0.0642–0.9038] | 0.1597 | 0.1458 | 0.052 |
| tEur,p1 | 3.4 | 7.5 | [4.3–11.1] | [2.3–15.9] | 4.0 | 3.1 | 0.161 |
| sEur,tEur,p1 | 0.1617 | 0.3969 | [0.1957–0.6606] | [0.0185–0.9564] | 0.3042 | 0.2439 | 0.056 |
| tEur,p2 | 19.1 | 18.7 | [17.7–19.2] | [11.7–19.6] | 3.0 | 2.9 | 0.115 |
| sEur,tEur,p2 | 0.0258 | 0.0684 | [0.0250–0.1501] | [0.0028–0.6021] | 0.2749 | 0.2254 | 0.057 |

offspring constituted the first Cabo Verdean generations, as shown by some recorded instances of legitimizations of the admixed offspring to allow them to legally inherit dating back to the early 16th century (*Carreira, 2000*).

Our results further indicate that two substantial admixture events from Africa and Europe, respectively, occur concomitantly to the early expansion of the TAST during the 17th-century, although we are unable to precisely date both events. Nevertheless, our results support a more tenuous genetic admixture history from Africa experienced by Cabo Verdeans from Santiago during the most intense period of the TAST (18th-century) and the height of plantation economy in European colonial empires (*Eltis and Richardson, 2015*; *Fortes-Lima and Verdu, 2021*). This apparent reduced influence of the most intense period of the TAST on admixture pattern in Santiago is also echoed in our ASD-MDS, ADMIXTURE, and SOURCEFIND results, where Santiago Cabo Verdeans today share ancestry with Senegambian Mandinka almost exclusively (*Figures 2–3*). However, historical records unquestionably demonstrated that numerous other populations from West Central and South Western Africa were enslaved and forcibly deported to Cabo Verde during this era (*Carreira, 2000*). Thus, our results highlight that the strong and recurrent demographic migrations of enslaved Africans brought to Santiago during the second half of the TAST do not seem to have left a similar genetic admixture signature. This is consistent with several, mutually non-exclusive, historical scenarios, where enslaved African populations brought to Santiago may have been mostly re-deported to the Americas and Europe after the 17th century, without substantially contributing to the genetic landscape of this island, whether due to rapid re-deportation, and/or to slave-owners and colonial society strongly preventing marriages and reproduction between newly arrived enslaved individuals and pre-existing enslaved or non-enslaved communities in Santiago (see **Discussion** in main text).

Conversely, our results indicate an intense introgression event from Africa in Santiago, occurring shortly after the TAST and the abolition of slavery in the Portuguese empire in the 1860s. This may have been caused by a relaxation of the socio-economic constraints that predominated during most of the TAST against marriages between enslaved and non-enslaved communities as well as the control of marriages among enslaved individuals imposed by slave-owners; concomitant with the strong intensification of the illegal slave-trade which deported numerous enslaved Africans in a short amount of time shortly after the end of the TAST (*Carreira, 2000*; *Albuquerque and Santos, 1991*; *Albuquerque and Santos, 1995*; *Albuquerque and Santos, 2007*).

Our results also clearly identify a European introgression event in Santiago having occurred at the beginning of the 20th century, which may be explained by the European migrations towards African colonies triggered by European empires' policies during the first half of this century. Interestingly, we find that this recent admixture event was of very limited intensity, thus indicating that European emigration to Santiago during the 20th century only had a quantitatively limited, albeit precisely detectable, influence on genetic admixture patterns.

Finally, Santiago-born individuals also show the lowest ROH levels and generally some of the shortest ROH in Cabo Verde (*Figure 4* and *Appendix 4—figures 1–2*). When examining ancestry patterns within ROH in Santiago-born individuals, although proportions within ROH are on average consistent with an individuals' overall genetic ancestry, there are some outlier individuals with much higher European ancestry within ROH compared to the remainder of the genome. This could possibly reflect historical marriage prohibitions and other social constraints against intercommunity reproductions (see **Discussion** in main text). These ROH results combined with the evidence for the relatively constantly increasing reproductive population in Santiago over the course of the peopling of Cabo Verde (*Figure 7C*), was consistent with this island being the administrative historical capital, and principal entry to the archipelago throughout its' history (*Albuquerque and Santos, 1991*; *Albuquerque and Santos, 1995*).

## Admixture history of Fogo (First official settlement 1460; Perennial settlement: 1570)

### *MetHis*-ABC inferences detailed results for Fogo

For individuals born on Fogo island, we find that both European admixture pulses were precisely inferred with NN-ABC, for the time of their occurrence and for their respective intensities, as posterior parameter distributions and 95% CI substantially depart from their respective priors (*Figure 7*, *Figure 7—figure supplements 1–3*, *Appendix 5—table 2*). The most recent European admixture pulse in Fogo occurred concomitantly to the abolition of the TAST and of slavery in the Portuguese empire in the 1860's (generation median = 15.5 after the founding of Cabo Verde at generation 0, mode = 16.7, 95%CI=[6.7–18.2]), and was of substantial intensity (median = 0.2428, mode = 0.2101, 95%CI=[0.0157–0.8485]). The older European pulse occurred much more remotely in the past (generation median = 5.3, mode = 2.5, 95%CI=[1.9–13.9]), shortly after the first perennial census above 100 individuals in 1570 for this island, and long before the increase in TAST intensity due to the expansion of Plantation Economy in the Americas in the second half of the 17th century. Furthermore, note that our results indicate that this European pulse is of relatively similar intensity compared to the most recent one (median = 0.2234, mode = 0.1149, 95%CI=[0.0223–0.8383]).

Interestingly, we find that the African admixture history of individuals from Fogo occurred at very different periods of time than the European ones, the timing for both African admixture pulses being also well estimated with substantial departure from the prior and relatively narrow 95% CI. Indeed, we find strong indications of a very recent pulse of African admixture of overall limited intensity (median = 0.2034, mode = 0.1174, 95%CI=[0.0120–0.8225]), having occurred around the first part of the 20th century (generation median = 19.3 after the founding of Cabo Verde at generation 0, mode = 19.5, 95%CI=[15.6–19.8]). Furthermore, we find that the previous African admixture pulse likely occurred between six and eight generations before the sampled one (generation median = 12.0 after Cabo Verde founding, mode = 13.3, 95%CI=[4.7–16.9]), in the midst of the most intense period of the TAST and plantation economy in the Americas, around the 1760s. However, we cannot recover a precise estimation for the intensity of this older pulse, as its' posterior parameter distribution seldom depart from its' prior. We also find that the founding admixture event set to occur at the beginning of the colonization history of the archipelago in the 1460's is substantially departing from its' prior distribution and very similar to the founding admixture event found for individuals born on Santiago (median = 0.6831, mode = 0.7407, 95%CI=[0.1679–0.9600]).

Finally, our posterior estimation of the demographic history of Fogo-born individuals substantially differs from that of Santiago. Indeed, while we find similar effective sizes towards the present between the two islands (median = 72,023, mode = 90,370, 95%CI=[18,011–98,846]), we find that the demographic increase was much more recent and rapid in Fogo ($u_{Ne}$ median = 0.150, mode = 0.115, 95%CI=[0.055–0.447]) than in Santiago.

**Appendix 5—table 2.** Fogo NN-ABC posterior parameter estimations for Fogo island, cross-validation posterior parameter errors, and cross-validation 95% Credibility Interval accuracies. Cross-validation errors and 95% CI accuracies are based on 1000 NN-ABC posterior parameter inferences using, in turn, each one of the 1000 simulations closest to the observed data used as pseudo-observed data, and the remaining 99,999 simulations as the reference table. Error calculations are based on the mode and, separately, the median point estimate for each 1000 pseudo-observed simulations posterior parameter estimation, compared to the known parameter used for simulation. We considered 10 neurons in the hidden layer and a tolerance level of 0.01 (1000 simulations) for all posterior parameter estimation in this analysis as identified in *Appendix 1—table 1*. Plotted distributions for posterior parameter estimations can be found in *Figure 7—figure supplements 1–3*.

| FOGO | Parameter posterior estimation | | | | 1000 cross-validation errors | | |
|---|---|---|---|---|---|---|---|
| Afr2P-Eur2P scenario parameters | Mode | Median | 50% Credibility Interval | 95% Credibility Interval | Mode mean absolute error | Median mean absolute error | 95% CI length accuracy |
| Ne.0 | 349 | 511 | [274 - 737] | [44 - 969] | 295 | 253 | 0.063 |
| Ne.20 | 90370 | 72023 | [50948–87248] | [18011–98846] | 27142 | 22749 | 0.057 |
| u.Ne | 0.115 | 0.150 | [0.104–0.230] | [0.055–0.447] | 0.085 | 0.081 | 0.074 |
| sAfr,0 | 0.7407 | 0.6831 | [0.5218–0.8152] | [0.1679–0.9600] | 0.2936 | 0.2432 | 0.069 |
| tAfr,p1 | 13.3 | 12.0 | [8.1–14.4] | [4.7–16.9] | 4.2 | 3.3 | 0.157 |
| sAfr,tAfr,p1 | 0.4667 | 0.4549 | [0.2286–0.6721] | [0.0251–0.9555] | 0.2498 | 0.2315 | 0.056 |
| tAfr,p2 | 19.5 | 19.3 | [18.9–19.5] | [15.6–19.8] | 2.6 | 2.5 | 0.100 |
| sAfr,tAfr,p2 | 0.1174 | 0.2034 | [0.1051–0.3560] | [0.0120–0.8225] | 0.2032 | 0.1891 | 0.060 |
| tEur,p1 | 2.5 | 5.3 | [2.7–8.8] | [1.9–13.9] | 3.6 | 3.0 | 0.216 |
| sEur,tEur,p1 | 0.1149 | 0.2234 | [0.1146–0.3943] | [0.0223–0.8383] | 0.2804 | 0.2371 | 0.062 |
| tEur,p2 | 16.7 | 15.5 | [13.1–16.8] | [6.7–18.2] | 3.2 | 3.0 | 0.070 |
| sEur,tEur,p2 | 0.2101 | 0.2428 | [0.1393–0.3966] | [0.0157–0.8485] | 0.1751 | 0.1686 | 0.060 |

## Discussion for the admixture history of Fogo

Our results indicate a very similar founding admixture event for Fogo island compared to that of Santiago, strongly suggesting that the admixture event at the root of Fogo peopling occurred early in Cabo Verde history, likely in Santiago island. This scenario is consistent with historical data suggesting an initial peopling of Fogo from Santiago at the end of the 15th century and the beginning of the 16th century, rather than independent founding events between islands (*Albuquerque and Santos, 1991*; *Albuquerque and Santos, 1995*).

Interestingly, we find a European pulse of admixture concomitant with the perennial peopling of the island at the end of the 16th century and the establishment of Fogo as an important center of agricultural and free-range cattle herding in the economic triangle between Europe, Cabo Verde, and continental Africa. Later on, we identify traces of an African admixture pulse having occurred during the most intense period of the TAST in the mid-18th century, thus consistent with increased African enslaved individuals' forced deportations to Fogo as showed by historical records, but we could not infer its intensity, suggesting the difficult identifiability of such admixture event in the genetic landscape of Fogo island today with the methods here deployed.

This history of admixture is likely related to the intense control on the island's economic expansion and regulated immigration imposed after 1532 by the central political, administrative, and commercial

power of Santiago, with the notable exception of direct commercial relationships allowed between Fogo and Europe under the rule of the Portuguese crown (*Albuquerque and Santos, 1991*; *Albuquerque and Santos, 1995*; *Albuquerque and Santos, 2007*). This could thus explain the limited African admixture during Fogo history and more intense European admixture on this island, identified with our ASD-MDS/ADMIXTURE, and SOURCEFIND results (*Figures 2–3*). Furthermore, these socio-political factors likely also explain the maintaining of a relatively low reproductive population size on Fogo until very recently, as opposed to the more continuous population increase on Santiago (*Figure 7C*). In addition, this demographic history may be further explained by the recurrent eruptions of the Fogo volcano, which unquestionably chronically disrupted the island's development, although historical records showed limited direct mortality due to these events and that communities fleeing the island often migrated back after each cataclysm (*Albuquerque and Santos, 1991*).

We identify a much more recent pulse of European admixture, having occurred in the first half of the 19th century, shortly after the abolition of the TAST in the Portuguese empire, possibly due to the profound socio-cultural changes in marital relationships between enslaved and non-enslaved communities during the abolition of slavery in colonial empires (see **Discussion** in main text). Finally, we identify precisely a weak African admixture pulse having occurred in Fogo during the 20th century possibly consistent with more recent work-related migrations from Africa to Cabo Verde (*Albuquerque and Santos, 2007*).

Finally, ROH patterns in Fogo are similar to Santiago, in that ROH levels are relatively low and individual ROH are relatively short (*Figure 4*, *Appendix 4—figures 1–2*). This is consistent with Fogo's history of economic control and influx of European individuals. Interestingly, when exploring ancestry proportions within ROH relative to the remainder of the genome, we see on average more European ancestry within ROH. This could possibly reflect the historical influx of Europeans onto the island and/or a legacy of preferential marriages between individuals of shared family origin.

## Admixture history of Santo Antão (First official settlement 1570; Perennial settlement: 1580)

### *MetHis*-ABC inferences detailed results for Santo Antão

For individuals born on Santo Antão island, we find that both European admixture pulses after the founding admixture event are substantially departing from their respective priors and have relatively narrow 95% CI, both for the timing of their occurrence and their respective intensities (*Figure 7*, *Figure 7—figure supplements 1–3*, *Appendix 5—table 3*). We find that Santo Antão-born individuals have experienced a recent European admixture pulse of weak intensity (median = 0.0847, mode = 0.0467, 95%CI=[0.0056–0.6382]), that occurred roughly three generations before the sampled generation, at the turn of the 20th century (generation median = 17.2 after Cabo Verde founding at generation 0, mode = 18.1, 95%CI=[7.6–19.1]). Furthermore, we find a much more ancient European admixture pulse of strong intensity (median = 0.3592, mode = 0.1542, 95%CI=[0.0216–0.9522]), occurring concomitantly (generation median = 5.8, mode = 3.2, 95%CI=[2.6–15.6]) with the first perennial census in this island dated in 1580, and before the massive increase of TAST during the 17th century.

Interestingly, we find that the first African admixture pulse after founding at generation 0 occurred at generation 4.9 (median, mode = 2.4, 95%CI=[1.9–11.9]), almost synchronically to the oldest European admixture pulse and the first perennial settlement of the island recorded in 1580. Furthermore, this African admixture pulse was intense (median = 0.6255, mode = 0.7574; 95%CI=[0.0451–0.9749]), which summed close to 100% with the European admixture pulse having occurred at very similar times. This obliterated any older admixture events, consistently with the wide 95% CI obtained for the intensity of the founding admixture event for this island (median $s_{Afr,0}$=0.5681, mode = 0.8047, 95%CI=[0.0260–0.9839]). Importantly, our NN-ABC posterior parameter estimation fail to infer the second African admixture pulse identified by our RF-ABC scenario-choice. Indeed, the posterior distributions of both the timing of this most recent pulse and its' intensity depart little from their respective priors.

Finally, despite the relatively poor performances of our inferences of demographic changes in Santo Antão, our results are indicative of a very reduced population in this island over the course of Cabo Verdean peopling history until a very recent increase up to 41,256 individuals (median, mode = 12,056, 95%CI=[3570–96,855]).

**Appendix 5—table 3.** Santo AntãoNN-ABC posterior parameter estimations for Santo Antão island, cross-validation posterior parameter errors, and cross-validation 95% Credibility Interval accuracies. Cross-validation errors and 95% CI accuracies are based on 1000 NN-ABC posterior parameter inferences using, in turn, each one of the 1000 simulations closest to the observed data used as pseudo-observed data, and the remaining 99,999 simulations as the reference table. Error calculations are based on the mode and, separately, the median point estimate for each 1000 pseudo-observed simulations posterior parameter estimation, compared to the known parameter used for simulation. We considered 11 neurons in the hidden layer and a tolerance level of 0.01 (1000 simulations) for all posterior parameter estimation in this analysis as identified in *Appendix 1—table 1*. Plotted distributions for posterior parameter estimations can be found in *Figure 7—figure supplements 1–3*.

| SANTO ANTAO | Parameter posterior estimation | | | | 1,000 cross-validation errors | | |
|---|---|---|---|---|---|---|---|
| Afr2P-Eur2P scenario parameters | Mode | Median | 50% Credibility Interval | 95% Credibility Interval | Mode mean absolute error | Median mean absolute error | 95% CI length accuracy |
| Ne.0 | 411 | 469 | [254 - 706] | [26 - 964] | 254 | 228 | 0.071 |
| Ne.20 | 12056 | 41256 | [17145–69559] | [3570–96855] | 32106 | 25308 | 0.051 |
| u.Ne | 0.035 | 0.053 | [0.033–0.093] | [0.015–0.334] | 0.072 | 0.067 | 0.056 |
| sAfr,0 | 0.8047 | 0.5681 | [0.2995–0.7964] | [0.0260–0.9839] | 0.3029 | 0.2586 | 0.068 |
| tAfr,p1 | 2.4 | 4.9 | [2.9–7.4] | [1.9–11.9] | 3.3 | 2.6 | 0.164 |
| sAfr,tAfr,p1 | 0.7574 | 0.6255 | [0.3644–0.7916] | [0.0451–0.9749] | 0.2512 | 0.2283 | 0.055 |
| tAfr,p2 | 13.6 | 11.6 | [7.7–14.9] | [3.0–18.5] | 2.8 | 2.7 | 0.072 |
| sAfr,tAfr,p2 | 0.6294 | 0.4716 | [0.2596–0.6733] | [0.0338–0.9233] | 0.2435 | 0.2150 | 0.049 |
| tEur,p1 | 3.2 | 5.8 | [3.2–10.2] | [2.6–15.6] | 2.6 | 2.4 | 0.265 |
| sEur,tEur,p1 | 0.1542 | 0.3592 | [0.1709–0.6051] | [0.0216–0.9522] | 0.2767 | 0.2347 | 0.059 |
| tEur,p2 | 18.1 | 17.2 | [14.8–18.2] | [7.6–19.1] | 3.2 | 3.0 | 0.074 |
| sEur,tEur,p2 | 0.0467 | 0.0847 | [0.0416–0.1656] | [0.0056–0.6382] | 0.1868 | 0.1763 | 0.059 |

## Discussion for the admixture history of Santo Antão

Our results for the admixture history of Santo Antão indicate that individuals born today on this island experienced an African-European admixture event concomitant with the first perennial settlement of the island in the 1580s, at the root of the genetic makeup of this island.

Most interestingly, we find no identifiable genetic admixture events, from Europe or Africa, throughout the height of the TAST on Santo Antão, albeit this island experienced a strong agricultural expansion during the plantation economy era which triggered recurrent African enslaved individuals' deportations to this island, as shown by historical records (*Albuquerque and Santos, 1991*; *Albuquerque and Santos, 1995*; *Albuquerque and Santos, 2007*). Thus, even more strikingly than observed in Santiago, our results indicate that the enslaved-African forced deportations to Santo Antão during most of the TAST did not leave a statistically significant genetic contribution to the genetic landscape of the island, for likely the same complex socio-cultural reasons suggested for the rest of Cabo Verde (see **Discussion** in main text).

In this context, our ADMIXTURE and SOURCEFIND results identify two different Senegambian populations, the Mandinka and the Wolof, sharing significant African haplotypic ancestry with Santo Antão-born individuals, as well as a much more limited shared ancestry with the West Central Africa Igbo population (*Figure 3B*). Our ABC results would thus be consistent with a scenario where different enslaved-African populations from Senegambia and West Central Africa where brought together to settle Santo Antão in the early stages of the TAST in the late 16th century, a pattern distinct to our findings for Santiago and Fogo. Alternatively, it is possible that the Igbo shared ancestry identified in our analyses stem from another admixture event that we failed to identify. This is likely due to an overall limited shared ancestry between this population

and Santo Antão identified only with haplotypic local ancestry inferences, as our *MetHis*-ABC procedure considered a single Senegambian population at the root of the admixture history of the island and a much more limited number of independent SNPs compared to the haplotypic local ancestry analysis.

Finally, we find that a weak but significant European admixture pulse occurred at the turn between the 19th and the 20th centuries, substantially later than the end of the TAST and the abolition of slavery and before the 20th European colonial migrations, whose putative historical causes remain to be elucidated. Furthermore, we find that the reproductive population of Santo Antão remained overall small during its' entire history until a very recent expansion, although much more limited than the expansions observed in Santiago and Fogo. This is somewhat surprising with the census records showing that Santo Antão was the second most peopled island after Santiago during most of Cabo Verde history (*Figure 7—source data 1*). Nevertheless, our results could be explained by the demographic collapse experienced by Santo Antão since the beginning of the 20th century, due to several intense famine episodes and rural-urban emigration from the island; a demographic history scenario thus much more complex than the relatively simple scenario here explored. Reflecting the historically small population size on the island, Santo Antão show high total ROH and generally quite long individual ROH among the islands (*Figure 4* and *Appendix 4—figures 1–2*).

## Admixture history of São Nicolau (First official settlement 1580; Perennial settlement: 1580)

### MetHis-ABC inferences detailed results for São Nicolau

For individuals born on São Nicolau island, we find that the two most recent admixture pulses from Europe and Africa, respectively, are precisely estimated with posterior distributions substantially departing from their priors and reduced 95% CI, for the pulses' timing and intensities (*Figure 7*, *Figure 7—figure supplements 1–3*, *Appendix 5—table 4*). We find that the most recent pulse of European admixture is weak in intensity (median = 0.0731, mode = 0.0297, 95%CI=[0.0038–0.6642]), and occurred very recently at the generation before that of the sampled individuals (generation median = 19.0 after the founding at generation 0, mode = 19.2, 95%CI=[10.3–19.8]). Furthermore, we find that the most recent African admixture pulse is very intense (median = 0.6398, mode = 0.7132, 95%CI=[0.0877–0.9380]), and occurred at generation 14.5 after founding (median, mode = 14.6, 95%CI=[8.3–16.4]), during the abolition of the TAST in the first half of the 19th century.

Interestingly, for both the more ancient admixture pulses from either Africa or Europe, our NN-ABC posterior parameter estimation provide reasonably informative posterior distributions pointing towards relatively weak African and European admixture intensities (median = 0.3133, mode = 0.1695, 95%CI=[0.0171–0.9262]; and median = 0.3240, mode = 0.1193, 95%CI=[0.0145–0.9460], respectively). However, our results only provide indications that both pulses were ancient and occurred synchronically (generation median = 5.2 after founding, mode = 2.4, 95%CI=[1.3–15.5]; generation median = 6.7, mode = 2.7, 95%CI=[2.0–16.1], for the African and European older pulses, respectively), as posterior distributions for the timing of these events are largely confounded with their respective priors.

Finally, we obtain a posterior distribution of the founding admixture intensity largely confounded with its prior, similarly as for the demographic parameters, thus showing a limit of our ABC inferences procedures.

### Discussion for the admixture history of São Nicolau

Our results for the admixture history of São Nicolau indicate that two moderate admixture pulses from Africa and Europe occurred synchronically between the date for the first perennial settlement of this island in the 1580s and the onset of the intensification of the TAST at the beginning of the 17th century. Much later in the course of Cabo Verde peopling history at the beginning of the 19th century, we find an intense introgression event from Africa, probably occurring for the same reasons as in Santiago (see above) and other Cabo Verde islands (see below and **Discussion** in main text).

Interestingly, our haplotypic local ancestry inferences identify three African populations with shared ancestries with São Nicolau-born individuals (*Figure 3*); Senegambian Wolof and Mandinka shared-ancestry representing the majority similarly to the rest of Cabo Verde and as expected historically (see **Discussion** in the main text, *Carreira, 2000*; *Albuquerque and Santos, 1991*; *Albuquerque*

**Appendix 5—table 4.** São NicolauNN-ABC posterior parameter estimations for São Nicolau island, cross-validation posterior parameter errors, and cross-validation 95% Credibility Interval accuracies. Cross-validation errors and 95% CI accuracies are based on 1000 NN-ABC posterior parameter inferences using, in turn, each one of the 1000 simulations closest to the observed data used as pseudo-observed data, and the remaining 99,999 simulations as the reference table. Error calculations are based on the mode and, separately, the median point estimate for each 1000 pseudo-observed simulations posterior parameter estimation, compared to the known parameter used for simulation. We considered 11 neurons in the hidden layer and a tolerance level of 0.01 (1000 simulations) for all posterior parameter estimation in this analysis as identified in *Appendix 1—table 1*. Plotted distributions for posterior parameter estimations can be found in *Figure 7—figure supplements 1–3*.

| SAO NICOLAU | Parameter posterior estimation | | | | 1000 cross-validation errors | | |
|---|---|---|---|---|---|---|---|
| Afr2P-Eur2P scenario parameters | Mode | Median | 50% Credibility Interval | 95% Credibility Interval | Mode mean absolute error | Median mean absolute error | 95% CI length accuracy |
| Ne.0 | 360 | 457 | [238 - 734] | [34 - 979] | 309 | 255 | 0.059 |
| Ne.20 | 28767 | 48793 | [25155–70475] | [3686–96753] | 27,502 | 23,452 | 0.055 |
| u.Ne | 0.135 | 0.238 | [0.133–0.358] | [0.025–0.483] | 0.132 | 0.113 | 0.057 |
| sAfr,0 | 0.3249 | 0.4730 | [0.2491–0.7393] | [0.0327–0.9752] | 0.3017 | 0.2461 | 0.051 |
| tAfr,p1 | 2.4 | 5.2 | [2.7–9.5] | [1.3–15.5] | 5.9 | 4.5 | 0.145 |
| sAfr,tAfr,p1 | 0.1695 | 0.3133 | [0.1584–0.5404] | [0.0171–0.9262] | 0.2897 | 0.2363 | 0.069 |
| tAfr,p2 | 14.6 | 14.5 | [13.2–14.9] | [8.3–16.4] | 1.5 | 1.4 | 0.522 |
| sAfr,tAfr,p2 | 0.7132 | 0.6398 | [0.4900–0.7736] | [0.0877–0.9380] | 0.1044 | 0.0981 | 0.055 |
| tEur,p1 | 2.7 | 6.7 | [3.5–10.8] | [2.0–16.1] | 4.6 | 3.5 | 0.139 |
| sEur,tEur,p1 | 0.1193 | 0.3240 | [0.1547–0.5711] | [0.0145–0.9460] | 0.2806 | 0.2394 | 0.048 |
| tEur,p2 | 19.2 | 19.0 | [17.7–19.4] | [10.3–19.8] | 3.3 | 2.9 | 0.199 |
| sEur,tEur,p2 | 0.0297 | 0.0731 | [0.0295–0.1465] | [0.0038–0.6642] | 0.2419 | 0.2067 | 0.065 |

*and Santos, 1995*; *Albuquerque and Santos, 2007*), and South West Central African Angolan Ovimbundu a small minority. Analogous to the Santo Antão inferences' limitations, our analyses cannot disentangle whether either or both African pulses in São Nicolau involved the limited South West Central African populations shared ancestry here identified.

Finally, we identify, much later during the 20th century, a weak event of European introgression for the genetic admixture history of São Nicolau (*Figure 7D*), similarly likely due to labor-induced migrations in the former Lusophone empire as for other islands, and in particular possibly due to the expansion of fishing and cannery industry in São Nicolau at this time (*Albuquerque and Santos, 2007*).

Altogether, our genetic data contain little identifiable information for the demographic history of this island (*Figure 7C*, *Figure 7—figure supplement 1*). This may be due to the recurrent demographic collapses of the permanent settlement in this island due to intense starvation events, epidemic outbreaks, and destructive pirate raids that affected the peopling of this island throughout history (*Figure 7—source data 1*). Nevertheless, one genetic pattern that was consistent with these recurrent demographic collapses is the high total levels of ROH observed in the data (*Figure 4* and *Appendix 4—figures 1–2*).

## Admixture history of Brava (First official settlement 1580; Perennial settlement: 1580)

### *MetHis*-ABC inferences detailed results for Brava

For individuals born on Brava island, we find that the most recent pulse of European admixture is relatively weak (median = 0.1697, mode = 0.1281, 95%CI=[0.0188–0.6504]), and occurred towards

the abolition of slavery in the Portuguese empire in the second half of the 19th century (generation median = 16.3 after the founding of Cabo Verde at generation 0, mode = 17.9, 95%CI=[4.2–19.3]), as indicated by posterior distributions of parameters for this event largely differing from their priors (*Figure 7*, *Figure 7—figure supplements 1–3*, *Appendix 5—table 5*). We identify a similarly intense pulse of admixture from Africa (median = 0.2473, mode = 0.1261, 95%CI=[0.0191–0.8266]), for which our inferences only bore indications that it occurred during the second half of the 18th century during the TAST, as the 95% CI for the timing of this event is wide (generation median = 13.6 after founding at generation 0, mode = 15.7, 95%CI=[2.4–19.2]).

More remotely in the past, we identify strong indications that a very weak European admixture pulse (median = 0.0352, mode = 0.0157, 95%CI=[0.0007–0.7239]) may have occurred at the beginning of the 17th century (generation median = 7.3 after founding, mode = 3.6, 95%CI=[2.8–16.1]). Furthermore, we identify a strong signal for an intense African admixture pulse (median = 0.6688, mode = 0.7403, 95%CI=[0.1358–0.9685]), occurring at the beginning of Cabo Verdean peopling history, during the early 16th century, at generation 2.8 after the founding of the archipelago at generation 0 (median, mode = 1.6, 95%CI=[1.1–12.2]), and thus largely before the first perennial settlement of this island.

Finally, our results strongly support a very limited effective size linear increase ($u_{Ne}$ median = 0.420, mode = 0.435, 95%CI=[0.243–0.494]) throughout the entire history of the island reaching 7337 effective individuals (median, mode = 3278, 95%CI=[647–73,114]) in the present.

**Appendix 5—table 5.** Brava NN-ABC posterior parameter estimations for Brava island, cross-validation posterior parameter errors, and cross-validation 95% Credibility Interval accuracies. Cross-validation errors and 95% CI accuracies are based on 1000 NN-ABC posterior parameter inferences using, in turn, each one of the 1000 simulations closest to the observed data used as pseudo-observed data, and the remaining 99,999 simulations as the reference table. Error calculations are based on the mode and, separately, the median point estimate for each 1000 pseudo-observed simulations posterior parameter estimation, compared to the known parameter used for simulation. We considered 10 neurons in the hidden layer and a tolerance level of 0.01 (1000 simulations) for all posterior parameter estimation in this analysis as identified in *Appendix 1—table 1*. Plotted distributions for posterior parameter estimations can be found in *Figure 7—figure supplements 1–3*.

| BRAVA | Parameter posterior estimation | | | | 1000 cross-validation errors | | |
|---|---|---|---|---|---|---|---|
| Afr2P-Eur2P scenario parameters | Mode | Median | 50% Credibility Interval | 95% Credibility Interval | Mode mean absolute error | Median mean absolute error | 95% CI length accuracy |
| Ne.0 | 495 | 501 | [309 - 696] | [63 - 919] | 308 | 254 | 0.068 |
| Ne.20 | 3278 | 7337 | [2908–17935] | [647–73114] | 31523 | 24989 | 0.056 |
| u.Ne | 0.435 | 0.420 | [0.374–0.455] | [0.243–0.494] | 0.077 | 0.073 | 0.050 |
| sAfr,0 | 0.7337 | 0.5666 | [0.3452–0.7698] | [0.0497–0.9737] | 0.2836 | 0.2405 | 0.059 |
| tAfr,p1 | 1.6 | 2.8 | [1.5–5.3] | [1.1–12.2] | 3.9 | 3.1 | 0.159 |
| sAfr,tAfr,p1 | 0.7403 | 0.6688 | [0.4987–0.7995] | [0.1353–0.9685] | 0.2418 | 0.2232 | 0.060 |
| tAfr,p2 | 15.7 | 13.6 | [8.4–16.3] | [2.4–19.2] | 2.8 | 2.7 | 0.069 |
| sAfr,tAfr,p2 | 0.1261 | 0.2473 | [0.1272–0.3647] | [0.0191–0.8266] | 0.2118 | 0.1905 | 0.058 |
| tEur,p1 | 3.6 | 7.3 | [4.1–11.1] | [2.8–16.1] | 3.4 | 2.9 | 0.204 |
| sEur,tEur,p1 | 0.0157 | 0.0352 | [0.0116–0.1161] | [0.0007–0.7239] | 0.2658 | 0.2338 | 0.064 |
| tEur,p2 | 17.9 | 16.3 | [12.5–18.0] | [4.2–19.3] | 3.3 | 3.1 | 0.054 |
| sEur,tEur,p2 | 0.1281 | 0.1697 | [0.1054–0.2554] | [0.0188–0.6504] | 0.1844 | 0.1716 | 0.057 |

## Discussion for the admixture history of Brava

We find an intense first African admixture pulse roughly three generations prior the first official and perennial peopling of the island in the 1580s (*Figure 7—source data 1*, *Figure 7*), thus

strongly suggesting that Brava was initially peopled by individuals already admixed and originating from another Cabo Verdean island, similarly to what was suggested for Fogo (see above). We hypothesize that Fogo admixed individuals were originally at the root of this peopling, as the patterns of shared haplotypic ancestry in Brava-born individuals strongly resemble that of Fogo, rather than Santiago, with shared ancestry exclusively with Senegambian Mandinka and in similar proportion, as well as a significant (albeit very reduced) signal for a shared ancestry with Tuscan TSI otherwise only found in Fogo (*Figure 3*). We find a limited introgression event from Europe during the late 17th century, possibly consistent with a known immigration event from Madeira and the Açores, and from Fogo after a volcano eruption, around this time (*Albuquerque and Santos, 1995*). We also find a relatively weak African admixture pulse at some point during the height of the TAST for which the timing remains poorly inferred, and which will require further investigations.

Finally, we identify a more intense pulse of European admixture shortly after the abolition of slavery in the Portuguese empire in the second half of the 19th century. This later event may be due to the expansion of the whaling industry at this time, particularly in Brava due to favorable maritime currents and routes for this industry, which brought European and European-North American sailors to the island and increased locally contacts with Europe, and North America. Most interestingly, this historical scenario is also consistent with the signal, virtually unique across Cabo Verde, of a significant albeit reduced shared haplotypic ancestry between Brava-born individuals and the British GBR.

Notably, our results indicate that the effective population of Brava remained low during the entire peopling history of Cabo Verde and until today (*Figure 7C*), consistent with this island being the smallest inhabited and the most south-west ward in the archipelago (*Figure 7—source data 1*). Surprisingly, in spite of the historically low effective population size, Brava shows low total ROH levels and relatively short ROH lengths (*Figure 4* and *Appendix 4—figures 1–2*), which remains to be elucidated in the future by considering larger sample sizes from the island.

## Admixture history of Maio (First official settlement 1529; Perennial settlement: 1650)

### *MetHis*-ABC inferences detailed results for Maio

For individuals born on Maio island, we find evidence that the most recent pulse of European admixture was of relatively modest intensity (median = 0.1472, mode = 0.1134, 95%CI=[0.0233–0.5957]), and likely occurred during the 18th century, in the midst of the TAST (generation median = 12.7 after founding at generation 0, mode = 14.1, 95%CI=[3.7–17.9]), as both posterior distributions substantially departe from their respective priors (*Figure 7*, *Figure 7—figure supplements 1–3*, *Appendix 5—table 6*). Although our RF-ABC scenario-choice favors several pulses of admixture from both sources, our NN-ABC posterior parameter estimation fails to capture sufficient information to infer satisfactorily the most recent pulse of admixture from Africa, as both the posterior distribution of the timing of this event and its intensity seldom depart from their respective priors and have wide 95% CI.

However, we capture substantial information strongly suggesting two, much older, admixture pulses from Europe and Africa, respectively, occurring virtually synchronically in the middle 16th century (generation median = 4.1 after founding for the European pulse, mode = 2.3, 95%CI=[1.7–14.2]; generation median = 4.2 after founding for the African pulse, mode = 2.4, 95%CI=[1.5–12.8]), thus much earlier than the first perennial settlement of this island dated in 1650 by historians (*Figure 7—source data 1*). Furthermore, we find that the respective intensities of the two pulses (median = 0.3260 for the European pulse, mode = 0.1075, 95%CI=[0.0117–0.9497]; median = 0.5589 for the African pulse, mode = 0.7703, 95%CI=[0.0346–0.9769]), sum close to 100%. This suggests a large replacement of the Maio population at that time, with indications that these synchronic admixture events occur on the basis of a founding admixed population largely of African origin (median $s_{Afr,0}$=0.6757, mode = 0.9135, 95%CI=[0.017–0.421]).

Finally, the demographic inference of reproductive size changes in Maio indicates, similarly as for Brava, that the reproductive size of the island remained relatively small during its entire history, and only very recently increased up to a median 21,454 effective individuals (mode = 7159), albeit with a large 95% CI ([914–93,656]).

**Appendix 5—table 6.** MaioNN-ABC posterior parameter estimations for Maio island, cross-validation posterior parameter errors, and cross-validation 95% Credibility Interval accuracies. Cross-validation errors and 95% CI accuracies are based on 1000 NN-ABC posterior parameter inferences using, in turn, each one of the 1000 simulations closest to the observed data used as pseudo-observed data, and the remaining 99,999 simulations as the reference table. Error calculations are based on the mode and, separately, the median point estimate for each 1000 pseudo-observed simulations posterior parameter estimation, compared to the known parameter used for simulation. We considered 10 neurons in the hidden layer and a tolerance level of 0.01 (1000 simulations) for all posterior parameter estimation in this analysis as identified in *Appendix 1—table 1*. Plotted distributions for posterior parameter estimations can be found in *Figure 7—figure supplements 1–3*.

| MAIO | Parameter posterior estimation | | | | 1000 cross-validation errors | | |
|---|---|---|---|---|---|---|---|
| Afr2P-Eur2P scenario parameters | Mode | Median | 50% Credibility Interval | 95% Credibility Interval | Mode mean absolute error | Median mean absolute error | 95% CI length accuracy |
| Ne.0 | 122 | 500 | [241 - 746] | [29 - 980] | 276 | 240 | 0.059 |
| Ne.20 | 7159 | 21454 | [7388–49176] | [914–93656] | 30301 | 25378 | 0.072 |
| u.Ne | 0.080 | 0.121 | [0.071–0.210] | [0.017–0.421] | 0.082 | 0.078 | 0.064 |
| sAfr,0 | 0.9135 | 0.6757 | [0.3486–0.8841] | [0.0177–0.9941] | 0.3150 | 0.2509 | 0.061 |
| tAfr,p1 | 2.4 | 4.2 | [2.5–6.8] | [1.5–12.8] | 4.1 | 3.2 | 0.126 |
| sAfr,tAfr,p1 | 0.7703 | 0.5589 | [0.2768–0.7758] | [0.0346–0.9769] | 0.2630 | 0.2350 | 0.051 |
| tAfr,p2 | 14.4 | 11.1 | [6.6–15.1] | [2.4–18.8] | 2.5 | 2.4 | 0.112 |
| sAfr,tAfr,p2 | 0.6543 | 0.5276 | [0.2918–0.7126] | [0.0452–0.9393] | 0.2168 | 0.1937 | 0.052 |
| tEur,p1 | 2.3 | 4.1 | [2.2–7.4] | [1.7–14.2] | 3.2 | 2.8 | 0.219 |
| sEur,tEur,p1 | 0.1075 | 0.3260 | [0.1376–0.5818] | [0.0117–0.9497] | 0.3014 | 0.2415 | 0.068 |
| tEur,p2 | 14.1 | 12.7 | [8.7–14.9] | [3.7–17.9] | 3.3 | 3.1 | 0.048 |
| sEur,tEur,p2 | 0.1134 | 0.1472 | [0.0957–0.2301] | [0.0233–0.5957] | 0.2145 | 0.1899 | 0.057 |

## Discussion for the admixture history of Maio

Our results for the admixture history of Maio indicate a relatively simple scenario with a single pulse of European and African synchronic admixture having occurred in the mid-16th century in another Cabo Verdean island, long before the first perennial settlement of the island in 1650, similarly to Fogo or Brava. We hypothesize that this event occurred on Santiago, as shared local haplotypic ancestry patterns are highly resembling between islands (*Figure 3*), and also consistently with Maio being located very close to Santiago and historical records showing unequivocally the strong relationships between islands throughout history (*Carreira, 2000*; *Albuquerque and Santos, 1991*; *Albuquerque and Santos, 1995*; *Albuquerque and Santos, 2007*).

After the descendants of this initial admixture event settled Maio, they experienced only a single identifiable admixture pulse of weak intensity from Europe, which likely occurred at the turn between the 18th and the 19th century during the TAST, and no clearly identifiable further signal of African admixture in the island. As for some other Cabo Verde islands, we cannot conclude about the timing of the African admixture event that brought the reduced signal of West Central and South West Central Africa shared haplotypic ancestry with Maio, observed in our ADMIXTURE and SOURCEFIND analyses (*Figure 3*).

Finally, the reduced population size throughout the history of Maio is consistent with the island being located close to Santiago and with much less water resources, having made it difficult to maintain and expand the settlement on this island, despite its first peopling having occurred early in the history of Cabo Verde almost exclusively dedicated to free-range cattle herding (*Carreira, 2000*; *Albuquerque and Santos, 1991*; *Albuquerque and Santos, 1995*).

ROH patterns are also consistent with this history, with high total ROH indicating long term isolation and population bottlenecks. Similar to Santiago there were some outlier individuals with

much higher European ancestry within ROH compared to the remainder of the genome (*Figure 4* and *Appendix 4—figures 1–2*). This could possibly reflect historical marriage prohibitions and other social constraints against intercommunity reproductions (see **Discussion** in main text).

## Admixture history of Boa Vista (First official settlement 1566; Perennial settlement: 1650)

### *MetHis*-ABC inferences detailed results for Boa Vista

For the admixture history of individuals born on Boa Vista, we find an intense admixture pulse from Africa (median = 0.6553, mode = 0.7516, 95%CI=[0.1262–0.8949]), that occurred in the mid-19th century (generation median = 15.0 after Cabo Verde founding at generation 0, mode = 14.8, 95%CI=[11.0–17.4]), during the abolition of the TAST, with posterior distributions for both these parameters substantially departing from their respective priors and relatively reduced 95% CI (*Figure 7*, *Figure 7—figure supplements 1–3*, *Appendix 5—table 7*). Prior to that event, we fail to estimate the timing and intensity of the older African admixture pulse into Boa Vista, as posterior parameter distributions seldom depart from their respective priors for this event.

However, our results strongly support a scenario with an intense period of recurring monotonically decreasing admixture from Europe (median admixture intensity at the period's start = 0.8057, mode = 0.9219, 95%CI=[0.2736–0.9907]; median admixture intensity at the period's end = 0.340, mode = 0.120, 95%CI=[0.021–0.880]), between the first official settlement of the island in 1566 (generation median at the start of the European period of admixture = 3.9 after founding, mode = 2.2, 95%CI=[1.7–11.8]), until shortly after its first perennial peopling dated in 1650 (generation median at the end of the European period of admixture = 8.2, mode = 8.6, 95%CI=[3.1–14.9]). Interestingly, the initial founding admixture event at the root of the genetic patterns of Boa Vista-born individuals supports a substantially lower amount of African admixture (median = 0.3917, mode = 0.1706, 95%CI=[0.1814–0.6678]), compared to the rest of Cabo Verde where this parameter could be reliably identified.

Finally, our results show that the demography of Boa Vista remained relatively constant throughout the history of Cabo Verde, and indicate a possible very recent increase in effective size, albeit posterior parameter estimation of the most recent effective size was ambiguous due to large 95% CI (median $Ne_{20}$=47,744, mode = 13,128, 95%CI=[2657–98,587]).

**Appendix 5—table 7.** Boa Vista NN-ABC posterior parameter estimations for Boa Vista island, cross-validation posterior parameter errors, and cross-validation 95% Credibility Interval accuracies. Cross-validation errors and 95% CI accuracies are based on 1000 NN-ABC posterior parameter inferences using, in turn, each one of the 1000 simulations closest to the observed data used as pseudo-observed data, and the remaining 99,999 simulations as the reference table. Error calculations are based on the mode and, separately, the median point estimate for each 1000 pseudo-observed simulations posterior parameter estimation, compared to the known parameter used for simulation. We considered 8 neurons in the hidden layer and a tolerance level of 0.01 (1000 simulations) for all posterior parameter estimation in this analysis as identified in *Appendix 1—table 1*. Plotted distributions for posterior parameter estimations can be found in *Figure 7—figure supplements 1–3*.

| BOA VISTA | Parameter posterior estimation | | | | 1,000 cross-validation errors | | |
|---|---|---|---|---|---|---|---|
| Afr2P-Eur Rec. scenario parameters | Mode | Median | 50% Credibility Interval | 95% Credibility Interval | Mode mean absolute error | Median mean absolute error | 95% CI length accuracy |
| Ne.0 | 731 | 627 | [357 - 793] | [57 - 977] | 263 | 230 | 0.049 |
| Ne.20 | 13128 | 47744 | [20486–76777] | [2657–98587] | 30386 | 25219 | 0.074 |
| u.Ne | 0.056 | 0.079 | [0.051–0.125] | [0.014–0.346] | 0.067 | 0.066 | 0.059 |
| sAfr,0 | 0.1706 | 0.3917 | [0.1814–0.6678] | [0.0188–0.9677] | 0.3186 | 0.2517 | 0.061 |
| tAfr,p1 | 14.6 | 10.4 | [5.8–14.2] | [2.4–16.8] | 4.6 | 3.7 | 0.137 |
| sAfr,tAfr,p1 | 0.2027 | 0.3760 | [0.1998–0.5640] | [0.0261–0.8771] | 0.2494 | 0.2213 | 0.056 |

*Appendix 5—table 7 Continued on next page*

*Appendix 5—table 7 Continued*

| BOA VISTA | Parameter posterior estimation | | | | 1,000 cross-validation errors | | |
|---|---|---|---|---|---|---|---|
| Afr2P-Eur Rec. scenario parameters | Mode | Median | 50% Credibility Interval | 95% Credibility Interval | Mode mean absolute error | Median mean absolute error | 95% CI length accuracy |
| tAfr,p2 | 14.8 | 15.0 | [14.2–16.0] | [11.0–17.4] | 1.7 | 1.7 | 0.113 |
| sAfr,tAfr,p2 | 0.7516 | 0.6553 | [0.4758–0.7671] | [0.1262–0.8949] | 0.1962 | 0.1801 | 0.048 |
| tEur,p1 | 2.2 | 3.9 | [2.3–6.6] | [1.7–11.8] | 3.7 | 3.1 | 0.192 |
| tEur,p2 | 8.6 | 8.2 | [6.0–10.2] | [3.1–14.9] | 3.3 | 3.2 | 0.090 |
| sEur,tEur,p1 | 0.9219 | 0.8057 | [0.6300–0.9141] | [0.2736–0.9907] | 0.2723 | 0.2319 | 0.076 |
| sEur,tEur,p2 | 0.120 | 0.340 | [0.1643–0.5832] | [0.021–0.880] | 0.1721 | 0.1579 | 0.071 |
| u.sEur | 0.4444 | 0.2783 | [0.151–0.398] | [0.0160–0.4907] | 0.151 | 0.128 | 0.065 |

## Discussion for the admixture history of Boa Vista

Altogether, our results for the admixture history of Boa Vista born-individuals indicate a largely differing admixture history compared to all other islands of the archipelago. First, we find that the admixture event at the root of the genetic peopling of the island, although occurring long before the first perennial peopling of the island similarly to other islands such as Fogo, Brava, and Maio, is largely biased towards European admixture, as opposed to all other islands in the archipelago (*Figure 7D*). Second, we find the beginning of a period of recurring admixture from Europe before the mid-16th century, concomitantly with the first official settlement of the island, and until its first perennial settlement in the 1650s (*Hakluyt, 2014*)[pp.529]. Interestingly, this corresponds to a period when Boa Vista has been used by Cabo Verde as a penitentiary island for non-enslaved individuals which may explain the recurring admixture process here identified (*Carreira, 2000*; *Albuquerque and Santos, 1991*; *Albuquerque and Santos, 1995*).

Later-on towards the end of the TAST in the 19th century, Boa Vista experienced an intense pulse of African admixture, similarly to Santiago and Saõ Nicolau and probably for similar socio-cultural reasons (see **Discussion** in the main text). However, overall limited historical records specific to this island render speculative the interpretation of our results based on genetics only (*Figure 7—source data 1*).

Finally, the effective size of Boa Vista remained constant and low throughout the history of this island (*Figure 7C*), which echoes its' small historical census sizes, mainly due to limited water resources on the island, up until today (*Figure 7—source data 1*). ROH patterns are also consistent with this history, with the highest total ROH among the Cabo Verdean islands indicating long term isolation and population bottlenecks (Figures F4 and *Appendix 4—figures 1–2*).

## Admixture history of São Vicente (First official settlement 1570; Perennial settlement: 1780)

### *MetHis*-ABC inferences detailed results for São Vicente

For the admixture history of individuals born on São Vicente, we find a recent European admixture pulse of substantial intensity (median = 0.2840, mode = 0.2596, 95%CI=[0.0397–0.8044]), having likely occurred towards the end of the 19th century (generation median = 17.4 after founding of Cabo Verde at generation 0, mode = 17.7, 95%CI=[10.6–18.9]); both posterior distributions have relatively narrow 95% CI and clearly depart from their respective priors (*Figure 7*, *Figure 7—figure supplements 1–3*, *Appendix 5—table 8*).

This most recent admixture event is preceded by a period of recurrent decreasing African admixture likely spanning the second half of the most intense period of the TAST starting in the early 18th century (generation median at the start of the African period of admixture = 9.7, mode = 12.6, 95%CI=[2.5–16.9]), and ending during the end of the TAST and the abolition of slavery in the Portuguese Empire in the mid-19th century (generation median at the end of the African period of admixture = 15.8, mode = 17.5, 95%CI=[7.4–19.1]), albeit our results are ambiguous for the estimate of the start of this admixture period as indicated by the large 95% CI. Furthermore, our results are also ambiguous to estimate the intensity of the African introgression during the period of recurring

admixture, as the posterior distributions for the onset and offset intensities, as well as the shape of the monotonic recurring admixture do not clearly depart from their respective priors and had large 95%CIs.

We find traces of an older European admixture pulse of substantial intensity (median = 0.3403, mode = 0.1772, 95%CI=[0.0149–0.9133]), which likely occurred at the beginning of the 17th century (generation median = 6.4 after founding, mode = 2.8, 95%CI=[1.9–15.1]). Our results indicate that this first admixture event occurred on a root-population that exhibited a large African admixture proportion (median $s_{Afr,0}$=0.5983, mode = 0.8596, 95%CI=[0.0529–0.9835]), albeit this result should be considered with cautious as CI is overall wide.

Finally, our results clearly indicate a strong effective population expansion since the perennial peopling of the island in the late 18th century, up to 66,715 effective individuals today (median, mode = 85,838, 95%CI=[12,159–98,375]).

**Appendix 5—table 8.** São Vicente NN-ABC posterior parameter estimations for São Vicente island, cross-validation posterior parameter errors, and cross-validation 95% Credibility Interval accuracies. Cross-validation errors and 95% CI accuracies are based on 1000 NN-ABC posterior parameter inferences using, in turn, each one of the 1000 simulations closest to the observed data used as pseudo-observed data, and the remaining 99,999 simulations as the reference table. Error calculations are based on the mode and, separately, the median point estimate for each 1000 pseudo-observed simulations posterior parameter estimation, compared to the known parameter used for simulation. We considered 11 neurons in the hidden layer and a tolerance level of 0.01 (1000 simulations) for all posterior parameter estimation in this analysis as identified in *Appendix 1—table 1*. Plotted distributions for posterior parameter estimations can be found in *Figure 7—figure supplements 1–3*.

| SAO VICENTE | Parameter posterior estimation | | | | 1,000 cross-validation errors | | |
|---|---|---|---|---|---|---|---|
| Afr Rec.- Eur2P scenario parameters | Mode | Median | 50% Credibility Interval | 95% Credibility Interval | Mode mean absolute error | Median mean absolute error | 95% CI length accuracy |
| Ne.0 | 109 | 262 | [115 - 486] | [19 - 918] | 288 | 243 | 0.061 |
| Ne.20 | 85838 | 66715 | [42741–84818] | [12159–98375] | 29947 | 23721 | 0.048 |
| u.Ne | 0.215 | 0.244 | [0.174–0.338] | [0.072–0.469] | 0.094 | 0.084 | 0.049 |
| sAfr,0 | 0.8596 | 0.5983 | [0.3334–0.8142] | [0.0529–0.9835] | 0.3231 | 0.2499 | 0.063 |
| tAfr,p1 | 12.6 | 9.7 | [5.7–13.1] | [2.5–16.9] | 3.9 | 3.4 | 0.153 |
| tAfr,p2 | 17.5 | 15.8 | [13.8–17.6] | [7.4–19.1] | 2.8 | 2.7 | 0.235 |
| sAfr,tAfr,p1 | 0.6796 | 0.7168 | [0.5698–0.8525] | [0.2806–0.9795] | 0.2031 | 0.1774 | 0.079 |
| sAfr,tAfr,p2 | 0.2149 | 0.2786 | [0.1561–0.4266] | [0.0256–0.7820] | 0.1538 | 0.1457 | 0.063 |
| u.sAfr | 0.072 | 0.228 | [0.111–0.360] | [0.015–0.484] | 0.161 | 0.126 | 0.056 |
| tEur,p1 | 2.8 | 6.4 | [3.1–10.0] | [1.9–15.1] | 4.1 | 3.3 | 0.214 |
| sEur,tEur,p1 | 0.1772 | 0.3403 | [0.1733–0.5442] | [0.0149–0.9133] | 0.2448 | 0.2126 | 0.055 |
| tEur,p2 | 17.7 | 17.4 | [16.3–18.0] | [10.6–18.9] | 2.0 | 2.1 | 0.143 |
| sEur,tEur,p2 | 0.2596 | 0.2840 | [0.1732–0.4177] | [0.0397–0.8044] | 0.1585 | 0.1498 | 0.059 |

## Discussion for the admixture history of São Vicente

Our results for the admixture history of individuals born on São Vicente indicate a strong European admixture event concomitant with the first official peopling of the island occurring at the same time as the nearby island of Santo Antão, consistently with historical records (*Figure 7—source data 1*). However, the perennial peopling of this island was difficult due to the very limited water resources, in particular compared to the much more hospitable, larger, and nearby Santo Antão. Thus, São Vicente essentially served as a coal and salt harbor depot during most of its history with a minimal

settlement (*Albuquerque and Santos, 2007*), which may explain the initial European admixture pulse we identify (*Figure 7*), and our local haplotypic shared ancestry results (*Figure 3*).

Later-on and unique in Cabo Verde, our results support a long period of recurring African admixture occurring during the entire second half of the TAST and ending with the abolition of slavery in the Portuguese empire. However, only the final admixture event of this period could be precisely inferred to have occurred during the mid-19th[h] century during the abolition of the TAST and of slavery in the Portuguese empire (*Figure 7D*). The small haplotypic ancestry shared between São Vicente-born individuals and West Central and/or South West Central African population may stem from illegal slave-trade known to have surged during this period. Nevertheless, the poor performances of our parameter inference for the other parameters of this recurring admixture period may indicate that more complex admixture scenarios than the ones considered occurred in this island, thus begging for future work further complexifying admixture histories to be reconstructed with ABC.

Interestingly, historical data show that São Vicente census only very recently started to increase, with newly available water sources and a protected deep-water harbor in the bay of Mindelo, unique in Cabo Verde, which favored the massive economic expansion of the island in the late 19th century and throughout the 20th century (*Figure 7—source data 1*, *Duarte et al., 2015*[p.36]). Furthermore, the end of the agro-slavery system and end of plantation economy in Cabo Verde at the end of the 18th century and beginning of the 19th century are known historically to have triggered emigration from Santo Antão and rural-urban migrations during the 19th and 20th century, essential for São Vicente perennial peopling (*Albuquerque and Santos, 2007*; *Patterson, 1988*). This is further consistent with extensive familial grand-parental birth-places in Santo Antão and extensive familial relationships reported by São Vicente-born individuals in our familial anthropology interviews.

Altogether, these historical, economic, and demographic processes may well explain the onset of the large reproductive population increase identified in our analyses (*Figure 7C*), as well as haplotypic local ancestry patterns largely resembling those obtained for Santo Antão (*Figure 3*). Furthermore, they are consistent with the substantial European admixture pulse identified at the turn of the 19th and 20th century with our analyses. Reflecting the historical small population size, São Vicente had high total levels of ROH (*Figure 4*). Interestingly, individuals from this island, on average, show more African ancestry in ROH compared to the remainder of the genome. This could possibly reflect historical marriage prohibitions or the historically high levels of recurrent African admixture, or both.

## Admixture history of Sal (First official settlement 1529; Perennial settlement: 1841)

### *MetHis*-ABC inferences detailed results for Sal

Finally, the admixture history of individuals born on Sal inferred with NN-ABC is overall ambiguous a posteriori (*Figure 7*, *Figure 7—figure supplements 1–3*, *Appendix 5—table 9*). Indeed, while our RF-ABC scenario-choice leaned towards a complex admixture history with a period of recurring admixture from Europe and two African admixture pulses after the initial founding admixture event for the population of Sal, the duration and intensity of the European admixture period, as well as the first African admixture pulse, all showed posterior parameter distributions seldom departing from their priors. Nevertheless, we clearly identify a recent African admixture pulse of moderate intensity (median = 0.2401, mode = 0.1780, 95%CI=[0.0197–0.6590]), which likely occurred at the end of the 19th century (generation median = 17.6 after the founding of Cabo Verde at generation 0, mode = 17.9, 95%CI=[12.5–19.0]). Furthermore, our results indicate a strong and very recent effective size expansion in Sal, up to 63,289 individuals (median, mode = 88,248, 95%CI=[5689–98,181]).

### Discussion for the admixture history of Sal

Sal was among the early Cabo Verdean islands to be officially settled and exploited, in 1529 (*Figure 7—source data 1*), as a source of sea-salt both exported and employed locally in the free-range herding meat-production and tannery industry at the historical root of Cabo Verde economy with Europe and Africa (*Albuquerque and Santos, 1991*; *Patterson, 1988*). However, its perennial peopling dated only from 1841, the last perennially peopled island of Cabo Verde, as water sources and agricultural surfaces are scarce on this island. This slow demographic expansion is well captured by our inference showing a reduced population size during most Cabo Verde history on Sal, followed very recently by a strong reproductive size increase (*Figure 7C*). Furthermore, in this context, our results clearly identifying only a recent admixture pulse of African origin at the beginning of the

**Appendix 5—table 9.** Sal NN-ABC posterior parameter estimations for Sal island, cross-validation posterior parameter errors, and cross-validation 95% Credibility Interval accuracies. Cross-validation errors and 95% CI accuracies are based on 1000 NN-ABC posterior parameter inferences using, in turn, each one of the 1000 simulations closest to the observed data used as pseudo-observed data, and the remaining 99,999 simulations as the reference table. Error calculations are based on the mode and, separately, the median point estimate for each 1000 pseudo-observed simulations posterior parameter estimation, compared to the known parameter used for simulation. We considered 11 neurons in the hidden layer and a tolerance level of 0.01 (1000 simulations) for all posterior parameter estimation in this analysis as identified in *Appendix 1—table 1*. Plotted distributions for posterior parameter estimations can be found in *Figure 7—figure supplements 1–3*.

| SAL | Parameter posterior estimation | | | | 1000 cross-validation errors | | |
|---|---|---|---|---|---|---|---|
| Afr2P-EurRec. scenario parameters | Mode | Median | 50% Credibility Interval | 95% Credibility Interval | Mode mean absolute error | Median mean absolute error | 95% CI length accuracy |
| Ne.0 | 525 | 518 | [285 - 745] | [41 - 966] | 306 | 250 | 0.061 |
| Ne.20 | 88248 | 63289 | [36457–82778] | [5689–98181] | 30007 | 24554 | 0.058 |
| u.Ne | 0.038 | 0.076 | [0.035–0.155] | [0.005–0.399] | 0.112 | 0.102 | 0.056 |
| sAfr,0 | 0.8853 | 0.5820 | [0.3164–0.8077] | [0.0247–0.9820] | 0.3039 | 0.2508 | 0.060 |
| tAfr,p1 | 3.2 | 9.0 | [4.8–13.1] | [1.8–16.8] | 4.9 | 4.1 | 0.122 |
| sAfr,tAfr,p1 | 0.7261 | 0.5166 | [0.2893–0.7374] | [0.0302–0.9626] | 0.2667 | 0.2275 | 0.063 |
| tAfr,p2 | 17.9 | 17.6 | [16.3–18.0] | [12.5–19.0] | 1.3 | 1.2 | 0.212 |
| sAfr,tAfr,p2 | 0.1780 | 0.2401 | [0.1374–0.3655] | [0.0197–0.6590] | 0.1588 | 0.1462 | 0.071 |
| tEur,p1 | 3.9 | 9.3 | [5.3–12.6] | [3.3–16.8] | 4.2 | 3.4 | 0.180 |
| tEur,p2 | 17.8 | 16.2 | [13.9–17.8] | [6.8–19.4] | 3.7 | 3.5 | 0.102 |
| sEur,tEur,p1 | 0.4931 | 0.5657 | [0.3810–0.7810] | [0.1100–0.9754] | 0.2585 | 0.2183 | 0.072 |
| sEur,tEur,p2 | 0.0730 | 0.2061 | [0.0842–0.4034] | [0.0113–0.7804] | 0.1926 | 0.1734 | 0.076 |
| u.sEur | 0.137 | 0.207 | [0.102–0.346] | [0.011–0.486] | 0.156 | 0.128 | 0.061 |

20[th] century could reflect the recent economic migrations of Western African populations to this island, during the development of its salt mining activities and its travel and tourism activities since then. Nevertheless, our scenario-choice results suggested the occurrence of a period of recurrent European admixture in Sal, which our posterior parameter inference largely failed to identify. This is possibly due to the complex recent peopling history of Sal that we fail to capture with genetic data, which will need to be clarified in the future in particular with additional samples from this island. Nevertheless, reflecting the historical small population size, Sal has high total levels of ROH (*Figure 4*).

**Appendix 5—table 10.** Cabo Verde NN-ABC posterior parameter estimations for 225 Cabo Verde-born individuals considered as a single random-mating population, cross-validation posterior parameter errors, and cross-validation 95% Credibility Interval accuracies.

Cross-validation errors and 95% CI accuracies are based on 1000 NN-ABC posterior parameter inferences using, in turn, each one of the 1000 simulations closest to the observed data used as pseudo-observed data, and the remaining 99,999 simulations as the reference table. Error calculations are based on the mode and, separately, the median point estimate for each 1000 pseudo-observed simulations posterior parameter estimation, compared to the known parameter used for simulation. We considered five neurons in the hidden layer and a tolerance level of 0.01 (1000 simulations) for all posterior parameter estimation in this analysis as identified in *Appendix 1—table 1*. Plotted distributions for posterior parameter estimations can be found in *Figure 7—figure supplements 1–3*.

| CABO VERDE Parameter posterior estimation | | | | | 1000 cross-validation errors | | |
|---|---|---|---|---|---|---|---|
| Afr2P-Eur2P scenario parameters | Mode | Median | 50% Credibility Interval | 95% Credibility Interval | Mode mean absolute error | Median mean absolute error | 95% CI length accuracy |
| Ne.0 | 343 | 490 | [267 - 726] | [34 - 968] | 296 | 238 | 0.058 |
| Ne.20 | 86901 | 61995 | [36958–82637] | [8050–98176] | 26496 | 23321 | 0.047 |
| u.Ne | 0.469 | 0.400 | [0.310–0.463] | [0.172–0.497] | 0.118 | 0.107 | 0.041 |
| sAfr,0 | 0.5801 | 0.5015 | [0.2560–0.7308] | [0.0271–0.9794] | 0.3000 | 0.2464 | 0.057 |
| tAfr,p1 | 3.1 | 9.0 | [5.0–13.0] | [2.0–18.0] | 5.7 | 4.2 | 0.090 |
| sAfr,tAfr,p1 | 0.4597 | 0.4831 | [0.2575–0.7249] | [0.0263–0.9644] | 0.2556 | 0.2310 | 0.049 |
| tAfr,p2 | 19.2 | 18.6 | [17.2–19.3] | [11.9–19.3] | 1.3 | 1.2 | 0.153 |
| sAfr,tAfr,p2 | 0.7056 | 0.6224 | [0.4317–0.7576] | [0.1164–0.9303] | 0.1483 | 0.1379 | 0.037 |
| tEur,p1 | 2.4 | 6.1 | [3.0–9.3] | [2.0–16.1] | 4.4 | 3.4 | 0.146 |
| sEur,tEur,p1 | 0.1193 | 0.4300 | [0.2038–0.7022] | [0.0216–0.9718] | 0.3006 | 0.2487 | 0.056 |
| tEur,p2 | 18.2 | 15.1 | [11.2–17.9] | [4.1–19.0] | 3.1 | 2.7 | 0.086 |
| sEur,tEur,p2 | 0.1882 | 0.4150 | [0.2029–0.6780] | [0.0313–0.9647] | 0.2471 | 0.2115 | 0.051 |

