## [Editor Report]

This study leverages genetic and linguistic data from the islands of Cabo Verde, and provides a valuable example of how genetic ancestry patterns vary across admixed populations due in part to their unique local history and social practices of that time. The empirical and computational analyses supporting the claims of the authors are solid, and the tools developed will be useful for the study of genetically admixed individuals. The work will be of interest to human evolutionary biologists and anthropologists.

---

## [Decision Letter]

**Decision letter after peer review:**

Thank you for submitting your article "The admixture histories of Cabo Verde" for consideration by *eLife*. Your article has been reviewed by 2 peer reviewers, including Emilia Huerta-Sanchez as Reviewing Editor and Reviewer #1, and the evaluation has been overseen by Molly Przeworski as the Senior Editor. The following individual involved in the review of your submission has agreed to reveal their identity: Xinjun Zhang (Reviewer #2).

Essential revisions:

Both reviewers enjoyed reading the paper and found the results of this study to be important as it provides insights into how admixture shapes patterns of genetic variation. Both reviewers would like to see a revised version of the manuscript. There were some clarifications and a few additional analyses that the reviewers feel could strengthen the paper. These include:

1) The authors infer demographic history using a previously developed method (meTHIS), and it would be great to see the model robustness when fixed parameters are misspecified: the ancestral population sizes, recent gene flow between Europeans and Africans (eg. back-to-Africa migrations), and the possibility of population size fluctuation in Cabo Verde in recent history (or at least, more evidence to justify a continuous population growth).

2) The authors suggest that longer ROH in one ancestry implies more admixture; is a higher proportion of admixture the only way to generate these ROH patterns (instead of a severe bottleneck in the ancestral population)?

3) Revealing that different island populations in Cabo Verde have different admixture histories provides support for variation in the way in which these islands were founded. The authors propose a set of interesting hypotheses for how the different islands were founded; could the authors use simulations to see if there is some support for these hypotheses?

4) The authors could motivate why they chose the methods applied to the data.

5) Please review and respond to the individual reviewer comments as well.

*Reviewer #1 (Recommendations for the authors):*

Line 333: "thus illustrating the strong signal of genetic isolation-by-distance (51) within Cabo Verde at very reduced geographical scales." Is this consistent/similar with what is observed for other island populations?

Line 335: It will be helpful for readers to include the definition of "utterance" in the main text.

Figure F6: What is h?

Methods:

Line 856: "We collapsed the local ancestry assignments for each SNP in each Cabo Verdean individuals hence obtained into" I didn't get what "collapsed" means in this context. Also, I thought that RFmix has a phasing step, so the phase that is given as input may be different from the phase it infers. So how do the authors deal with regions that are heterozygous in ancestry (e.g. one chromosome is African and the other is European) with their calls of runs of homozygosity for their ancestry-specific ROH sizes?

Line: 916: "we first design MetHis v1.0 (37) forward-in-time simulations of four competing" Could the authors say more about their forward-in-time simulations? Are these simulations created by the authors? It seems the authors are using real data. How did the authors make 20000 haploid genomes from the Mandinka and IBS who have really small sample sizes? Could they say more here to explain how these forwards in time simulations are different than what we typically use (e.g. SLIM)? The real data is not sequencing data, so how did the authors recreate that in the simulations? I assume that might affect their ROH calling and ancestry calling?

*Reviewer #2 (Recommendations for the authors):*

There are a number of areas where the paper can improve and strengthen.

Specific revision suggestions:

1) Title: the current title "the admixture histories of Cabo Verde" is underwhelming and does not capture the main findings of the paper. The authors can consider a different title that reflects the admixture history reconstructed by revealing the co-shifting of genetic and sociocultural diversity, which is storytelling about TAST.

2) The genetic relationship analyses

a. the authors used MDS to draw relationships between Cabo Verdeans and other European and African continental populations. I am not an MDS expert (nor do I expect the general audience to be), and it is not clear to me why ADS-MDS is used here instead of a traditional PCA in pair with classic admixture statistical tests such as F statistics. Do we learn anything new from ADS-MDS that PCA and F stats can't tell us? Would like to see more explanation and justification in the text. It would also be helpful to include F statistic tests and how they compare to the MDS results, at least in the Supplementary Figures.

b. Could the authors comment on the proportion of missing data in the genotypes, and whether the removal of them is expected to change MDS results in any way?

c. Furthermore, I find Figure F2 very hard to read as I need to constantly scroll back to Figure F1 for color legends – the authors should consider either (1) adding legends to Figure F2, or (2) merging Figure F1 and F2 as the same figure, highlighting the current Figure F2 panel A-C in the context of geographical locations in Figure F1, and move the other panels to Supplementary Figures.

3) Admixture analysis:

a. In Verdu et al. 2017 Current Biology, the authors performed similar MDS and ADMIXTURE analyses and made similar conclusions on Cabo Verdeans being related to Senegambians and Southern Europeans. What new information are we learning from here? Are we only confirming the previous observations on all CV islands? In any case, if there are novel discoveries from here that I am missing from reading the text, the authors should at least consider clarifying and re-emphasizing the messages.

b. For the ADMIXTURE analysis, could the authors explain the justification for choosing the maximum value of K=10? What would the plot look like when K is larger?

c. The authors mentioned that Asian populations are being compared with Cabo Verdeans in the admixture analyses, but I don't see them being displayed on the plot (Figure F3) – in general, is there any evidence there's ancestry contribution in CV from populations outside of Europe and Africa? How would the admixture plot look like when other worldwide populations are included?

4) ROH and local ancestry inference

a. What are the proportions of long ROHs among all ROHs in these island populations?

b. The authors interpreted the finding of more long-ROHs exclusively in one ancestry as essentially an increase in admixture proportion from the respective source population, either through recurring admixtures or relatively recent admixture events. However, I wonder if two source populations contributed equally to the admixture, but one source experienced more bottleneck themselves and therefore carry more long-ROH in the first place, would that lead to the enrichment of long ROHs in that ancestry? Could the authors run some simulations to test whether alternative model(s) (eg. different ancestral population sizes, admixture proportions and times, etc.) could lead to a similar distribution of long ROH and their overlapping with respective ancestry?

c. Additionally, I wonder if the long ROHs in these island populations are enriched in low recombination regions or functionally important regions. And if so, would the authors expect any of such factors could affect the distribution of ROHs and the interpretations thereafter? Furthermore, how would the authors expect the difference in fine-scale recombination rate (and mutation rates) between Africans and Europeans possibly affect the ROH and ancestry distributions?

d. For Figure F4D – it's a bit hard to read the differences here as these violin plots appear uniformly distributed across populations despite slight differences. For example, is the overrepresentation of ancestries in certain populations significant, such as European contribution in Fogo and Brava and African contribution in Sal and Sao Vicente? Could the authors provide some p-values here?

e. Line 358-359: why is it surprising to find that individuals from distant islands differ in African admixture level if we already see that their ADMIXTURE-revealed ancestry components differ between islands?

5) Estimation of admixture history

a. From what I see, the RF model considers many admixture-related parameters but doesn't consider the evolutionary history in the source populations before 20 generations ago. Can the authors comment on how robust the RF model is to demographic model misspecifications, especially related to the ancestral population size in the source populations? Most importantly, are the ABC posteriors sensitive to the demographic parameters that are fixed in the simulations?

b. For the recipient population, the population sizes are projected to be increasing over the 20 generations in all 4 admixture scenarios – is that realistic according to the historical record? Was there any reduction in Ne in Cabo Verde islands during the TAST? How is the inference robust to alternative models where the admixed population's size fluctuates over time?

c. Also, would the historical gene flows between Southern Europeans and Northern Africans and within Africa (eg. Moorjani et al. 2011 Plos Genetics; Busby et al. 2016 *eLife*; Botigue et al. 2013 PNAS) confound the inference of admixture history?

---

## [Author Response]

Essential revisions:Both reviewers enjoyed reading the paper and found the results of this study to be important as it provides insights into how admixture shapes patterns of genetic variation. Both reviewers would like to see a revised version of the manuscript. There were some clarifications and a few additional analyses that the reviewers feel could strengthen the paper. These include:(1) The authors infer demographic history using a previously developed method (meTHIS), and it would be great to see the model robustness when fixed parameters are misspecified: the ancestral population sizes, recent gene flow between Europeans and Africans (eg. back-to-Africa migrations), and the possibility of population size fluctuation in Cabo Verde in recent history (or at least, more evidence to justify a continuous population growth).

We thank both reviewers and the editor for pointing out that these much-needed aspects were lacking in the previous version of our manuscript. To address these points, we deeply modified our manuscript, in particular in the material and methods, results, and Discussion sections, and added a novel MetHis-ABC analysis. Detailed modifications are presented below and in the detailed answers to each reviewer specifically. Altogether, we believe our manuscript strongly benefited from these major changes.

Note that all the sections and line-references of the changes brought to our initial version indicated in the detailed answers below refer to the FULL Article version (including all appendix and supplementary figures and tables in a single document) of our re-submission.

1. We clarified our methods’ design and discussed extensively its limitations with respect to ancestral populations’ sizes mis-specifications. Indeed, ancestral source population sizes are not modelized in our MetHis-ABC approach. Instead, we consider that the observed proxy source populations from Africa and Europe are at the drift-mutation equilibrium and are large since the initial and recent founding of Cabo Verde in the 1460’s. We thus use observed genetic variation patterns in these populations to build virtual gamete reservoirs for the admixture history of Cabo Verde with the MetHis-ABC framework. Therefore, while we cannot evaluate explicitly the influence of ancestral source population sizes differences on our inferences in Cabo Verde, as we now state in the revised version of our manuscript: “we nevertheless implicitly take the real demographic histories of these source populations into account in our simulations, as we use observed genetic patterns themselves the product of this demographic history to create the virtual source populations at the root of the admixture history of each Cabo Verdean island.”. We then discuss the outcome of such an approach which mimics satisfactorily the real data for ABC inference. See in particular the revised versions of the Material and Methods L1454-1491 novel section “Simulating the admixed population from source-populations for 60,000 independent SNPs with MetHis”, and Results L637-649.

2. Concerning the possibilities for population-size changes in the admixed population in our simulations and ABC inferences, we clarified our Material and Methods and explanations of our Results to better show that we readily consider various possible scenarios (for each island separately). Indeed, with our MetHis simulation design, given values of model-parameters correspond either to a constant, a linearly increase, or a hyperbolic increase in reproductive size in the admixed population over time. We further clarified our Results and Discussion pointing out that we find, a posteriori, indeed, different demographic regimes among islands.

Nevertheless, reviewers are right that we did not test the possibility for bottlenecks. We thus substantially expanded the Results and Discussion sections in multiple locations to highlight this limitation and the challenges involved in overcoming it in future work. See in particular Material and Methods L1386-1404 section “Hyperbolic increase, linear increase, or constant reproductive population size in the admixed population”, Results L739-742, and Discussion L934-941, and Perspectives.

3. In the revised version of the manuscript, we now extensively discuss the important point raised by the reviewers concerning possible recent gene-flow between Europe and Africa which may have influenced genetic patterns among the proxy populations at the source of admixture in Cabo Verde, and how they may affect our inferences. See in particular Discussion L903-916.

4. Finally, in order to evaluate the robustness of our inferences, we conducted a novel MetHis-ABC inference considering all Cabo Verde-born individuals in a single random-mating population in addition to the analyses conducted separately for each island of birth. Most interestingly, we find that our ABC inferences fail to accurately reconstruct the detailed admixture history of Cabo Verde when considered as a whole instead of per each island of birth separately. This is due to admixture histories substantially differing across islands of birth of individuals, also consistent with the significantly differentiated genetic patterns within Cabo Verde obtained from ADMIXTURE, local-ancestry inferences, ROH, and isolation-by-distance analyses. These results are now implemented throughout the revised version of the manuscript and in supplementary figures and tables. See in particular Results L758-769, and Appendix1-figures and tables, Figure7—figure supplement 1-3, and Appendix 5-table 10.

(2) The authors suggest that longer ROH in one ancestry implies more admixture; is a higher proportion of admixture the only way to generate these ROH patterns (instead of a severe bottleneck in the ancestral population)?

We thank the reviewer for this comment and clarified our text to avoid such suggestion that we did not intend originally. In particular, see discussion of ROH patterns: Discussion Lines 1020-1024.

(3) Revealing that different island populations in Cabo Verde have different admixture histories provides support for variation in the way in which these islands were founded. The authors propose a set of interesting hypotheses for how the different islands were founded; could the authors use simulations to see if there is some support for these hypotheses?

The reviewers are perfectly right that formally testing the serial founder model here hypothesized is of major interest, and we gratefully thank them for pointing the lack of such discussion in the previous version of our manuscript. This is a challenging task due to the vast amounts of models that would need to be formally tested for each possible routes of series of founding events for each island in the archipelago. We now explicitly discuss this important point and refer to future work in the revised version of our manuscript; in particular in Discussion L942-953 and in Perspectives L1066-1072, as we are currently preparing a novel investigation dedicated specifically to this task.

(4) The authors could motivate why they chose the methods applied to the data.

We extensively modified our manuscript in both Results and Material and Methods, to clarify and justify our reasoning for each one of the analyses conducted, and to discuss pros and cons of the methods used. We thank the reviewers for this request, as we believe it allowed us to strongly improve the accessibility of our work in particular for the less specialized audience, as well as equally crucially improve replicability of our work for specialists. See in particular Results L185-193, L245-250, L368-371, L380-386, L495—511, L567-571, L606-621, and the corresponding Material and Methods sections.

(5) Please review and respond to the individual reviewer comments as well.

We thank the reviewers for their detailed comments and suggestions, and respond to them individually and in details below.

Reviewer #1 (Recommendations for the authors):Line 333: "thus illustrating the strong signal of genetic isolation-by-distance (51) within Cabo Verde at very reduced geographical scales." Is this consistent/similar with what is observed for other island populations?

The reviewer is right that this original result of our article was not sufficiently contextualized nor discussed in the previous version of the manuscript. We developed these aspects throughout the revised version of the manuscript. See Abstract, Introduction L145-153, Results L495-511, and the corresponding Discussion section.

Line 335: It will be helpful for readers to include the definition of "utterance" in the main text.

We clarified the revised version of our manuscript according to the reviewer’s comment in particular in the Results L506-508.

Figure F6: What is h?

*h*_g_ is the proportion of parents of individuals in the admixed population at generation g coming from the admixed population itself at the previous generation. We clarified this point directly in the legend of Figure 6 as well as in Material and Methods.

Methods:Line 856: "We collapsed the local ancestry assignments for each SNP in each Cabo Verdean individuals hence obtained into" I didn't get what "collapsed" means in this context.

Any ancestry call that was assigned a population from the African continent was assigned a category of “AFR”. Any ancestry call that was assigned a population from the European continent was assigned a category of “EUR”. And, finally, any ancestry call that was assigned a population from the Asian continent was assigned a category of “ASN”. We have clarified this in different parts of the text, and in particular in Material and Methods L1282-1285.

Also, I thought that RFmix has a phasing step, so the phase that is given as input may be different from the phase it infers. So how do the authors deal with regions that are heterozygous in ancestry (e.g. one chromosome is African and the other is European) with their calls of runs of homozygosity for their ancestry-specific ROH sizes?

Regions of heterozygous ancestry within an ROH were rare. We provide this result in Figure4-source data 3, which we comment in the Material and Methods of the main text L1297-1298.

For Figure 4C, these regions are excluded from the plot. For Figure 4D, these regions were counted as fractionally contributing to either AFR or EUR totals. We have added text to clarify this in Material and Methods L1308-1316 in addition to the novel supplementary source-data tables.

Line: 916: "we first design MetHis v1.0 (37) forward-in-time simulations of four competing" Could the authors say more about their forward-in-time simulations? Are these simulations created by the authors? It seems the authors are using real data. How did the authors make 20000 haploid genomes from the Mandinka and IBS who have really small sample sizes? Could they say more here to explain how these forwards in time simulations are different than what we typically use (e.g. SLIM)? The real data is not sequencing data, so how did the authors recreate that in the simulations? I assume that might affect their ROH calling and ancestry calling?

We have considerably clarified the revised version of our Material and Methods as well as Results to address the points raised by the reviewer. Indeed, we now thoroughly explain how the forward-in-time simulations individual-centered are conducted with *MetHis* L1454-1486, which shows that our simulation algorithm is equivalent to what is implemented in SLIM and other forward-in-time simulators, as shown in the original *MetHis* paper (Fortes-Lima et al. 2021).

Next, we clarified in the revised version of the manuscript that we simulate only independent genotype data, and that all summary statistics are based on these data, and thus do not include ROH calculations for the ABC inference. In particular we moved the summary-statistics Table originally presented in the Supplementary Material into the main text (Table 3 in the revised version) of our paper to further avoid the legitimate confusion raised by the reviewer.

Reviewer #2 (Recommendations for the authors):There are a number of areas where the paper can improve and strengthen.Specific revision suggestions:(1) Title: the current title "the admixture histories of Cabo Verde" is underwhelming and does not capture the main findings of the paper. The authors can consider a different title that reflects the admixture history reconstructed by revealing the co-shifting of genetic and sociocultural diversity, which is storytelling about TAST.

We now propose for title:

A genetic and linguistic analysis of the admixture histories of the islands of Cabo Verde

(2) The genetic relationship analysesa. the authors used MDS to draw relationships between Cabo Verdeans and other European and African continental populations. I am not an MDS expert (nor do I expect the general audience to be), and it is not clear to me why ADS-MDS is used here instead of a traditional PCA in pair with classic admixture statistical tests such as F statistics. Do we learn anything new from ADS-MDS that PCA and F stats can't tell us? Would like to see more explanation and justification in the text. It would also be helpful to include F statistic tests and how they compare to the MDS results, at least in the Supplementary Figures.

As stated above, concerning ASD-MDS versus classical PCA, we clarified our choice in the main text of the article in Results L185-192.

Furthermore, as recommended by the reviewer, we now explain and discuss Patterson’s f3 methods and provide novel results in Results L376-386 and corresponding Figure 3—figure supplement 2.

b. Could the authors comment on the proportion of missing data in the genotypes, and whether the removal of them is expected to change MDS results in any way?

We found a fraction of 7.0x10^-4^ missing genotypes on average per individual within island of birth in Cabo Verde (SD=3.0x10^-4^ across islands), among the 455,705 autosomal SNPs in our working dataset used for the ASD-MDS analysis, which should not affect our results. Indeed, even if missing data do not overlap the same SNPs for a given pair of individual, they should represent, on average per individual pairwise comparison, in the order of 1000 un-used SNPs per ASD calculus (and thus more than 450,000 SNPs used). In-turn, such very low levels of missingness will not be expected to have a major effect on the final MDS projection of the ASD matrix. We have added a sentence to clarify this point in the Material and Methods L1120-1121.

c. Furthermore, I find Figure F2 very hard to read as I need to constantly scroll back to Figure F1 for color legends – the authors should consider either (1) adding legends to Figure F2, or (2) merging Figure F1 and F2 as the same figure, highlighting the current Figure F2 panel A-C in the context of geographical locations in Figure F1, and move the other panels to Supplementary Figures.

According to the reviewer’s legitimate concern, we modified Figure 2 to show only Cabo Verdean panels alongside the legend for population locations and symbols, and moved other panels concerning ASW, ACB and PUR, to a novel Figure 2—figure supplement 1. We updated the former Table ST1 (now Figure 1-source data 1) information for population inclusion in each analysis accordingly.

(3) Admixture analysis:a. In Verdu et al. 2017 Current Biology, the authors performed similar MDS and ADMIXTURE analyses and made similar conclusions on Cabo Verdeans being related to Senegambians and Southern Europeans. What new information are we learning from here? Are we only confirming the previous observations on all CV islands? In any case, if there are novel discoveries from here that I am missing from reading the text, the authors should at least consider clarifying and re-emphasizing the messages.

We substantially modified this section of the previous version of the manuscript to better emphasize the novelty of our findings for the ADMIXTURE analyses with respect to previous results, in order to clarify the reviewer’s legitimate concern (in particular at L266-281).

Furthermore, we also clarified the novelty of our approach in the following local-ancestry inference section, as such approach was not endeavored in previous studies across Cabo Verdean islands. Indeed, note that this LAI approach formally tests and identifies the respective genetic contribution from varied possible source populations to the gene-pool of admixed individuals, which is not the case of ASD-MDS nor ADMIXTURE analyses, a point that we further clarified in the revised version of the manuscript such as in Results L368-386.

b. For the ADMIXTURE analysis, could the authors explain the justification for choosing the maximum value of K=10? What would the plot look like when K is larger?

We thank the reviewer for this suggestion. We now provide ADMIXTURE results until K=15 and describe the obtained results in the corresponding Result L289-314. Furthermore, we extended the text in corresponding Appendix 3 and Appendix 3-figure 1 to describe all alternative ADMIXTURE solutions identified until K=15.

In short here, this extended ADMIXTURE analysis allowed us to identify additional genetic sub-structures within Cabo Verde, differentiating five genetic clusters specific to different groups of islands of birth within the archipelago. These patterns are further consistent with our next results concerning Isolation-By-Distance across Cabo Verdean islands, both results which were largely unsuspected from previous studies, including ours.

c. The authors mentioned that Asian populations are being compared with Cabo Verdeans in the admixture analyses, but I don't see them being displayed on the plot (Figure F3) – in general, is there any evidence there's ancestry contribution in CV from populations outside of Europe and Africa?

The Chinese CHB sample can be found at the extreme left of the ADMIXTURE analyses in Figure 3. We moved the previous supplementary text concerning the interpretations of this result to the main text Results L305-310 to address the legitimate concern of the reviewer.

How would the admixture plot look like when other worldwide populations are included?

ADMIXTURE analyses capture the same information and recapitulate higher-order of pairwise individual variation from ASD-MDS or PCA analyses, as we now clarified in particular in Results L245-250. Therefore, to answer the reviewer’s interesting comment we expect the ADMIXTURE plots for a worldwide dataset to resemble, for lower values of K, what we identified here in the worldwide ASD-MDS analyses presented in Appendix 2 figures 1-4: *K*=5 would mainly differentiate African, European, Asian, American, and Oceanian individuals, and recently admixed populations would be found with intermediate membership proportions’ patterns between their historical sources. However, conducting such analysis is computationally intensive for very large worldwide datasets and has already been conducted in previous studies, we prefer not to include them in the current article as we believe such analysis is outside our specific focus of interest.

(4) ROH and local ancestry inferencea. What are the proportions of long ROHs among all ROHs in these island populations?

We now report the mean proportion of total long ROH among all called ROH. See Results L422-423 and a novel supplemental Figure 4-source data 1.

b. The authors interpreted the finding of more long-ROHs exclusively in one ancestry as essentially an increase in admixture proportion from the respective source population, either through recurring admixtures or relatively recent admixture events. However, I wonder if two source populations contributed equally to the admixture, but one source experienced more bottleneck themselves and therefore carry more long-ROH in the first place, would that lead to the enrichment of long ROHs in that ancestry? Could the authors run some simulations to test whether alternative model(s) (eg. different ancestral population sizes, admixture proportions and times, etc.) could lead to a similar distribution of long ROH and their overlapping with respective ancestry?

We thank the reviewer for this comment and clarified our text to avoid such suggestion that we did not intend originally. We have thus added more nuanced language to our discussion of ROH patterns, Discussion L1020-1024.

c. Additionally, I wonder if the long ROHs in these island populations are enriched in low recombination regions or functionally important regions. And if so, would the authors expect any of such factors could affect the distribution of ROHs and the interpretations thereafter? Furthermore, how would the authors expect the difference in fine-scale recombination rate (and mutation rates) between Africans and Europeans possibly affect the ROH and ancestry distributions?

As we mention in the answer to the reviewers above, we note that in Figure 8 Pemberton 2012 (AJHG 91:275-292) shows that occurrence of long ROH at the same genomic location across individuals is correlated with low recombination rates, although the effect is relatively weak unless in extreme recombination cold spots. Unless there were many extreme recombination cold spots that were different among the islands or ancestral populations, we anticipate fine-scale recombination rate differences not to matter very much for total ROH levels in these data. Similarly, we do not expect large genome-wide differences in mutation rate, and therefore we don’t anticipate minor local variation in mutation rates to make a systematic difference in total ROH levels. We briefly clarified these important points in the revised version of the manuscript in particular in Results L414-415.

d. For Figure F4D – it's a bit hard to read the differences here as these violin plots appear uniformly distributed across populations despite slight differences. For example, is the overrepresentation of ancestries in certain populations significant, such as European contribution in Fogo and Brava and African contribution in Sal and Sao Vicente? Could the authors provide some p-values here?

We have now performed a permutation test to assess which of these patterns among the islands are significant. For each individual in each island, we randomly permuted the location of all long ROH (ensuring that no permuted ROH overlap), re-computed the local AFR ancestry proportion falling within these permuted ROH, and then subtracted the global ancestry proportion. We then take the mean of this difference across all individuals for each island and repeat the process 10,000 times. As there is negligible ASN ancestry across these individuals, the AFR and EUR proportions essentially add to 1, and therefore we consider an over/under representation of AFR ancestry in long ROH to be equivalent to an under/over representation of EUR ancestry in long ROH, respectively. This permutation test shows that AFR ancestry in long ROH is under represented (and therefore EUR ancestry in long ROH is over represented) on Fogo, whereas on Sao Vicente and Santo Antão there is an over representation of AFR ancestry in long ROH (and therefore an under representation of EUR ancestry in long ROH). We have added this information to the manuscript, in Results L432-441 and Material and Methods L1308-1316. We have also updated Figure 4D with the significance information added in detail in Figure 4-source data 2.

e. Line 358-359: why is it surprising to find that individuals from distant islands differ in African admixture level if we already see that their ADMIXTURE-revealed ancestry components differ between islands?

We thank the reviewer for pointing out this unclear passage. We reshaped the introduction to this section, as well as the Results in multiple locations to clarify the importance and originality of these findings. We have thus profoundly reshaped large parts of our text as can be seen in the revised version of our manuscript in the Results sections 5 and 6 starting L495 and L567 respectively, and in Discussion L942-L953.

(5) Estimation of admixture historya. From what I see, the RF model considers many admixture-related parameters but doesn't consider the evolutionary history in the source populations before 20 generations ago. Can the authors comment on how robust the RF model is to demographic model misspecifications, especially related to the ancestral population size in the source populations? Most importantly, are the ABC posteriors sensitive to the demographic parameters that are fixed in the simulations?

As mentioned above in the answers to reviewers:

1. We clarified our methods’ design and discussed extensively its limitations with respect to ancestral populations’ sizes mis-specifications. Indeed, ancestral source population sizes are not modelized in our MetHis-ABC approach. Instead, we consider that the observed proxy source populations from Africa and Europe are at the drift-mutation equilibrium and are large since the initial and recent founding of Cabo Verde in the 1460’s, and thus use observed genetic variation patterns in these populations to build virtual gamete reservoirs for the admixture history of Cabo Verde with the MetHis-ABC framework. Therefore, while we cannot evaluate explicitly the influence of ancestral source population sizes differences on our inferences in Cabo Verde, as we now state in the revised version of our manuscript: “we nevertheless implicitly take the real demographic histories of these source populations into account in our simulations, as we use observed genetic patterns themselves the product of this demographic history to create the virtual source populations at the root of the admixture history of each Cabo Verdean island.”. We then discuss the outcome of such an approach which mimics satisfactorily the real data for ABC inference. See in particular the revised versions of the Material and Methods L1454-1491 novel section “Simulating the admixed population from source-populations for 60,000 independent SNPs with MetHis”, and Results L637-649.

2. Then, we now extensively discussed the important point raised by the reviewer about the possible admixture history in the African and European populations at the historical source of Cabo Verde, having possibly occurred before the colonization of the archipelago, or even after that. See in particular Discussion L903-916.

3. Finally, in order to evaluate the robustness of our inferences, we conducted a novel MetHis-ABC inference considering all Cabo Verde-born individuals in a single random-mating population in addition to the analyses conducted separately for each island of birth. Most interestingly, we find that our ABC inferences fail to accurately reconstruct the detailed admixture history of Cabo Verde when considered as a whole instead of per each island of birth separately. This is due to admixture histories substantially differing across islands of birth of individuals, also consistent with the significantly differentiated genetic patterns within Cabo Verde obtained from ADMIXTURE, local-ancestry inferences, ROH, and isolation-by-distance analyses. These results are now implemented throughout the revised version of the manuscript and in supplementary figures and tables. See in particular Results L758-769, and Appendix1-figures and tables, Figure7—figure supplement 1-3, and Appendix 5-table 10.

b. For the recipient population, the population sizes are projected to be increasing over the 20 generations in all 4 admixture scenarios – is that realistic according to the historical record? Was there any reduction in Ne in Cabo Verde islands during the TAST? How is the inference robust to alternative models where the admixed population's size fluctuates over time?

As mentioned above in response to the reviewer’s legitimate and accurate comment, concerning the possibilities for population-size changes in the admixed population in our simulations and ABC inferences, we clarified our Material and Methods and explanations of our Results to better show that we readily consider various possible scenarios (for each island separately). Indeed, with our MetHis simulation design, given values of model-parameters correspond either to a constant, a linearly increasing, or a hyperbolic increase in reproductive size in the admixed population over time. We further clarified our Results and Discussion pointing out that we find, a posteriori, indeed, different demographic regimes among islands.

Concerning the reduction of Ne in the history of Cabo Verde, historical data are scarce for detailed censuses before the 1860’s (see Figure 7-source data 1, former Supplementary Table ST5). While there have unquestionably been some events that likely impacted census sizes before then, such as epidemics or hunger, it is not obvious how they might have impacted effective sizes as population sizes were altogether small until very recently. Our historical table in Figure 7-source data 1 provides details of the known changes in census sizes over the entire history of Cabo Verde and the only things that are precisely known are the very strong very recent increase in census in Cabo Verde mostly during the 20^th^ century.

Nevertheless, reviewers are right that we did not test the possibility for bottlenecks. We thus substantially expanded the Results and Discussion sections in multiple locations to highlight this limitation and the challenges involved in overcoming it in future work. See in particular Material and Methods L1386-1404 section “Hyperbolic increase, linear increase, or constant reproductive population size in the admixed population”, Results L739-742, and Discussion L934-941, and Perspectives.

c. Also, would the historical gene flows between Southern Europeans and Northern Africans and within Africa (eg. Moorjani et al. 2011 Plos Genetics; Busby et al. 2016 eLife; Botigue et al. 2013 PNAS) confound the inference of admixture history?

We thank the reviewer for this interesting discussion point that we overlooked in the previous version of our manuscript. In our revised version, we now explicitly discuss it in a novel paragraph Discussion L903-L916, and in Appendix 2.